# Foundation Models in Robotics: A Comprehensive Review of Methods, Models, Datasets, Challenges and Future Research Directions

**Aggelos Psiris**                                                                  *aggelospsiris@hua.gr*
*Department of Informatics and Telematics, Harokopio University of Athens, Athens, Greece*

**Vasileios Argyriou**                                              *Vasileios.Argyriou@kingston.ac.uk*
*Department of Networks and Digital Media, Kingston University, London, United Kingdom*

**Evangelos K. Markakis**                                                          *emarkakis@hmu.gr*
*Department of Electrical and Computer Engineering, Hellenic Mediterranean University, Heraklion, Greece*

**Panagiotis Sarigiannidis**                                                 *psarigiannidis@uowm.gr*
*Department of Electrical and Computer Engineering, University of Western Macedonia, Kozani, Greece*

**Efstratios Gavves**                                                                  *e.gavves@uva.nl*
*University of Amsterdam, Amsterdam, Netherlands*
*Archimedes, Athena Research Center, Athens, Greece*

**Kostas Bekris**                                                                   *kb572@cs.rutgers.edu*
*Computer Science Department, Rutgers University, New Brunswick, NJ, USA*

**Arash Ajoudani**                                                              *Arash.Ajoudani@iit.it*
*Istituto Italiano di Tecnologia, Genova, Italy*

**Georgios Th. Papadopoulos**                                            *g.th.papadopoulos@hua.gr*
*Department of Informatics and Telematics, Harokopio University of Athens, Athens, Greece*
*Archimedes, Athena Research Center, Athens, Greece*

**Reviewed on OpenReview:** *https://openreview.net/forum?id=qF7vdPrpPk*

## Abstract

Over the recent years, the field of robotics has been undergoing a transformative paradigm shift from fixed, single-task, domain-specific solutions towards adaptive, multi-function, general-purpose agents, capable of operating in complex, open-world, dynamic environments. This tremendous advancement is primarily driven by the emergence of Foundation Models (FMs), i.e., large-scale neural-network architectures trained on massive, internet-scale, heterogeneous datasets that provide unprecedented capabilities in multi-modal understanding/reasoning, long-horizon planning, and cross-embodiment generalization. In this context, the current study provides a holistic, thorough, systematic, and in-depth review of the research landscape of FMs in robotics. In particular, the evolution in the field is initially delineated through five distinct research phases, spanning from the early incorporation of native Natural Language Processing (NLP) and Computer Vision (CV) models to the current frontier of multi-sensory generalization and real-world deployment. Subsequently, a highly-granular, multi-criteria, taxonomic investigation of the literature methods is performed, examining the following key aspects: a) The employed foundation model types (i.e., LLMs, VFMs, VLMs, and VLAs), b) The underlying neural network architectures, c) The adopted learning paradigms, d) The different learning stages of knowledge incorporation, e) The most common robotic tasks (including perception, planning, navigation, manipulation, and human-robot interaction), and

f) The main real-world application domains. For each defined criterion/aspect, a methodical comparative analysis of the various categories of approaches and critical insights are provided. Moreover, a synthesis of the publicly available datasets, required for model training and evaluation, is provided, organized around the main recurring dataset families along with their typical uses and current gaps. Furthermore, a comprehensive and hierarchical discussion on the current open challenges and promising future research directions in the field is incorporated.

# 1 Introduction

Over the recent years, the field of robotics has witnessed unprecedented and transformative technological advancements, which have boosted the evolution from fixed, single-task, domain-specific setups to adaptive, general-purpose, open-world solutions (Newbury et al., 2023; Mokssit et al., 2023). Apart from significant developments in hardware and material sciences, a key driving force for this technological rise comprises the progress in the fields of Artificial Intelligence (AI) and Machine Learning (ML) (Soori et al., 2023). In particular, nowadays robotic platforms exhibit increased levels of efficiency, dexterity, autonomy, precision, and adaptation across a wide set of diverse tasks, while operating in complex and dynamic environments (Billard et al., 2025).

Robotic research has so far been dominated by two main (though not mutually exclusive) modeling paradigms, namely automatic control and machine learning approaches (Lee et al., 2024b). Classic automation control relies on the fundamental principle of initially defining a mathematical model of a system for predicting its behavior and, subsequently, designing a controller for enabling it to perform a specific task. Such approaches (often termed model-based) require explicit programming and have been proven to be efficient for implementing tasks in structured and predictable environments (Lee et al., 2024b). However, they are characterized by low adaptability (reprogramming is needed) and they are typically mathematically complex (Rakhmatillaev et al., 2025). On the other hand, ML methods focus on enabling robots to learn from data and experiences. To this end, ML approaches are shown to exhibit high adaptability (e.g., tackling novel and previously unseen circumstances) and to be efficient in handling tasks in complex, unstructured, and dynamic (or even unknown) environments (Tang et al., 2025). Nevertheless, ML methods are often computationally expensive and typically require large datasets for training purposes (Hu et al., 2023).

Foundation Models (FMs) constitute a recent and, yet, very powerful paradigm in the fields of AI and ML (Awais et al., 2025). In particular, FMs are constructed through training on massive, internet-scale, multi-modal datasets and can be adapted to a wide range of diverse downstream tasks, such as language, vision, and audio processing. In practice, FMs serve as a versatile and reusable basis for efficiently developing specialized or domain-specific (multi-task) applications, avoiding the need for training from scratch and using extensive training datasets. More recently, FMs have also been introduced in the field of robotics, exhibiting the following, among others, advantageous characteristics (that extend the ones of traditional ML-based approaches) (Firoozi et al., 2025; Xiao et al., 2025): a) Improved transferability across related tasks, environments, and embodiments, b) More reusable and generalizable representations, c) Increased semantic understanding and open-world capabilities, d) Support for sim-to-real transfer and cross-domain adaptation, e) Multi-modal integration and semantic alignment, f) Enhanced versatility in language-conditioned and perception-driven robotic behavior, g) Interpretation of natural language instructions, h) Hierarchical and long-horizon task decomposition and planning, and i) Improved policy generalization. The above favorable attributes, though, are accompanied by critical/unique challenges, including, indicatively (Firoozi et al., 2025; Xiao et al., 2025): a) Inference latency and high computational cost, b) Limited real-time deployability, c) Lack of semantic and physical grounding, d) Data scarcity and embodiment bias, e) Safety risks and unforeseen failure modes, f) Limited interpretability, transparency, and diagnosability, and g) Ethical, alignment, and regulatory imperatives.

As outlined above, FMs induce transformative effects on and lead to unprecedented performance/capability accomplishments in the field of robotics, fundamentally reforming robot design, learning, programming, and deployment practices. In this context, the current study aims to holistically and comprehensively

Table 1: Comparative analysis of recent surveys in the field of foundation models in robotics.

| Article | Strengths | Limitations |
|---|---|---|
| Hu et al. (2023) (arXiv) | • Broad scope
• Empirical per-task performance meta-analysis
• CV/NLP vs. native FM separation | • Early works only
• No systematic comparison
• Misses recent advances
• No application domains |
| Xu et al. (2024c) (arXiv) | • Manipulation focus
• Planning/control taxonomy | • Narrow scope
• Limited coverage
• No systematic comparison
• No empirical benchmarking
• No application domains |
| Ma et al. (2024c) (arXiv) | • VLA/embodied-AI focus
• Systematic per-category comparison | • Only VLAs
• Not multi-criteria
• No empirical benchmarking
• No application domains |
| Jang et al. (2024) (IJCAS) | • Autonomy focus
• Perception/planning/control taxonomy
• Platforms & simulators | • Not multi-criteria
• No theoretical comparison
• No empirical benchmarking
• No application domains |
| Kawaharazuka et al. (2024) (AR) | • FM-component replacement focus
• Input-output analysis | • Not multi-criteria
• No theoretical comparison
• No empirical benchmarking
• No application domains |
| Firoozi et al. (2025) (IJRR) | • Broad robotics & embodied-AI scope
• Decision/planning/control analysis | • Not multi-criteria
• No theoretical comparison
• No empirical benchmarking
• No application domains |
| Xiao et al. (2025) (Neurocomputing) | • Robot-learning focus
• Systematic hierarchical analysis | • Not multi-criteria
• No theoretical comparison
• No empirical benchmarking
• No application domains |
| Tayyab Khan & Waheed (2025) (arXiv) | • Deployment/integration focus
• Sim-to-real
• Feasibility analysis | • Not multi-criteria
• No theoretical comparison
• No empirical benchmarking
• No application domains |
| Kawaharazuka et al. (2025) (IEEE Access) | • VLA full-stack (hardware & software)
• Architecture & learning analysis | • Only VLAs
• Coarse categorization
• No theoretical comparison
• No empirical benchmarking
• No application domains |
| Sapkota et al. (2025) (arXiv) | • VLA evolution focus
• Application domains covered | • Only VLAs
• Not multi-criteria
• No theoretical comparison
• No empirical benchmarking |
| **Current survey** | • Holistic & systematic review (6 databases, screening)
• 5 research evolution phases
• 6-criteria taxonomy
• Per-criterion comparison & insights
• Challenges & future directions | • No empirical benchmarking
• Snapshot of a fast-evolving field |

investigate, map, and analyze in depth the research landscape of robotic FM methods. In particular, the main contributions of this work are:

- Outline of the **evolution in robotic FM research**, focusing on describing the most common FMs proposed in the literature and the main observed phases, which comprise the following ones: a) Phase 1 (2018-2021): Integration of native Natural Language Processing (NLP) and Computer Vision (CV)

models; b) Phase 2 (2021-2022): Grounded planning with Vision-Language (VL) representations; c) Phase 3 (2022-2023): Embodied Vision-Language-Action (VLA) policies; d) Phase 4 (2023-2024): Memory, autonomous task composition, and Web-to-robot transfer; and e) Phase 5 (2024-present): Multi-sensory generalization and real-world deployment;

- Holistic, thorough, systematic, highly-granular, multi-criteria, **taxonomic investigation of robotic FM approaches**, taking into account the following main criteria: a) The type of the employed FM with respect to the input-output modalities involved; b) The nature of the underlying Neural Network (NN) architecture; c) The learning paradigm adopted for developing a robotic FM; d) The learning stage at which knowledge is incorporated to a FM; e) The task controlled by a robotic FM; and f) The application domain where a robotic FM is used. For each defined criterion, a methodical comparative analysis of the various categories of approaches and critical insights are provided;

- Synthesis of the **public datasets/benchmarks** used for training/evaluation purposes, organized around the main recurring dataset families, along with their typical uses and current gaps;

- Extensive discussion of **current challenges** and **future research directions** in the field.

Regarding existing surveys in the field, Table 1 comparatively analyzes the current work with the relevant literature reviews of (Hu et al., 2023; Xu et al., 2024c; Ma et al., 2024c; Jang et al., 2024; Kawaharazuka et al., 2024; Firoozi et al., 2025; Xiao et al., 2025; Tayyab Khan & Waheed, 2025; Kawaharazuka et al., 2025; Sapkota et al., 2025), summarizing their main strengths and limitations. Examining Table 1, it can be seen that literature works exhibit the following limitations: a) They often remain relatively specific/narrow in scope (i.e., adopting a task-, model-, application-, learning-, or integration-oriented perspective), leading to a non-thorough investigation of the overall research landscape, b) They consider a single or very few literature analysis criteria, resulting into a non-comprehensive examination of the literature works, c) They commonly adopt a narrative-style discussion of the literature, avoiding to provide a systematic comparison of the different approaches, and d) with the exception of the meta-analysis of Hu et al. (2023), they do not include any empirical/experimental benchmarking of the surveyed methods. On the contrary, the current survey provides a thorough and in-depth analysis of the research landscape of the use of FMs in robotics, demonstrating the following key advantageous/distinctive characteristics: a) It targets a holistic investigation of the overall field, b) It adopts a structured and systematic literature review methodology, supporting the search in 6 major databases, the use of specific inclusion/exclusion criteria, and the application of an iterative screening process, c) It documents the main 5 distinct research evolution phases, as well as the key trends associated with each of them, d) It supports a highly-granular, multi-criteria (6), taxonomic investigation of the literature, examining the different FM types, NN architectures, learning paradigms, learning stages, robotic tasks, and application domains, e) It incorporates a per criterion methodical comparative analysis of the different approaches and facilitates the reporting of critical insights, and f) It provides a comprehensive and hierarchical discussion on the current challenges and future research directions in the field. At the same time though, the current survey does not include any empirical/quantitative benchmarking of the surveyed methods and it inevitably constitutes a snapshot of a rapidly evolving field. Moreover, a more detailed version of the above comparison is provided in Section A of the supplementary document, which reports, for each surveyed work, its survey scope, review methodology, primary contributions, and main limitations.

The remainder of the manuscript is organized as follows: Section 2 outlines the adopted methodology for reviewing the relevant literature. Section 3 describes the evolution in robotic FM research and indicates the most widely adopted models. Section 4 delineates the criteria used for analyzing the literature, as well as the resulting categories of robotic FM methods structured in the form of a taxonomy. Sections 5-10 discuss in detail the various categories of robotic FM approaches, taking into account the type of the employed FM, the nature of the underlying NN architecture, the adopted learning paradigm, the learning stage at which knowledge is integrated to the FM, the performed robotic task, and the selected application domain, respectively. Section 11 reviews the publicly available datasets for training and evaluating robotic FM methods, organized into the main recurring dataset families, along with their typical uses and current gaps. Sections 12 and 13 discuss the current challenges and future research directions in the field, correspondingly, while Section 14 concludes the paper.

## 2 Literature review methodology

**Overview**: In order to efficiently and thoroughly identify/map the robotic FM literature, while at the same time detecting key concepts and trends, a structured and systematic review methodology was adopted, ensuring comprehensiveness and relevance of the selected research works. The main aspects of this methodology are summarized below.

**Scope and objectives**: In terms of scope and objectives, the survey targets approaches for various robotic tasks (namely, perception, planning, navigation, manipulation, and human-robot interaction) whose execution relies on the use of an underlying FM, emphasizing the following objectives: a) The main categories of methods based on multiple criteria, b) The utilized datasets, c) Current challenges, and d) Future research directions.

**Literature search**: The literature search involved querying several major scientific databases, namely IEEE Xplore, Google Scholar, Scopus, DBLP, arXiv, and Web of Science, by combining targeted keywords/terms (e.g., "foundation model", "robotics", "vision-language-action", "large language model", etc.) with Boolean operators. To guarantee contemporary relevance, the search primarily focused on research works published within the last five years, while certain earlier seminal studies were also included. It needs to be clarified that the actual search process was applied iteratively with successive keyword refinement, and was further supplemented by an extensive backward reference-checking/chaining procedure. This multi-pronged design renders the overall literature review process relatively robust against missing critical/important works in the field, including ones whose original terminology predates the current FM established one. Moreover, this keyword plus reference search strategy was deliberately favored over a pure embedding-based semantic retrieval over paper embeddings, since it offers improved reproducibility, transparency, and precision/recall control; embedding-based semantic search can, nevertheless, serve as a useful complementary tool.

**Screening and selection**: A multi-stage screening process was then applied, excluding duplicate records, non-English papers, and articles without full-text access, followed by title/abstract filtering and in-depth full-text review. The latter retained only studies where a FM comprises a key algorithmic component and which exhibit substantial theoretical and/or experimental contributions, with priority given to prominent robotics and AI/ML publication venues. Eventually, a total of 438 articles were selected for analysis and were included as references in the current manuscript.

**LLM assistance**: Regarding the use of automated tools during the preparation process of the manuscript, LLM assistance was employed only on specific occasions and solely as a complementary aid for cross-checking potential gaps in the literature search described above (i.e., as a supplementary means of identifying relevant works that might have been missed by the database queries). However, all substantive scholarly work was carried out manually by the authors; in particular, the taxonomy design, the paper inclusion/exclusion decisions, all comparative methodical analyses, the study of individual works, and the inclusion and verification of all references were carefully performed and validated by the authors. The authors take full responsibility for the entire content of the manuscript.

**Additional details**: The full details of the review methodology, including the exact database queries and key bibliometric analytics (article types and most popular publication venues) of the selected literature, are provided in Section B of the supplementary document, whereas in-depth analysis of the identified robotic FM works is provided in Sections 3-11.

## 3 Robotic FM research evolution

### 3.1 Research phases

Although foundation models have only relatively recently been introduced in the field of robotics, they have decisively contributed towards transformative effects, while gradual advancements and emerging research trends can be identified in the literature. In particular, the evolution of robotic FM research can be roughly classified into subsequent and distinct phases, each corresponding to a critical paradigm shift regarding how perception, reasoning, and control procedures are consolidated in a robotic system. The overall research

progress concentrates on repositioning from isolated/modular designs to integrated/general-purpose agents (Reed et al., 2022; Driess et al., 2023). The considered research phases are graphically illustrated in Fig. 1, along with key/milestone works associated with each of them, while they are explained in detail in the followings.

The delineation of the above phases follows a systematic rationale, where boundaries are combinatorially defined on the basis of the following parallel axes: a) The dominant source of training data (successively comprising Web text/image corpora, aligned image-text pairs, large-scale real robot trajectories, mixed Web and robot experiences, and multi-sensory simulation and real-world data), b) The level of system integration (ranging from modular pipelines relying on hand-engineered controllers to end-to-end, general-purpose policies), and c) The model output space (progressively encompassing perception/language grounding, task plans, and executable action sequences). Additionally, a given work is designated as a key/milestone one, in case that it first demonstrates a capability characteristic of a given phase, it is widely adopted as a reference point in the community, and it marks the transition to a subsequent phase. Moreover, it needs to be mentioned that the reported year ranges are approximate and partially overlapping, reflecting the gradual shifts in research and are anchored to the appearance of representative works (rather than to strict cut-offs).

Regarding alignment with the existing literature, it should be noted that current surveys (Table 1) predominantly organize the field around capability- or model-centric taxonomies, rather than explicitly dated timelines. For the cases that prior works do address temporal progression, most notably the VLA-oriented reviews of Kawaharazuka et al. (2025); Ma et al. (2024c); Sapkota et al. (2025), the described trajectory (from early vision-language grounding, to end-to-end VLA policies, and then to real-world generalist systems) is consistent with Phases 2-5 of the current work. However, the explicit five-phase research decomposition introduced in the current study renders this implicit chronology concrete and extends it to more recent (2024-2026) developments in multi-sensory generalization and real-world deployment, which largely fall outside the coverage period of earlier surveys.

### 3.1.1 Phase 1: Integration of native Natural Language Processing (NLP) and Computer Vision (CV) models (2018-2021)

In the early attempts of incorporating FMs in robotic systems, native large-scale off-the-shelf NLP and CV models are used, aiming at enhancing robotic platforms with improved perception capabilities (for example, identifying and tracking objects in camera video sequences, translating human verbal requests to robot symbolic goals, etc.) (Hong et al., 2021). The data required for training such FMs largely originate from standard Web text and image repositories (information sources that contain no robot action policies, no time-varying sensor streams, and no interactions with the physical world). These networks are integrated following a modular approach, requiring a separate conventional controller for realizing motion and action planning. Due to this fact, NLP and CV FMs serve a supportive role (boosting perception and language grounding tasks), while the robot's behavior remains determined by a hand-engineered downstream pipeline (Shridhar et al., 2022; Hong et al., 2021).

### 3.1.2 Phase 2: Grounded planning with Vision-Language (VL) representations (2021-2022)

With the emergence of vision-language and scalable language models, increased capabilities for integrating richer semantic priors and more flexible instruction grounding to robots are introduced. In particular, multimodal embeddings are used by robotic systems for linking visual input with natural language commands and generating (or evaluating) task sequences (Sun et al., 2022; Brohan et al., 2023b). Such developments take advantage of broader access to aligned image-text data and pretrained language models (Radford et al., 2021), although robotic data remains relatively limited. The latter constrains robots to utilize a relatively decreased number of own recorded/captured experiences; hence, most training samples originate from third-party datasets (e.g., curated demonstration benchmarks or Web-scraped image-text pairs), forcing models to generalize from proxy sources, rather than direct embodied trials (Shah et al., 2023a).

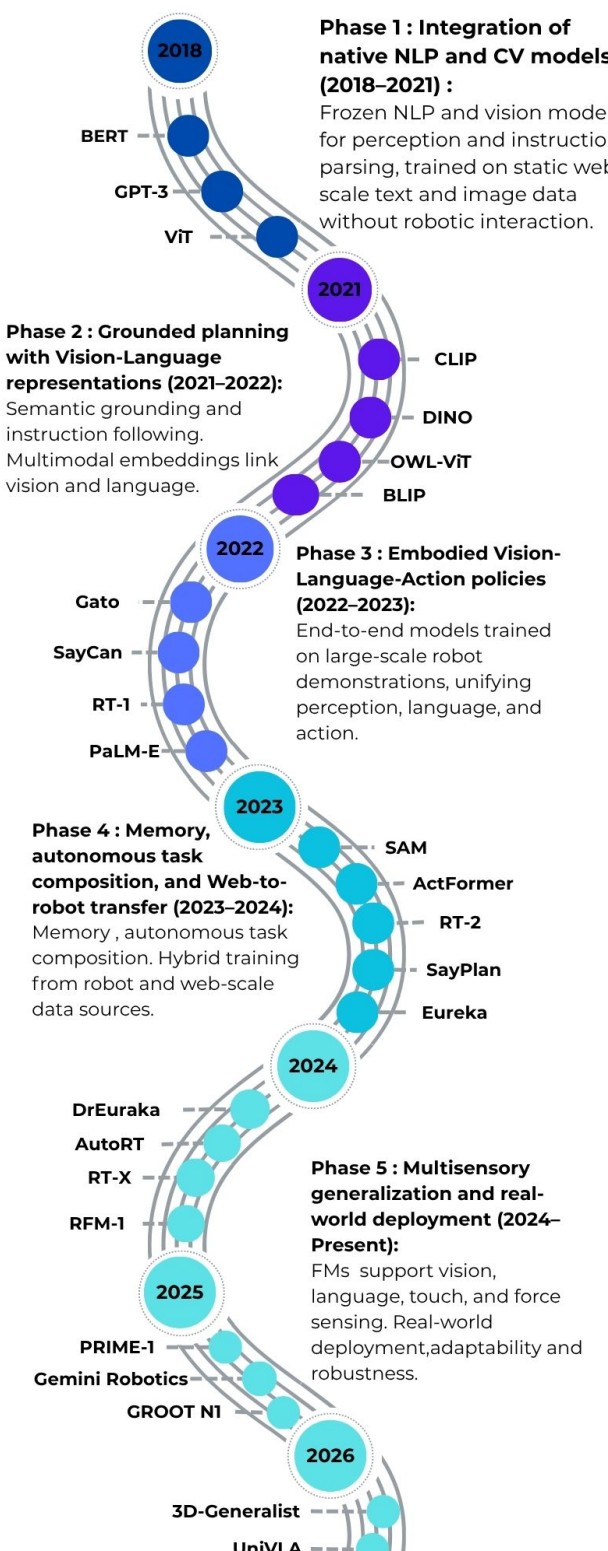

Figure 1: Main phases in robotic FM research and key/milestone works.

### 3.1.3 Phase 3: Embodied Vision-Language-Action (VLA) policies (2022-2023)

This phase marks the introduction of comprehensive and unified robot policies, derived directly from training using large-scale robot-demonstration datasets. In particular, FMs process vision, language, and task context simultaneously, while outputting action sequences using the same architecture (Brohan et al., 2023a; Reed et al., 2022; Driess et al., 2023). This shift towards end-to-end training is enabled by the availability of real-world robotic data at scale (O'Neill et al., 2024), collected considering hundreds of tasks and corresponding variations (e.g., different object types, lighting conditions, camera viewpoints, robot embodiments, etc.). As a consequence, robotic platforms are enhanced by incorporating the capability to generalize across goals and environments without explicit task engineering (Shridhar et al., 2023; Zitkovich et al., 2023).

### 3.1.4 Phase 4: Memory, autonomous task composition, and Web-to-robot transfer (2023-2024)

Building on prior advances, this phase concerns systems capable of long-horizon planning (Ajay et al., 2023), world-state tracking (Wu et al., 2023b), and autonomous skill discovery (Nam et al., 2023). In this respect, FMs are trained or adapted using both structured robot trajectories and unstructured Web-scale corpora (Kim et al., 2025). Additionally, data diversity increases, combining multimodal internet data with robot-collected experiences (Mees et al., 2024). Moreover, robotic agents synthesize and evaluate their own tasks based on FM reasoning pipelines, boosting semantic autonomy and self-improvement (Parakh et al., 2024).

### 3.1.5 Phase 5: Multi-sensory generalization and real-world deployment (2024-Present)

More recently, research advancements have focused on building robust generalist robotic systems, capable of efficiently operating in unstructured real-world environments (Bjorck et al., 2025; Team et al., 2025). The utilized FM-based solutions support multimodal inputs, including vision, language, touch, force, and proprioception (Li et al., 2026b), as well as real-time adaptation (Routray et al., 2026). Additionally, robotic agents are trained and refined using a combination of simulation and real-world interaction data, significantly extending the boundaries in robustness, transferability, safety, and autonomy (Zhao et al., 2026a; Guo et al., 2026). Consequently, FM-based solutions are being widely adopted across industrial, assistive, and mobile platforms (Sohn et al., 2024).

## 3.2 Key foundation models

Throughout the different research phases described in Section 3.1, key FM architectures have been introduced, which, on the one hand, have led to significant technological advancements and capabilities, and, on the other hand, have served as the basis for numerous methods in the field. In particular, the most common and widely adopted robotic FMs, along with their main characteristics, are briefly summarized in Table 2. For each model, the latter reports its year of publication, category, type, input and output modalities, adoption of embodied design ('Emb.'), number of parameters ('Param.'), public availability of an implementation ('Public'), and key innovation. Regarding the category criterion, a distinction is made between standalone robotic foundation models (denoted as 'FM') and FM-based robotic systems (denoted as 'System'); the latter comprise modular/compositional approaches that orchestrate one or more existing FMs, rather than constituting standalone models themselves. Regarding the embodiment criterion, a model is considered embodied ('Emb.' = Y) only when it is trained on, or directly produces, robot-executable actions/observations of a physical or simulated agent; hence, general-purpose models that are widely adopted in robotics but operate on generic vision/language data are not characterized as embodied themselves.

## 4 Key criteria and main categories of robotic FM methods

This section provides a systematic overview of the landscape of robotic FM methods. For facilitating the analysis, a set of complementary and diverse criteria are defined (each focusing on a specific/key aspect of a robotic FM system), resulting into the classification of the literature works into a corresponding set of main categories. The different criteria used and the resulting categories are graphically illustrated in Fig. 2 and detailed as follows:

Table 2: Most common and widely adopted FMs in robotics.

| Model | Year | Category | Type | Input | Output | Emb. | Param. | Public | Key innovation |
|---|---|---|---|---|---|---|---|---|---|
| BERT (Devlin et al., 2019) | 2019 | FM | LLM | Text | Text embedding | N | 110M | Y | Natural-language command interpretation and translation to action sequences |
| GPT-3 (Brown et al., 2020) | 2020 | FM | LLM | Text | Text | N | 175B | N | Zero/few-shot reasoning for instruction-to-plan/code translation |
| ViT (Yuan et al., 2021) | 2021 | FM | VFM | Image | Class logit | N | 86M | Y | Visual perception capturing long-range dependencies and global context |
| CLIP (Radford et al., 2021) | 2021 | FM | VLM | Text, image | Embedding | N | 428M | Y | Zero-shot object grounding and task specification from free-form text |
| DINO (Caron et al., 2021) | 2021 | FM | VFM | Image | Embedding | N | 86M | Y | Self-supervised object attention maps aiding manipulation |
| OWL-ViT (Minderer et al., 2022) | 2022 | FM | VLM | Text, image | Box, label | N | 100M | Y | Zero-shot open-vocabulary object detection from text |
| BLIP (Li et al., 2022) | 2022 | FM | VLM | Text, image | Text, embedding | N | 480M | Y | Zero-shot vision-language understanding and generation |
| Gato (Reed et al., 2022) | 2022 | FM | VLA | Text, image, state | Action | Y | 1.18B | N | Generalist multi-task, multi-embodiment policy (600+ tasks) |
| SayCan (Brohan et al., 2023b) | 2023 | System | LLM | Text, environment | Action | Y | – | N | Affordance-grounded, feasible robotic action planning |
| RT-1 (Brohan et al., 2023a) | 2023 | FM | VLA | Text, image | Action | Y | – | Y | End-to-end mobile-manipulation policy from 130K+ real trajectories |
| PaLM-E (Driess et al., 2023) | 2023 | FM | VLA | Text, image | Text, action | Y | 562B | N | Embodied multimodal reasoning and long-horizon plan generation |
| SAM (Kirillov et al., 2023) | 2023 | FM | VFM | Image, prompt | Mask | N | 636M | Y | Promptable zero-shot image segmentation |
| ActFormer (Xu et al., 2023) | 2023 | FM | VFM | Action class | 3D motion | N | – | N | Action-conditioned 3D human motion generation (GAN transformer) |
| RT-2 (Zitkovich et al., 2023) | 2023 | FM | VLA | Text, image | Action | Y | – | N | Web-scale and robot co-training for generalization to unseen tasks/objects |
| SayPlan (Rana et al., 2023) | 2023 | System | LLM | Text, scene graph | Plan | Y | – | N | Scalable long-horizon planning over 3D scene graphs |
| Eureka (Ma et al., 2024a) | 2024 | System | LLM | Task description | Reward | N | – | Y | Iterative LLM generation of robot reward-function code |
| DrEureka (Ma et al., 2024b) | 2024 | System | LLM | Simulation, task configuration | Reward | N | – | Y | Prompt-driven automation of the sim-to-real training pipeline |
| AutoRT (Ahn et al., 2024) | 2024 | System | VLA | Text, image | Action | Y | – | N | Safety-constrained orchestration of robot action generation |
| RT-X (O'Neill et al., 2024) | 2024 | FM | VLA | Text, image | Action | Y | – | Y | Cross-embodiment policy across 22 platforms (527 skills) |
| RFM-1 (Sohn et al., 2024) | 2024 | FM | VLA | Text, image, video | Action | Y | 8B | N | Multimodal physics-informed reasoning for complex tasks |
| PRIME-1 (Inc., 2025) | 2025 | FM | VLA | Image | 3D features, action | Y | – | N | Real-world adaptive control for multi-task operational settings |
| Gemini robotics (Team et al., 2025) | 2025 | FM | VLA | Text, image | Action | Y | – | N | On-device multimodal reasoning for dexterous bi-manual tasks |
| GR00T N1 (Bjorck et al., 2025) | 2025 | FM | VLA | Text, image | Action | Y | 2B | Y | Single, multi-task, general-purpose architecture for humanoid robots |
| 3D-GENERALIST (Sun et al., 2026) | 2026 | FM | VLM | Text, image | Action code, 3D scene | N | – | N | VLM-as-policy generation of simulation-ready 3D environments |
| UniVLA (Wang et al., 2026b) | 2026 | FM | VLA | Text, image, video | Action, image | Y | 8.5B | Y | Unified token-space vision-language-action modeling with video world-model post-training |

- **Foundation model type**: FMs in robotics can be grouped taking into account the number and the nature of the input-output modalities involved, which critically dictates their overall capabilities and how they manage robot perception, reasoning, and action procedures. The main types of FMs are:

  - **Large Language Models (LLMs)**: These receive textual streams of data as input (sometimes also multimodal information) and, subsequently, generate high-level action policies. Their fundamental functionality relies on translating natural language instructions into task goals, programs, or supervision signals (Brohan et al., 2023b; Liang et al., 2023; Driess et al., 2023).

  - **Vision Foundation Models (VFMs)**: These receive as input visual information streams (e.g., RGB, LiDAR, thermal, etc.) and output corresponding delicate representations (e.g., object

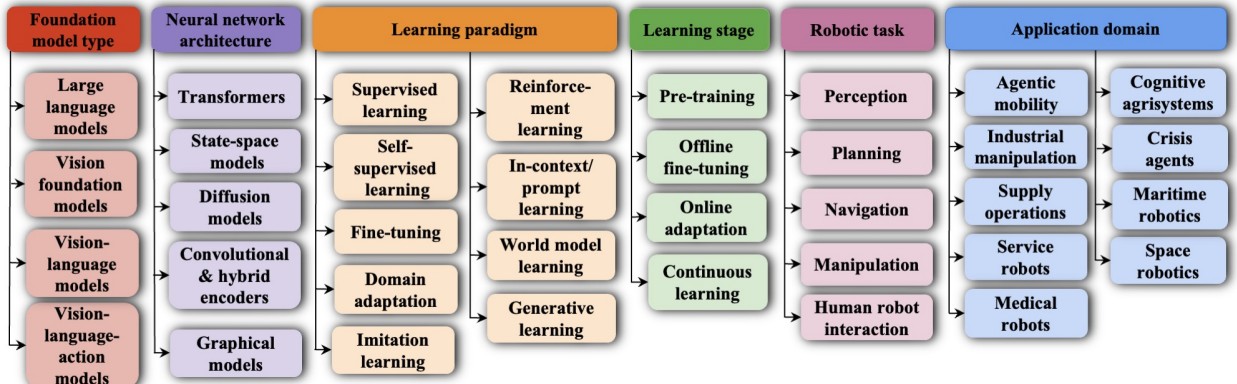

Figure 2: Key criteria and main resulting categories of robotic FM methods.

detection masks, depth maps, dense embeddings, etc.) for facilitating subsequent robotic tasks (Kirillov et al., 2023; Zhang et al., 2023).

– **Vision-Language Models (VLMs)**: These exploit correlations and inter-dependencies among the input visual and textual data, in order to enable complex robot operations (e.g., visual grounding, language-conditioned mapping, question answering, etc.) (Radford et al., 2021; Liu et al., 2024c).

– **Vision-Language-Action models (VLAs)**: These comprise native robotic models that map multi-modal inputs directly/end-to-end to generalist action policies, across multiple types of tasks and embodiments (Zitkovich et al., 2023; Kim et al., 2025; Bjorck et al., 2025).

- **Neural Network (NN) architecture**: The type of the underlying neural network architecture that is employed in any FM solution largely defines its capabilities, efficiency, and limitations in robotic applications. The main NN types used in robotic FM methods are:

– **Transformers**: These rely on self-attention to capture long-range dependencies and to model complex multimodal inputs in a unified way (Vaswani et al., 2017; Driess et al., 2023; Zitkovich et al., 2023).

– **State-Space Models (SSMs)**: Their fundamental functionality is grounded on the use of a set of first-order differential or difference equations for modeling complex, dynamic operations, typically involving multiple input and output signals (Gu & Dao, 2024; Liu et al., 2024b; Lenz et al., 2025).

– **Diffusion models**: These employ an iterative denoising process, in order to generate diverse and realistic data samples (e.g., action policies) (Ho et al., 2020; Chi et al., 2025).

– **Convolutional and hybrid encoders**: Convolutional Neural Network (CNN) encoders are efficient in learning hierarchical feature representations and modeling local patterns in the input data, while they are often combined with transformer or diffusion networks for further enhancing visual perception (Nair et al., 2023; Brohan et al., 2023a; Chi et al., 2025).

– **Graphical models**: These allow the processing of data that exhibit irregular and/or complex relations, while also enabling the generation of context-aware representations (Rana et al., 2023; Gu et al., 2024; Patel & Song, 2025).

- **Learning paradigm**: In order to develop a robust robotic FM solution, different/diverse learning techniques, principles, and approaches can be adopted. The most commonly met learning paradigms in the literature, which are typically combined in a comprehensive learning methodology, are the following:

– **Supervised Learning (SL)**: This relies on training a FM on large-scale, labeled datasets of input-output pairs (e.g., observations/instructions mapped to target outputs or actions), so that

the model directly learns the desired input-to-output mapping from explicit, human-provided supervisory signals (Zitkovich et al., 2023; Driess et al., 2023; Team et al., 2025).

– **Self-Supervised Learning (SSL)**: This enables ML models to generate their own supervisory signals directly from raw, unlabeled data, thereby circumventing the need for costly external human-provided labels (Gao et al., 2025b; Nazeri et al., 2025).

– **Fine-tuning**: This allows generalist FMs to efficiently adapt to specific tasks, environments, and robotic platforms, while demonstrating significant efficiency and cost-effectiveness (Mees et al., 2024; Yadav et al., 2026).

– **Domain Adaptation (DA)**: This aims at adjusting a FM, originally trained on a source domain, to maintain its accuracy and performance when applied to a new target domain (Li et al., 2026a; Zheng et al., 2026).

– **Imitation Learning (IL)**: Also known as Learning from Demonstration (LfD) or Robot programming by Demonstration (PbD), it relies on the consideration of an autonomous agent learning to execute tasks or acquiring new skills, by observing/emulating demonstrations provided by an expert (Wan et al., 2024; Fu et al., 2025; Cai et al., 2024b).

– **Reinforcement Learning (RL)**: This is grounded on the use of an agent learning to make sequential decisions and to adjust its behavior through trial-and-error interactions with its surrounding environment, taking into account feedback received in the form of rewards or penalties for its actions (Ma et al., 2024a; Tziafas & Kasaei, 2024; Wang et al., 2024c).

– **In-context/prompt learning**: This paradigm enables inference-time adaptation of FMs, by guiding model behavior through demonstrations, examples, or task-specific instructions. As such, it supports flexible task adaptation and behavior specification, allowing pre-trained models to generalize to new scenarios through contextual conditioning alone (Huang et al., 2023b; Grigorev et al., 2025; Yin et al., 2025c).

– **World Model (WM) learning**: WMs enable the grounding of FM knowledge into physically, real-world, plausible predictions, by modeling environmental dynamics and predicting the consequences of robot actions (Gao et al., 2025b; Zhou et al., 2025b).

– **Generative Learning (GL)**: This facilitates towards reducing the reliance on extensive real-world datasets during training, by artificially synthesizing diverse and high-quality robot experiences (Zhao et al., 2026a; Heppert et al., 2026).

- **Learning stage**: The particular phase, during the overall learning process, at which knowledge is incorporated to a FM, largely defines the type/nature of the acquired knowledge, algorithmic/development details, and key assumptions about the model behavior. In this respect, the main learning stages identified in the literature are summarized as follows:

– **Pre-training**: This is the first and by-far the most computationally intensive step in the FM development life-cycle, which involves the processing of massive, internet-scale, diverse, and usually multi-modal datasets for learning general-purpose feature representations (Zitkovich et al., 2023; Driess et al., 2023).

– **Offline fine-tuning**: Following generic pre-training, this step focuses on adjusting the FM knowledge structures to the requirements/nuances of particular application domains or downstream tasks, making use of (minimal) additional training data (Mees et al., 2024; Li et al., 2026a).

– **Online adaptation**: This corresponds to real-time adjustments of a robot's behavior for maintaining performance during deployment, involving the acquisition of new skills, response to novel tasks, or handling of unforeseen environmental conditions (Wang et al., 2024c; Grigorev et al., 2025).

– **Continuous learning**: This aims at enabling FMs to continuously acquire new skills, to refine existing ones, and to maintain performance in dynamic, real-world environments in the long term (Wan et al., 2024; Murillo-González & Liu, 2025).

- **Robotic task**: The introduction of FMs has induced transformative effects in the materialization and execution of all core robotic tasks, i.e., specific jobs, actions, or functions that robots perform in

order to achieve a goal. The most common, pronounced robotic tasks, where FMs are applied to, are as follows:

- **Perception**: FMs equip robots with enhanced capabilities to perceive and reason about their surrounding environment, largely relying on visual information processing streams and often combined with additional modalities (e.g., natural language inputs) (Radford et al., 2021; Jiang et al., 2024; Yamazaki et al., 2024; Nguyen et al., 2024).
- **Planning**: FMs enable robots to interpret complex, high-level, human-like commands (e.g., in natural language form) and to subsequently translate them into (long-horizon) sequences of low-level, executable, and discrete actions (Brohan et al., 2023b; Liang et al., 2023; Chen et al., 2024b; Singh et al., 2023).
- **Navigation**: FMs significantly boost robot navigation capabilities, by moving away from traditional, task-specific models and heading towards more generalized, adaptable schemes for efficient operation in complex, unstructured, and dynamic environments (Shah et al., 2023c; Wang et al., 2024a; Huang et al., 2023a; Xu et al., 2024b).
- **Manipulation**: Often equally termed motor control, it is enhanced by the use of FMs by shifting away from task-specific programming to more generalized, adaptable approaches, also supporting more dexterous and precise manipulation tasks (Brohan et al., 2023a; Zitkovich et al., 2023; Driess et al., 2023; Bjorck et al., 2025).
- **Human-Robot Interaction (HRI)**: FMs enable robotic platforms to understand and interact/respond with/to humans in a more intuitive, natural, flexible, and human-like way (Izquierdo-Badiola et al., 2024; Liu et al., 2024d; Bärmann et al., 2024; Irfan et al., 2024).

- **Application domain**: FMs have significantly enhanced several aspects of robot capabilities (e.g., autonomy, complex decision-making, human-robot interaction, etc.) in challenging real-world settings; hence, further boosting their widespread use, including the following main/common application domains:

  - **Agentic mobility**: FMs significantly extend the capabilities of the conventional autonomous driving stack (i.e., perception, prediction, planning, and control), by transforming it into a single, cohesive, end-to-end decision-making framework (Wu et al., 2024; Xu et al., 2024b; Wang et al., 2024b).
  - **Industrial manipulation**: FMs revolutionize industrial automation pipelines, by converting rigid, task-specific solutions into flexible, general-purpose agents that are capable of handling dynamic, unstructured manufacturing tasks (Sohn et al., 2024; Inc., 2025; Kim et al., 2025; Zitkovich et al., 2023).
  - **Supply operations**: FMs dramatically increase flexibility, intelligence, and generalization, enabling robots to move beyond repetitive, structured tasks to handle unstructured, dynamic, and complex operations (Nicoletti & Appolloni, 2024; Xu et al., 2024a; Nicoletti, 2025).
  - **Service robots**: Robots are more efficient in operating safely and intelligently in challenging household environments, while greatly capitalizing on their ability of receiving instructions in natural language (Wu et al., 2023a; Mon-Williams et al., 2025).
  - **Medical robots**: Robotic platforms incorporate comprehensive, fine-grained medical knowledge, enabling them to robustly undertake high-stake, high-variability tasks, to provide context-aware intelligence, and to support consistent, precise assistance (Cui et al., 2024; Zeinoddin et al., 2024; He et al., 2024).
  - **Cognitive agrisystems**: Robotic platform capabilities evolve from conventional, field-level task execution to efficient, resilient, and sustainable precision farming, i.e., moving beyond simple automation to genuine, autonomous intelligence (Yin et al., 2025b).
  - **Crisis agents**: FMs enable robot operations to elaborate from conventional, remote-controlled settings to the handling of autonomous reasoning and adaptation circumstances in unpredictable, highly dangerous environments (Driess et al., 2023; Kim et al., 2025; Zitkovich et al., 2023).
  - **Maritime robotics**: Robots are reinforced with advanced capabilities so as to efficiently overcome typical, extreme environmental challenges in under- and open-water settings, in

principle relying on robust, generalized, and multi-sensorial intelligence/reasoning pipelines (Zheng et al., 2024b).

– **Space robotics**: FMs enable the functioning of robots under extreme operating conditions, involving limited resource availability and highly variable, unknown environments, largely relying on their enhanced autonomous decision-making capabilities (Giannakis et al., 2024; Zhao & Ye, 2024).

# 5 Foundation model types

FMs used in robotics can be organized into groups with respect to the number and the nature of the input-output modalities involved. The latter also largely affects their exhibited capabilities. In particular, the main types of FMs are: a) Large Language Models (LLMs), b) Vision Foundation Models (VFMs), c) Vision-Language Models (VLMs), and d) Vision-Language-Action models (VLAs), as discussed in Section 4 and further detailed below.

## 5.1 Large Language Models (LLMs)

In the context of robotics, LLMs are primarily used as high-level, cognitive task planners and reasoning engines that generate the sequence of operations that are necessary for accomplishing a stated goal (Brohan et al., 2023b; Liang et al., 2023; Chen et al., 2024b). In practice, they translate high-level natural language inputs to low-level robot behaviors, turning free-form instructions and short state summaries into typed goals, multi-step plans, executable code, constraints, and run-time feedback, which renders them particularly useful for tasks requiring sophisticated reasoning and complex decision-making (Li et al., 2025a).

In terms of supported functionality/operation, LLM-based systems can be classified into the following main categories:

- Goal/constraint grounding and context: LLMs map abstract human instructions (goal and context) to low-level robot actions (grounding). SayCan (Brohan et al., 2023b) filters skills via PaLM-based value affordances to stay within the robot's capabilities, while LM-Nav (Shah et al., 2023a) converts instructions into visually grounded way-points for a navigation planner.

- Command interpretation and code synthesis: LLMs act as natural language to code translators, letting robots accept high-level instructions instead of hand-written code. Code-as-Policies (Liang et al., 2023) compiles instructions into inspectable, reusable robot API code, while AutoTAMP (Chen et al., 2024b) translates requests into TAMP specs checked by a symbolic planner for feasibility.

- Task planning and long-horizon reasoning: LLMs decompose abstract goals into sequential, grounded actions with contextual awareness across steps. SELP (Wu et al., 2025b) maps instructions to temporal logic via constrained decoding so that plans meet safety and efficiency constraints, while LLM-GROP (Zhang et al., 2025e) cross-checks instructions against motion feasibility in cluttered settings.

- Perception-aware and multimodal integration: LLM reasoning is enhanced by accounting for the robot's visual and physical surroundings. PaLM-E (Driess et al., 2023) injects visual and proprioceptive streams into inference so that decisions match real-world observations, while Chain-of-Modality (Wang et al., 2025a) derives a plan and control parameters from human videos and auxiliary signals.

- Navigation and spatial understanding: LLMs turn abstract navigation commands into grounded paths via structured textual scene representations. LM-Nav (Shah et al., 2023a) links instructions to landmarks and routes through CLIP-based grounding, while SayPlan (Rana et al., 2023) plans over a 3D scene graph and re-plans on infeasible steps for large-scale missions.

- Conversational interfaces and teleoperation: LLMs create conversational interfaces and shared-control teleoperation, making robots accessible to non-experts. TidyBot (Wu et al., 2023a) learns user-specific tidying conventions and transfers them to new homes, while LAMS (Tao et al., 2025) predicts intent and auto-switches teleoperation modes to lower cognitive load.

- Execution-time validation, error handling, and recovery: LLMs recast run-time failures as semantic problems for dynamic recovery, instead of brittle pre-defined routines. STATLER (Yoneda et al., 2024) interprets robot state and tool feedback to suggest targeted repairs without restarting, while CAPE (Raman et al., 2024) re-prompts and proposes fixes for precondition failures.

- Adaptation, efficiency, and safety: LLMs facilitate robots to adapt and to operate safely under novel situations and disruptions. Eureka (Ma et al., 2024a) writes and refines GPT-4 reward code to speed skill acquisition across platforms, while DrEureka (Ma et al., 2024b) co-designs rewards and domain randomization for efficient sim-to-real transfer.

- Knowledge retrieval and memory: LLMs access and store external knowledge and past episodes to overcome limitations of immediate sensor data. ELLMER (Mon-Williams et al., 2025) couples GPT-4 with retrieval-augmented memory so that a mobile manipulator incorporates context, adapts plans on the fly, and completes multi-step household tasks as conditions change.

## 5.2 Vision Foundation Models (VFMs)

The ultimate goal of VFMs is to address the perceptual requirements of embodied AI systems, by providing generalized, high-quality visual representations necessary for interaction with the physical world (Kirillov et al., 2023; Oquab et al., 2024). In the robotics setting, VFMs distill raw pixel information into rich, transferable visual features or embeddings, enabling a robust and generalized visual understanding that serves as the input information stream for modulating downstream control policies (Shang et al., 2024).

In terms of supported functionality/operation, VFM-based systems can be classified into the following main categories:

- Object recognition: VFMs enable generalized visual recognition via transferable representations that boost task-specific training. SAM (Kirillov et al., 2023) yields class-agnostic, promptable segmentation masks for isolating objects, while DINOv2 (Oquab et al., 2024) provides robust dense features that transfer across scenes under domain shift.

- Localization: VFMs improve localization with robust, semantic, globally consistent representations beyond geometric methods. DINO-VO (Azhari & Shim, 2025) uses DINOv2 features and ViT-based keypoints for robust monocular visual odometry at high throughput, while LiteVLoc (Jiao et al., 2025) enables long-range re-localization for image-goal navigation.

- Object tracking: VFMs support trackers rich semantic understanding and long-term memory, improving robustness to occlusion and viewpoint change. Zhong et al. (2024) extract text-prompted segmentation masks to train a recurrent policy via offline RL, while open-vocabulary cues enable instance tracking of novel objects over time (Guo et al., 2025).

- Depth perception: VFMs enable robust, high-fidelity depth estimation, where sensors or traditional methods under-perform. Metric3D v2 (Hu et al., 2024) uses geometric priors for zero-shot metric depth estimation across diverse cameras, while Prompt-Depth-Anything (Lin et al., 2025a) demonstrates that a small LiDAR 'metric prompt' can steer a FM to accurate, high-resolution depth estimation.

- Semantic map creation: VFMs supply robust, semantic-aware features that improve map accuracy and informativeness. Busch et al. (2025) build reusable open-vocabulary feature maps with probabilistic-semantic updating, while combining VFM features with Gaussian-splatting supports robust long-horizon missions in dynamic environments (Zheng et al., 2025a).

- Visual-inertial fusion: VFMs enhance the visual part of visual-inertial odometry (VIO) systems, which is crucial for drift correction and estimation of metric scale. Specifically, features that improve VO (Azhari & Shim, 2025) or metric priors from depth FMs (Hu et al., 2024) are combined with Inertial Measurement Unit (IMU) data in standard VIO estimators.

- Environment mapping: VFMs construct maps by producing dense visual embeddings fused into persistent, semantically enriched scenes. FMGS (Zuo et al., 2025) combines FM features with 3D Gaussian splatting for semantic reconstruction and open-vocabulary understanding, while OpenGS-SLAM (Yang et al., 2025) adds FM-derived semantic features for robust real-time tracking and mapping.

## 5.3 Vision-Language Models (VLMs)

VLMs combine computer vision with natural language processing capabilities for establishing a coherent, concrete semantic understanding of the world, enabling interpretation and generation of language descriptions of the observed visual entities (Radford et al., 2021). In the context of robotics, VLMs enable robots to simultaneously interpret visual data and natural language commands, allowing for intuitive human-robot interaction and robust task execution in unstructured environments (Zhou et al., 2025c).

With respect to supported functionalities/operations, VLM-based systems can be classified into the following main categories:

- Manipulation grounding and control signals: VLMs map high-level semantic intents into concrete, actionable constraints near the robot. OmniManip (Pan et al., 2025c) converts VLM reasoning into object-centric primitives with dual closed-loop planning and execution for precise 3D constraints generation, while RoboGround (Huang et al., 2025a) feeds grounded target and placement masks into a low-level policy.

- Semantic mapping, referring expressions, and navigation: VLMs build human-readable maps, interpret spatial language, and ground navigation goals. One-Map-to-Find-Them-All (Busch et al., 2025) forms a reusable open-vocabulary map for zero-shot multi-object-based navigation, while VLFly (Zhang et al., 2025h) performs grounded vision-language UAV navigation without active ranging sensors.

- Execution-time check and progress verification: VLMs enable closed-loop semantic self-monitoring, turning sensor feedback into human-understandable checks. ExploreVLM (Lou et al., 2025) integrates perception, planning, and execution validation in real time, while Ahmad et al. (2025) verify skill pre- and post-conditions and suggest recovery skills for failure handling.

- Closed-loop mobile manipulation: VLMs supply continuous feedback and adaptation for long-horizon execution in unstructured, dynamic environments. COME-robot (Zhi et al., 2025) uses GPT-4 for situated reasoning and iterative feedback to recover from failures, while HomeRobot (Yenamandra et al., 2023) navigates homes to grasp novel objects and to place them on receptacles.

## 5.4 Vision-Language-Action models (VLAs)

VLAs aim at integrating multi-modal understanding with direct physical execution, targeting to serve as the basis for autonomous embodied task execution (Ma et al., 2024c). In particular, a VLA model receives multi-modal inputs (typically, vision, language, and robot state) and generates real-world physical actions or control policies in real-time, often designed in an end-to-end fashion (Sapkota et al., 2025).

In terms of supported functionality/operation, VLA-based systems can be classified into the following main categories:

- Scaling and web-to-robot transfer: VLAs transfer internet-scale semantic and visual knowledge to control policies, boosting cross-task and cross-embodiment generalization. RT-1 (Brohan et al., 2023a) scales imitation learning with language-tied tokenized actions for long-tail tasks, while RT-2 (Zitkovich et al., 2023) adds web-scale vision-language pretraining for open-vocabulary transfer to real robots.

- Fusion and action parameterization: VLAs unify perception, reasoning, and control by pairing a VLM with an action decoder. GR00T N1 (Bjorck et al., 2025) couples a VLM with a diffusion

Table 3: Foundation model types: Comparative analysis and key insights.

| Aspect | LLMs | VFMs | VLMs | VLAs |
|---|---|---|---|---|
| Primary functions | • Cognitive task planning 
 • Symbolic reasoning 
 • Language-to-action translation 
 • Task decomposition | • Visual representations 
 • Dense features/embeddings 
 • Object/instance differentiation 
 • Geometric reconstruction | • Visual language grounding 
 • Open-vocabulary recognition 
 • Visual reasoning 
 • Semantic alignment | • Policy execution 
 • Hardware actuation 
 • Embodied task execution 
 • Multi-modal physical alignment |
| Input | • Text tokens 
 • NL instructions/goals 
 • Reasoning traces 
 • Code snippets 
 • Environment descriptions 
 • Conversation/memory | • RGB images 
 • RGB-D video 
 • 3D point clouds 
 • LiDAR 
 • Camera specs | • Image-text pairs 
 • Visual prompts 
 • RGB-D video 
 • Scene descriptions | • RGB-D video 
 • Text instructions 
 • Proprioceptive states 
 • Haptic/tactile feedback 
 • Action trajectories 
 • Success/failure signals |
| Output | • Logic-based sub-goals 
 • Code snippets 
 • Symbolic plans 
 • Safety constraints 
 • Feedback messages | • Visual features/embeddings 
 • Segmentation masks 
 • Detection/tracking 
 • Depth estimates 
 • Surface keypoints | • Image/video captions 
 • Semantic descriptions 
 • VQA answers 
 • Grounded maps 
 • Multi-modal alignment | • Motor commands 
 • End-effector poses 
 • Action policies 
 • Failure management |
| Strengths | • Instruction translation 
 • Task decomposition/sequencing 
 • Strong generalization 
 • Easy interaction | • Transfer learning 
 • Open-world perception 
 • Spatial awareness 
 • Distortion robustness | • Semantic understanding 
 • Open-vocabulary recognition 
 • Flexible perception 
 • Novel-entity generalization | • End-to-end simplicity 
 • Cross-platform generalization 
 • Action-reasoning integration |
| Limitations | • No embodiment/grounding 
 • Hallucinations 
 • High latency 
 • Input bias/sensitivity | • Domain specificity 
 • Weak physics modeling 
 • High compute cost | • No precise actions 
 • Incomplete grounding 
 • Needs external policy | • Large data demand 
 • Cross-platform inefficiency 
 • High latency 
 • Complex failure handling |
| Indicative models | • BERT (Devlin et al., 2019), GPT-3 (Brown et al., 2020), Llama 3 (Grattafiori et al., 2024), DeepSeek-V3 (Liu et al., 2024a) | • ViT (Yuan et al., 2021), DINOv2 (Oquab et al., 2024), SAM (Kirillov et al., 2023), R3M (Nair et al., 2023), VC-1 (Majumdar et al., 2023) | • CLIP (Radford et al., 2021), OWL-ViT (Minderer et al., 2022), BLIP (Li et al., 2022), SigLIP (Zhai et al., 2023) | • RT-1 (Brohan et al., 2023a), PaLM-E (Driess et al., 2023), RT-2 (Zitkovich et al., 2023), OpenVLA (Kim et al., 2025), Octo (Mees et al., 2024) |

transformer for real-time motor actions, while $\pi_{0.5}$ (Black et al., 2025) casts action generation as flow matching for stable continuous control at moderate cost.

- Specialization, adapters, and mixture-of-experts: VLAs exploit massive pre-trained backbones without running the full network per action. MoRE (Zhao et al., 2025a) activates a few sparse LoRA experts per step to expand capacity at no extra inference cost, while OpenVLA (Kim et al., 2025) pairs a Llama 2 model with a visual encoder and efficient fine-tuning for generalizable visuo-motor policies.

- Navigation and locomotion: VLAs act as semantic navigation planners translating language into movements for mobile and legged robots, often hierarchically. NaVILA (Cheng et al., 2025) generates mid-level actions as language that drive a visual locomotion RL policy, while VAMOS (Castro et al., 2025) decouples a generalist planner from a specialist affordance model encoding physical constraints.

- Operations, deployment, and safety: VLAs combine flexible, generalized reasoning with safety guarantees for real-world deployment. SafeVLA (Zhang et al., 2025a) frames safety alignment as constraint learning so that operation respects task rules beyond post-hoc filtering, while VLATest (Wang et al., 2025g) generates robotic manipulation scenes for systematically testing VLAs.

## 5.5 Comparative analysis and key insights

Having discussed in detail the various types of FMs (Sections 5.1-5.4), this section systematically examines the literature methods, providing a comparative analysis and critical insights for each methodological category. In this respect, Table 3 summarizes for each type of FM its: a) Primary functions, b) Input types, c) Output types, d) Key strengths, e) Critical limitations, and f) Indicative models. Moreover, representative literature methods per FM type are illustrated in Fig. 3.

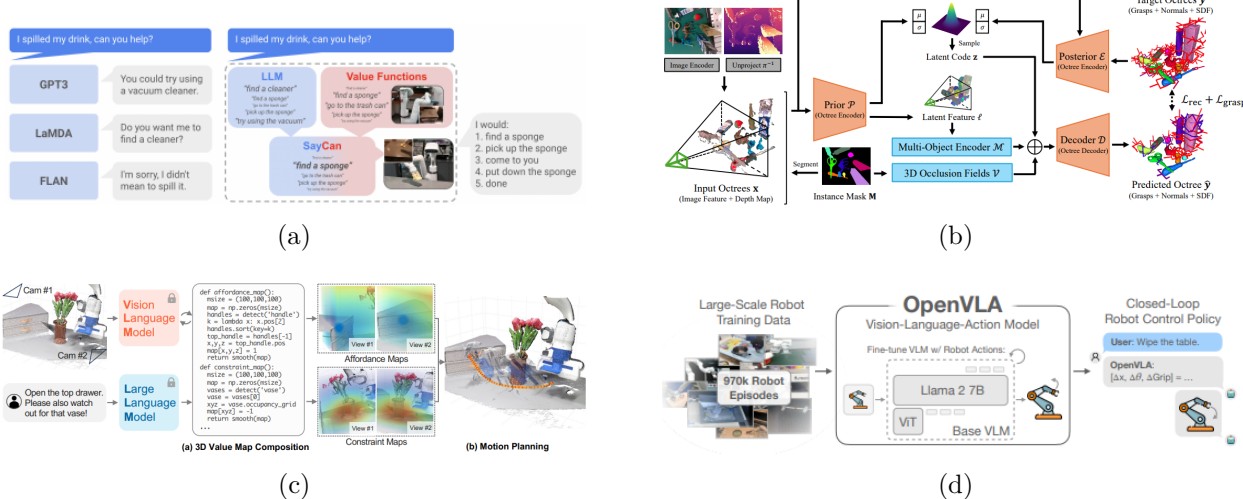

Figure 3: Representative literature methods per FM type: (a) LLMs (SayCan (Brohan et al., 2023b)), (b) VFMs (ZeroGrasp (Iwase et al., 2025)), (c) VLMs (VoxPoser (Huang et al., 2023c)), and (d) VLAs (OpenVLA (Kim et al., 2025)).

Based on Table 3, the following main observations can be drawn: a) LLMs excel at high-level, long-horizon planning and symbolic reasoning, operating on textual inputs but lacking physical grounding, b) VFMs are optimized for real-time, low-level perception, offering strong transfer and robustness, yet remaining domain-specific, c) VLMs bridge language and visual perception, but cannot define precise actions and need external policy generators, d) VLAs provide complete, end-to-end policies mapping multi-modal inputs to actions, at the cost of high data demands and latency, and e) overall, semantic breadth (LLMs, VLMs) trades off against real-time, embodied execution (VFMs, VLAs), motivating hybrid systems.

The full details of the FM types discussed in this section are provided in Section C of the supplementary document. In particular, for each FM type (i.e., LLMs, VFMs, VLMs, and VLAs), the latter contains the complete category definition, the corresponding main advantages and limitations, as well as the extensive list of representative methods per sub-category of supported functionality.

## 6 Neural network architectures

The type of the underlying neural network architecture that is employed in a FM solution largely dictates its capabilities. The main categories of NNs used in robotic FM methods are: a) Transformers, b) State-Space Models (SSMs), c) Diffusion Models (DMs), d) Convolutional and hybrid encoders, and e) Graphical models, as discussed in Section 4 and further detailed below.

### 6.1 Transformers

In the context of robotics, transformers are widely used for diverse tasks (e.g., high-level task planning, low-level policy learning, perception, and human-robot interaction), relying on the fundamental principle of formalizing them as a sequence modeling problem (Firoozi et al., 2025). By converting input data (e.g., states, actions, and images) into numerical tokens that are processed as a sequence, they enable long-range dependency modeling, architectural homogenization across tasks and modalities, and efficient high-level reasoning (Sanghai & Brown, 2024).

With respect to the input data modality, transformer-based systems can be classified into the following main categories:

- Vision Transformers (ViTs): ViTs comprise the fundamental architecture of multiple VFMs in robotics, using self-attention to capture global image relationships, unlike traditional local convolu-

tional methods (Yuan et al., 2021). DINOv2 (Oquab et al., 2024) applies self-supervised learning on web-scale datasets to estimate general-purpose visual features that transfer across tasks without extensive fine-tuning, supporting perception pipelines for scene understanding, semantic mapping, and low-level geometric control.

- Text transformers: Text transformers act as the language interface, planner, programmer, memory, and supervisor across robotic pipelines, providing a common semantic layer that connects high-level intent to grounded perception and control. Decoder-only models such as PaLM (Kim et al., 2024) and GPT-4 (Achiam et al., 2023) benefit from scale to improve robotic reasoning and planning, translating free-form language into structured plans, executable policy code, and run-time corrections.

- Multi-modal transformers: These models transform each sensorial stream into a sequence of tokens, project them to a shared latent space, and align them via a cross-attention or gated fusion mechanism, so that a single backbone can understand scenes, follow instructions, and select actions (Kim et al., 2025; Zitkovich et al., 2023). They also fuse vision with proprioceptive, geometric, tactile, audio, and thermal cues to generate embodiment-aware policies that scale across diverse robots and tasks.

### 6.2 State-space models

SSMs are increasingly adopted in robotics for realizing real-time control and long-horizon reasoning, by treating the sensorial input streams as latent states and producing output predictions in a step-by-step way (Liu et al., 2024b). By learning end-to-end system matrices with hardware-aware modeling (Gu & Dao, 2024), they offer linear-complexity scaling, stable long-horizon memory, and deployment efficiency suitable for embedded robotic solutions (Gu et al., 2022).

With respect to the input modalities, SSM-based systems can be classified into the following main categories:

- Visual SSMs: Visual SSMs replace attention mechanisms with selection-based ones, resulting into linear-complexity encoders that can be plugged into recognition, dense prediction, and tracking heads (Liu et al., 2024e). Long temporal context can be modeled at constant cost, which is particularly suitable for long video streams and multi-camera setups (Park et al., 2024).

- Policy/control SSMs: SSMs enable data-driven nonlinear reduction in complex systems, such as modeling hysteresis and memory effects. FlowRAM (Wang et al., 2025c) adopts region-aware selective-state policies with flow-matching objectives to learn precise skills from limited demonstrations, while vision-driven locomotion incorporates depth and proprioception through stacked selective-state formalisms and end-to-end RL (Wang & Tao, 2026).

- Multimodal SSMs: A single SSM trunk can be pretrained on long videos, robot logs, and demonstrations to support perception, planning interfaces, and action heads. SSM-based VLA models fuse vision, language, and proprioception as token streams to generate actions with lower latency and memory than attention-only implementations (Tsuji, 2025), making long-horizon policies more practical (Liu et al., 2024b).

### 6.3 Diffusion models

DMs generate robot behaviours by reversing a gradual noising process, where a forward pass adds Gaussian noise to the input data and a learned reverse model denoises back to the original space (e.g., actions, trajectories, and sub-goals) (Ho et al., 2020). Implemented as conditional denoising policies, they are robust across manipulation tasks and serve as generative heads on pretrained backbones, offering multi-modal modelling, composite conditioning, and trajectory-level decision making (Chi et al., 2025; Liang, 2025).

With respect to the conditioning type, DM-based systems can be classified into the following main categories:

- Vision-conditioned DMs: DMs translate visual goals into structured information for control. Image-goal generation and rearrangement priors estimate object- and scene-level targets that downstream

controllers follow (Kapelyukh et al., 2023), while pretrained image-editing DMs generate sub-goal images from language instructions and current camera views to guide goal-conditioned policies in real-world settings (Black et al., 2024).

- Proprioception-, force-, and haptic-conditioned DMs: Visuomotor diffusion policies treat action sequences as denoised samples conditioned on images and robot states, handling multi-modal actions and improving stability for manipulation (Chi et al., 2025). When contact requirements are present, conditioning on haptics and force signals can be incorporated, e.g., in visual-tactile slow-fast policies for contact-rich skills (Shukla et al., 2025).

- Language-conditioned DMs: Textual inputs serve as a guiding signal, where the denoising mechanism modulates goals, trajectories, or sub-goals, simplifying task setup and execution (Bjorck et al., 2025). Further works demonstrate the growing use of language prompts for manipulation and planning tasks based on diffusion backbones (Wolf et al., 2025).

- Human behaviour-conditioned DMs: Diffusion objectives can target early-stage human motion prediction to infer intent, improving intuitiveness and comfort in human-robot interaction without changing the controller structure. The Legibility Diffuser (Bronars et al., 2024) generates intent-expressive collaborative motions that humans find easier to understand, while still completing the task efficiently (Ng et al., 2023).

## 6.4 Convolutional and hybrid encoders

Visual encoders typically comprise the main perception module of any robotic solution, translating raw pixel data into latent representations that a robot can use for subsequent planning (Nair et al., 2023). The choice between CNNs and hybrid CNN-transformer implementations trades off local spatial precision against global context modeling, offering zero-shot generalization, robustness to noise, and reduced need for training data (Brohan et al., 2023a).

With respect to the encoder and integration type, the following main categories can be identified:

- CNN encoders: CNNs excel at capturing low-level spatial details such as edges, textures, and object boundaries, due to their local receptive fields. R3M (Nair et al., 2023) freezes a ResNet-50 trained on Ego4D to improve manipulation in both simulation and real-world scenarios, while EfficientNet-B3 is employed as a visual encoder for real-time, goal-conditioned navigation and exploration (Sridhar et al., 2024).

- CNN-transformer hybrids: Hybrid architectures combine a CNN for pixel-level information with a transformer for context and action aspects. RT-1 (Brohan et al., 2023a) encodes frames with a FiLM-conditioned EfficientNet, compresses them with TokenLearner, and predicts discrete actions using a transformer, while BC-Z/PaLM-SayCan (Brohan et al., 2023b) couple a lightweight ResNet with a shallow attention network for instruction-conditioned policies.

- CNN tokenizers inside generalist agents: Generalist agents convert high-resolution images into compact tokens prior to sequence modeling. Gato (Reed et al., 2022) employs a small ResNet image tokenizer feeding visual, text, and proprioception information to a single transformer, while RoboCat (Bousmalis et al., 2024) uses a pretrained VQ-GAN tokenizer and a transformer to adapt across robots and tasks via self-improvement cycles.

- CNN-conditioned diffusion policies: Diffusion policies condition a temporal U-Net on CNN features to generate diverse, yet feasible action chunks. Diffusion Policy (Chi et al., 2025) employs ResNet-18 features for manipulation while preserving low added latency, whereas DiffuserLite (Dong et al., 2024) uses progressive refinement with a frozen MobileNet-V3 encoder to reach real-time prediction rates on embedded platforms.

## 6.5 Graphical models

In the context of robotic FM methods, graphs introduce additional capabilities for connecting low-level, raw sensorial data with high-level, structured reasoning (Maggio et al., 2024). Unlike architectures that process data as matrices or sequences, graphs model the environment as a set of interconnected entities (e.g., scene graphs with nodes for objects and edges for spatial, semantic, or functional relationships), offering combinatorial generalization, permutation invariance, sample efficiency, and increased explainability (Gu et al., 2024).

With respect to the graph and function type, the following main categories can be identified:

- Scene graphs: Open-vocabulary 3D scene graphs associate vision-language features with real-world entities, remaining compact compared to dense maps and enabling robots to query targets, to reason about relationships, and to define sub-goals to planners (Gu et al., 2024). More recently, graphs are constructed online from RGB-D streams, using hierarchical structures for language-grounded navigation (Werby et al., 2024).

- Shared graphs: Compressed-form scene graphs allow bandwidth-limited sharing and map merging, while maintaining open-vocabulary query capabilities. Decentralized visual FMs estimate peer poses and produce local Bird's-Eye View maps on embedded hardware, reducing communication requirements without losing key semantic information (Blumenkamp et al., 2025; Gu et al., 2025).

- Graph neural networks: Graph Neural Networks (GNNs) enable message passing over task, object, and agent graphs for allocation, scheduling, and policy conditioning in a data-driven way. Recent hybrid cognitive pipelines couple GNN-based scene graphs with LLM or symbolic planners, keeping plans physically feasible while still following language goals (Tong et al., 2026; Strader et al., 2025).

- Embodiment graphs: Embodiment graphs encode robot joint information and the links between them, allowing a single learned policy to adapt across platforms. Attention or message passing follows the learned graph connectivity, boosting zero-shot transfer to new morphologies and supporting reusable controllers across different robots (Patel & Song, 2025).

## 6.6 Comparative analysis and key insights

Having discussed in detail the various types of neural network architectures (Sections 6.1-6.5), this section systematically examines the literature methods, providing a comparative analysis and critical insights for each category. In this respect, Table 4 summarizes for each type of architecture its: a) Primary functions, b) Main mechanisms, c) Key strengths, d) Critical limitations, and e) Indicative models. Moreover, representative literature methods incorporating different NN architecture types are illustrated in Fig. 4.

Based on Table 4, the following main observations can be drawn: a) Transformers excel at multi-modal alignment and high-level reasoning through global self-attention, but their quadratic complexity and discrete tokenization hinder real-time control, b) SSMs provide linear-complexity temporal modeling and stable long-horizon memory for real-time edge control, yet offer weaker global context than full attention, c) DMs generate multi-modal, high-precision actions via score-based denoising, at the cost of sampling latency and the absence of built-in safety guarantees, d) CNN and hybrid encoders serve as robust, zero-shot perceptual backbones for pixel-level grounding, but suffer high latency and sensitivity to distribution shifts, and e) Graphical models support structured, relational, and causal reasoning with strong sample efficiency, though they incur message-passing overhead and struggle with dynamic topologies.

The full details of the neural network architectures discussed in this section are provided in Section D of the supplementary document. In particular, for each architecture type (i.e., transformers, SSMs, DMs, convolutional and hybrid encoders, and graphical models), the latter contains the complete category definition, the corresponding main advantages and limitations, as well as the extensive list of representative methods per sub-category.

Table 4: Neural network architectures: Comparative analysis and key insights.

| Architecture | Transformers | SSMs | DMs | CNNs/hybrid | Graphical models |
|---|---|---|---|---|---|
| Primary functions | • Multi-modal alignment
• High-level reasoning
• Task decomposition
• Cross-embodiment transfer | • Sequence modeling
• Real-time edge control
• State estimation
• Contextual memory | • Action generation
• Precise manipulation
• Receding-horizon control
• Score estimation | • Feature detection
• Spatial grounding
• Multi-objective perception
• Object classification | • Causal reasoning
• Structured planning
• Relational grounding
• State transitions |
| Main mechanisms | • Global self-attention
• Positional encoding
• Autoregressive prediction
• Chain-of-thought | • Dynamics discretization
• Selective scan operators
• Hardware-aware kernels
• Input-dependent gating | • Score-based denoising
• Langevin dynamics
• Action chunking
• Latent space diffusion | • Convolutional layers
• Local connectivity
• Parameter sharing
• Pooling operators | • Graph neural networks
• Symbolic reasoning
• Entity masking
• Scene graph serialization |
| Strengths | • Long-range dependency
• Architectural homogenization
• Planning efficiency | • Linear scaling
• Long-horizon memory
• Deployment efficiency | • Multi-modal modelling
• Composite conditioning
• Trajectory decision-making | • Zero-shot generalization
• Noise robustness
• Data efficiency | • Combinatorial generalization
• Permutation invariance
• Sample efficiency |
| Limitations | • Quadratic complexity
• Discrete tokenization
• Context contradiction | • Weak cross-token check
• Limited global context
• Hybrid-design need | • Sampling latency/cost
• No safety guarantees
• Conditioning drift | • High latency
• Fine-detail loss
• Distribution shifts | • Computational overhead
• Dynamic topology
• Latent space integration |
| Indicative models | • RT-2 (Zitkovich et al., 2023), PaLM-E (Driess et al., 2023), Gato (Reed et al., 2022), OpenVLA (Kim et al., 2025), Octo (Mees et al., 2024) | • RoboMamba (Liu et al., 2024b), Mamba (Gu & Dao, 2024), AnoleVLA (Takagi et al., 2026), Decision Mamba (Huang et al., 2024b) | • Diffusion Policy (Chi et al., 2025), Diffuser (Janner et al., 2022), Motion Planning Diffusion (Carvalho et al., 2023), M2Diffuser (Yan et al., 2025) | • RT-1 (Brohan et al., 2023a), R3M (Nair et al., 2023), VC-1 (Majumdar et al., 2023), MVP (Wei et al., 2022) | • GRID (Ni et al., 2024), ConceptGraphs (Gu et al., 2024), HOV-SG (Werby et al., 2024), Open3DSG (Koch et al., 2024) |

## 7 Learning paradigms

In order to develop robust, real-world robotic FM solutions, different/diverse learning techniques, principles, and approaches can be adopted. The most commonly met learning paradigms in the literature, which are typically combined in a comprehensive learning methodology, are: a) Supervised Learning (SL), b) Self-Supervised Learning (SSL), c) Fine-tuning, d) Domain Adaptation (DA), e) Imitation Learning (IL), f) Reinforcement Learning (RL), g) In-Context/prompt Learning (ICL), h) World Model (WM) learning, and i) Generative Learning (GL), as discussed in Section 4 and further detailed below.

### 7.1 Supervised learning

Supervised learning trains a FM on large-scale, labeled datasets, directly fitting a mapping from inputs (e.g., images, language instructions, and proprioceptive states) to target outputs (e.g., class labels, tokens, or action commands) under explicit human-provided supervision (Xiao et al., 2025). This offers high task accuracy, stable optimization, and effective transfer of broad semantic priors, when paired with internet-scale labeled corpora (Zitkovich et al., 2023). In this context, PaLM-E (Driess et al., 2023) is trained on web-scale labeled multi-modal data jointly with embodied experiences, while RT-2 (Zitkovich et al., 2023) couples web-scale vision-language supervision with labeled robot manipulation data to enable open-vocabulary, instruction-following control.

### 7.2 Self-supervised learning

SSL techniques employ 'pretext tasks' (e.g., predicting the next video frame or reconstructing a masked image patch) and large quantities of unlabeled data streams to enable FMs to acquire common-sense knowledge regarding physics, object permanence, and spatial relationships (He et al., 2022; Oquab et al., 2024). This

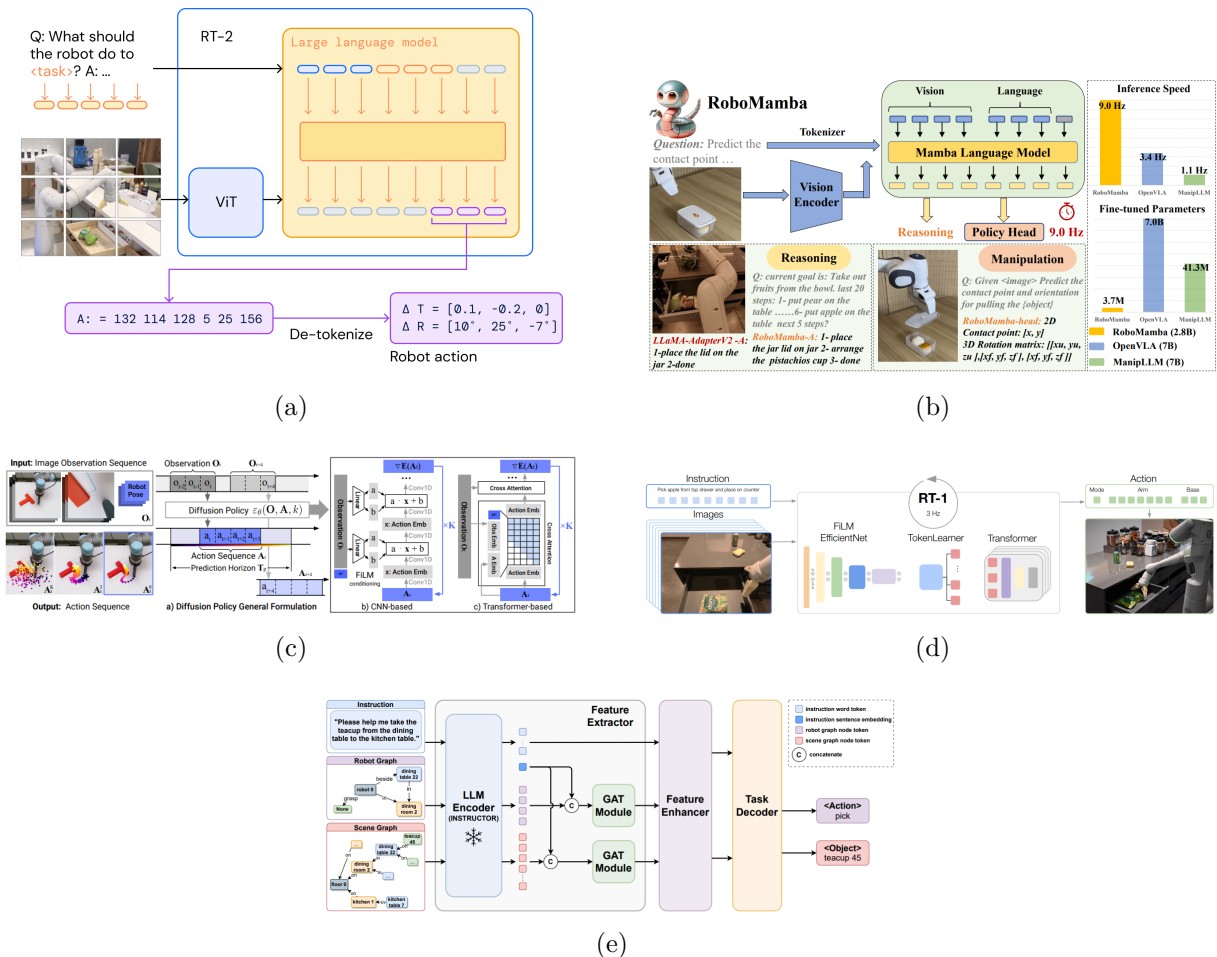

Figure 4: Representative literature methods incorporating different NN architecture types: (a) Transformers (RT-2 (Zitkovich et al., 2023)), (b) State-space models (RoboMamba (Liu et al., 2024b)), (c) Diffusion models (Diffusion Policy (Chi et al., 2025)), (d) Convolutional and hybrid encoders (RT-1 (Brohan et al., 2023a)), and (e) Graphical models (GRID (Ni et al., 2024)).

offers massive scalability, sample efficiency, zero-shot generalization, and autonomous improvement, without requiring human supervision (Nair et al., 2023; Assran et al., 2025).

The most common SSL techniques used for developing robotic FM solutions are:

- Masked Autoencoder (MAE): The model learns by reconstructing missing or corrupted parts of the input data, modeling robust spatial and temporal features. Image MAE (He et al., 2022) and its video counterpart VideoMAE (Tong et al., 2022) are widely used for producing robust visual backbone networks.

- Contrastive learning: This aligns matched views (and separates mismatched ones) to create discriminative features and to support open-vocabulary grounding, linking language to perception. CLIP (Radford et al., 2021) establishes vision-language alignment at scale, while egocentric robot features such as R3M (Nair et al., 2023) improve manipulation sample-efficiency from human-performing videos.

- Autoregressive sequence modeling: This predicts the next token in vision, language, or action streams to capture long-horizon structure and to enable unified perception-to-policy modeling. Generalist

agents, such as Gato (Reed et al., 2022), condition on images, text, and proprioception to produce actions across many tasks, while LLMs, like GPT-4 (Achiam et al., 2023), showcase cross-domain reasoning in embodied pipelines.

- World model learning: Also reported as an individual paradigm (Section 7.8), WMs learn to predict how the world changes in response to specific actions, encoding causal relations. Representative methods include Dreamer-style agents on physical robots (Wu et al., 2023b) and compositional video world models such as RoboDreamer (Zhou et al., 2024a).

### 7.3 Fine-tuning

Fine-tuning adapts the internet-scale acquired knowledge of a pretrained FM to specific physical-world execution settings, adjusting its common-sense knowledge structures to a specific robot, sensor suite, or task, using a smaller, in-domain, annotated dataset (Yu et al., 2025b). This offers sample efficiency, domain adaptation, and improved precision for the target task (Mees et al., 2024).

The most common fine-tuning techniques in robotics are:

- Full fine-tuning: This updates all FM weights using a relatively small, in-domain robot dataset, to re-target a pretrained policy to a new robot, sensor setup, or task. Recent generalist policies report fast adaptation to new observation and action spaces on standard GPUs, using full fine-tuning as the baseline learning method (Mees et al., 2024; Kim et al., 2025).

- Low-Rank Adaptation (LoRA): This keeps the original FM weights unchanged and adds two low-rank matrices that incorporate the required modifications. OpenVLA (Kim et al., 2025) demonstrates LoRA-based tuning on the large-scale Open-X-Embodiment dataset, while more recent quantized variants target adaptation on resource-constrained platforms (Williams et al., 2026).

- Quantized parameter-efficient fine-tuning: This extends PEFT by combining low-precision weights with LoRA-style adapters to keep latency and memory low on embedded hardware, while retaining most full-precision performance. LiteVLA (Williams et al., 2026) reports that 4-8 bit quantization plus adapters preserves high recognition rates, while enabling real-time control on smaller platforms (Williams et al., 2025).

- Action space remapping: This integrates lightweight encoders/decoders or tokenizers so that the core policy can be fine-tuned to new sensors or actuators without retraining the FM from scratch. Generalist policies, such as Octo (Mees et al., 2024) and RT-2 (Zitkovich et al., 2023), re-target a FM to multiple robots and grippers with modest additional data.

### 7.4 Domain adaptation

Domain Adaptation (DA) bridges the critical gap between FMs pre-trained on massive internet-scale data and their deployment in real-world environments, equipping them with the required physical-world grounding for given application scenarios (the 'sim-to-real' transfer challenge) (Da et al., 2025). This reduces the need for real-world samples, improves robustness to sensor shifts, and lowers the risk of damage during the learning phase (Tayyab Khan & Waheed, 2025).

The most common DA techniques in robotics are:

- Sim-to-real transfer: This trains a FM using large corpora of simulated/synthetic data, evaluates performance in simulation suites, and eventually calibrates on real-world hardware (typically using some real-world data). Humanoid and manipulation solutions demonstrate the efficiency of this approach (Luo et al., 2026; Deng et al., 2025).

- Real-to-sim-to-real transfer: This replays real-world robot trajectories in high-fidelity simulations, diversifies the scenes and objects, synthesizes new training data (often with domain randomization), and eventually calibrates the FM to real-world specifications. Various works demonstrate the validity of this approach in diverse operational settings (Zhu et al., 2025; Fang et al., 2025).

### 7.5 Imitation learning

Imitation Learning (IL) enables a model to learn directly from a (human) expert via teleoperation or video demonstrations of the desired skills, providing efficient multi-modal alignment that maps high-level language instructions and visual inputs to low-level motor commands (Zare et al., 2024). This requires no reward-signal definition, needs relatively few demonstrations, and offers easy training supervision (Kawaharazuka et al., 2025).

The most common IL techniques in robotics are:

- Behavioral cloning: This is a particular type of Supervised Fine-Tuning (SFT), imitating human expert demonstrations. RT-1 (Brohan et al., 2023a) learns a direct mapping of observations and language goals to actions using large-scale transformers, while RT-2 (Zitkovich et al., 2023) extends this with web-scale vision-language pretraining to open-vocabulary, instruction-following control.

- Diffusion-based IL: This represents a robot's behavior as a conditional denoising process, treating actions as a data distribution iteratively refined from random noise. Diffusion Policy (Chi et al., 2025) and Diff-Dagger (Lee et al., 2025b) generate action sequences that match expert behavior and handle multi-modal inputs, improving stability for long-horizon manipulation.

- In-context IL: This enables zero- or few-shot task adaptation on the fly, without updating model weights. ICRT (Fu et al., 2025) uses next-token prediction over sensorimotor streams for real-robot in-context imitation, while prompt demonstrations are augmented with explicit visual reasoning traces to infer task intent more reliably in ambiguous environments (Nguyen et al., 2026).

- Continual IL: This addresses the long-term memory and evolution challenges of robotic FMs, enabling a robot to sequentially acquire new skills over time without forgetting previously learned ones. LOTUS (Wan et al., 2024) introduces a continual imitation learning framework for skill acquisition by a real robot.

### 7.6 Reinforcement learning

Reinforcement Learning (RL) serves as the optimization formalism that bridges the gap between high-level, semantic reasoning (supported by FMs) and low-level, physical robot actions, involving the FM in a continuous, self-improvement cycle of perception, action, and evaluation (Tang et al., 2025). This enables self-improvement beyond the training data, increased generalization, and efficient sim-to-real implementation (Ter et al., 2025).

The most common RL techniques in robotics are:

- SFT-to-RL: This is a two-stage process where BC first learns a policy and RL subsequently improves it. RT-1 (Brohan et al., 2023a) and RT-2 (Zitkovich et al., 2023) show how large-scale BC produces powerful priors refined through interaction, while ExploRLLM (Ma et al., 2025) combines an LLM-guided exploration policy with a residual RL head to improve sample efficiency.

- LLM-guided reward design: This leverages LLMs to write/refine RL reward code and tune domain randomization, improving robustness and transfer. Eureka (Ma et al., 2024a) automates reward design and outperforms expert rewards on multiple tasks, while DrEureka (Ma et al., 2024b) extends this to the sim-to-real setting by jointly optimizing rewards and randomization.

- Preference-based RL: This replaces detailed/numeric rewards with preferences produced by VLMs (or adapted ones with small-scale human intervention). RL-VLM-F (Wang et al., 2024c) models rewards from VLM comparisons over image observations and text, while VARP (Singh et al., 2025) regularizes VLM-derived preferences with the agent's own rollouts to reduce misalignment.

- Offline-to-online RL: This trains a model offline on massive datasets, refines it via offline RL, and eventually applies online RL for real-world deployment. Embodied visual tracking is combined with a text-promptable encoder and offline RL for improved perception (Zhong et al., 2024), while FLaRe (Hu et al., 2025) applies large-scale RL fine-tuning on a pre-trained VLA for adaptive manipulation.

- World-model RL: World-model RL learns a generative world model with language-aware structure and uses it for RL-based policy improvement. RoboDreamer (Zhou et al., 2024a) factors video generation into compositional parts conditioned by language and visual goals, exhibiting robust performance on long-horizon tasks.

## 7.7 In-context/prompt learning

ICL and prompt learning adapt FMs at inference time by conditioning their behavior on demonstrations, examples, or task-specific instructions, without updating the underlying model weights (Fu et al., 2025; Yin et al., 2025c). This offers generalization to novel settings, high-level task planning from natural-language guidance, and flexible multimodal task specification (Yao et al., 2023).

The most common in-context/prompt learning techniques in robotics are:

- Language prompting: This uses natural language instructions to guide a robot's behavior, decision-making, and physical actions. Few-shot language prompts can encode demonstrations or templates, so that an LLM can output low-level actions or acquire new skills (Yin et al., 2025c; Liang et al., 2023).

- Reason-act prompting (ReAct): This interleaves natural language reasoning with physical actions, allowing a robot to decompose complex goals, validate its progress, and dynamically adjust its plan. Planning, execution, and re-planning can be performed in a single loop, including an LLM-based verification step (Yao et al., 2023; Grigorev et al., 2025).

- In-context imitation: This enables a FM to perform a novel task by observing a few videos or sensorimotor demonstrations, without permanent weight changes. A causal policy can parse short teleoperation trajectories as a prompt and predict the next action for new tasks without fine-tuning (Fu et al., 2025).

## 7.8 World model learning

World Models (WMs) allow robots to predict environmental changes in response to their actions, decoupling perception from action so that, instead of operating only on pixel values, the robot learns the underlying physics of the world (Li et al., 2025b). This incorporates 'imagined' experiences that reduce physical-world trials, models the rules of physics, and reduces operational delays (Zhang et al., 2025d).

The most common WM learning techniques in robotics are:

- Feature-space WMs: Instead of predicting each pixel, which is expensive and noisy, feature-space WMs map visual inputs to an abstract feature space and predict the future there. Future DINOv2 patch embeddings are predicted from offline trajectories and action sequences are optimized in the embedding space for zero-shot planning (Zhou et al., 2025b).

- Latent-action WMs: These learn to model the underlying physics and intent of actions, rather than specific skills. Continuous latent actions are discovered from videos and an auto-regressive WM conditioned on those actions transfers skills across scenes and embodiments with small-scale finetuning (Gao et al., 2025b).

- Compositional video WMs: These factorize the environment into its constituent parts (objects, relationships, and action primitives) and recombine them to generate future scenarios. Videos are factorized into objects and relations so the model can synthesize plans for unseen goal-scene combinations and guide long-horizon decisions (Zhou et al., 2024a).

- JEPA-style WMs: These predict an abstract meaning of what will happen next, enabling robots to plan complex tasks without being distracted by irrelevant noise. Joint predictions of short-horizon actions and abstract observations couple imitation with predictive learning to reduce control-error accumulation (Vujinovic & Kovacevic, 2025).

### 7.9 Generative learning

Generative Learning (GL) enables robots to imagine future states, to synthesize training data, and to propose complex action sequences, leveraging the capability of generative FMs to produce large quantities of data samples and alleviating the need for extensive high-quality robotic interaction data (Zhang et al., 2025c). This offers increased zero-shot generalization, multi-modality handling, and long-horizon planning (Liu et al., 2025).

The most common GL techniques in robotics are:

- Autoregressive sequence modeling: This predicts the next action or state based on previous observations. PACT (Bonatti et al., 2023) trains a causal transformer to predict the next observation-action token, so that a single model captures long-horizon structure across tasks, while long-horizon manipulation is modeled through sequential generation of action tokens (Zhang et al., 2025g).

- Diffusion-based action policies: This employs a diffusion model to generate a chunk of actions at once via a gradual denoising process. Diffusion Policy (Chi et al., 2025) learns a conditional denoising process that samples action sequences for multi-modal behaviors and stable visuomotor control, while the Legibility Diffuser (Bronars et al., 2024) is an intent-expressive variant.

- Generative video and scene synthesis: This creates a model of physical reality, enabling robots to imagine, simulate, and plan actions prior to real-world execution. RoboDreamer (Zhou et al., 2024a) employs compositional video WMs, while ReBot (Fang et al., 2025) uses a real-to-sim-to-real synthesis approach.

### 7.10 Comparative analysis and key insights

Having discussed in detail the various types of learning paradigms (Sections 7.1-7.9), this section systematically examines the literature methods, providing a comparative analysis and critical insights for each type. In this respect, Table 5 summarizes for each learning paradigm type its: a) Primary function, b) Main mechanisms, c) Primary data source, d) Key strengths, e) Critical limitations, and f) Indicative models.

Based on Table 5, the following main observations can be drawn: a) Supervised and self-supervised learning build general-purpose representations, b) Fine-tuning and domain adaptation ground generalist models to specific hardware and tasks, at the risk of catastrophic forgetting and negative transfer, c) Imitation learning offers easy, reward-free supervision from demonstrations, yet suffers covariate shift and data-quality dependence, d) Reinforcement learning enables self-improvement beyond demonstrations, but is sample-inefficient and reward-engineering intensive, and e) In-context, world-model, and generative learning support continual adaptation and future-state reasoning, though they face prompt sensitivity, hallucinations, and inference latency.

The full details of the learning paradigms discussed in this section are provided in Section E of the supplementary document. In particular, for each learning paradigm (i.e., supervised learning, self-supervised learning, fine-tuning, domain adaptation, imitation learning, reinforcement learning, in-context/prompt learning, world model learning, and generative learning), the latter contains the complete category definition, the corresponding main advantages and limitations, as well as the extensive list of representative methods per sub-category.

## 8 Learning stages

During the overall learning process of a robotic FM, the particular phase at which knowledge is incorporated largely defines the type/nature of the acquired skills, algorithmic/development details, and key assumptions about the model behavior. In this context, the main learning stages identified in the literature are: a) Pre-training, b) Offline fine-tuning, c) Online adaptation, and d) Continuous learning, as discussed in Section 4 and further detailed below.

Table 5: Learning paradigms: Comparative analysis and key insights.

| Learning paradigm | Primary function | Main mechanisms | Primary data source | Strengths | Limitations | Indicative models |
|---|---|---|---|---|---|---|
| Supervised learning | • General representation learning | • Large-scale labeled supervision | • Internet-scale data | • Data efficiency
• Zero-shot generalization
• Knowledge transfer
• Emergent reasoning | • Embodiment gap
• Sim-to-real gap
• High compute cost
• Safety concerns | • PaLM-E (Driess et al., 2023), RT-2 (Zitkovich et al., 2023), Octo (Mees et al., 2024), OpenVLA (Kim et al., 2025) |
| Self-supervised learning | • Label-free representation learning | • Pretext tasks | • Unlabeled robot trajectories
• Egocentric videos
• Raw sensor data | • Scalability
• Sample efficiency
• Zero-shot generalization
• Autonomous improvement | • Embodiment gap
• High compute cost
• Hallucinations | • R3M (Nair et al., 2023), MVP (Wei et al., 2022), DINOv2 (Oquab et al., 2024), Masked Autoencoders (He et al., 2022) |
| Fine-tuning | • Task/domain adaptation | • Supervised update | • Action-labeled demonstrations
• Domain-specific instructions | • Sample efficiency
• Domain adaptation
• Improved precision | • Catastrophic forgetting
• Overfitting
• Annotation need | • OpenVLA (Kim et al., 2025), Octo (Mees et al., 2024), RoboCat (Bousmalis et al., 2024), RT-2 (Zitkovich et al., 2023) |
| Domain adaptation | • Sim-to-real bridging | • Feature alignment
• Adversarial training
• Distribution reweighting | • Synthetic & sparse real data | • Reduced real-world data
• Sensor-shift robustness
• Transferability | • Negative transfer
• Training instability
• Low predictability | • DrEureka (Ma et al., 2024b), Gen2Sim (Katara et al., 2024), ReBot (Fang et al., 2025), VR-Robo (Zhu et al., 2025) |
| Imitation learning | • Expert behavior replication | • Behavioral cloning | • Expert teleoperation
• Kinesthetic teaching
• Human videos | • No reward signal
• Data efficiency
• Easy supervision | • Data-quality dependence
• Covariate shift
• Causal confusion | • RT-1 (Brohan et al., 2023a), Gato (Reed et al., 2022), Octo (Mees et al., 2024), BC-Z (Jang et al., 2022) |
| Reinforcement learning | • Interaction-based policy learning | • Reward-driven optimization | • Simulation/real interaction data | • Self-improvement
• Generalization
• Efficient sim-to-real | • Sample inefficiency
• Reward engineering
• Credit assignment | • Eureka (Ma et al., 2024a), DrEureka (Ma et al., 2024b), RL-VLM-F (Wang et al., 2024c), ExploRLLM (Ma et al., 2025) |
| In-context/prompt learning | • Prompt-based task adaptation | • Frozen-model inference | • Multi-modal instructions
• Observation-action pairs | • Novel-setting generalization
• Task planning
• Multi-modal flexibility | • Prompt sensitivity
• Limited grounding
• Inference latency | • SayCan (Brohan et al., 2023b), VIMA (Jiang et al., 2023), ICRT (Fu et al., 2025), Instruct2Act (Huang et al., 2023b) |
| World model learning | • Environment dynamics prediction | • Latent transition functions | • Interaction data | • Imagined experiences
• Physics modeling
• Reduced delays | • Hallucinations
• Cumulative errors
• High compute cost | • Dreamer (Hafner et al., 2020), RoboDreamer (Zhou et al., 2024a), ACT-JEPA (Vujinovic & Kovacevic, 2025), AdaWorld (Gao et al., 2025b) |
| Generative learning | • Data/plan/trajectory synthesis | • Data distribution modeling | • Massive multi-modal datasets | • Zero-shot generalization
• Multi-modality handling
• Long-horizon planning | • Sim-to-real gap
• Hallucinations
• Inference latency | • Diffuser (Janner et al., 2022), Diffusion Policy (Chi et al., 2025), DALL-E-Bot (Kapelyukh et al., 2023), Gen2Sim (Katara et al., 2024) |

## 8.1 Pre-training

The pre-training stage estimates robust, general-purpose representations of robotic data by processing massive (often internet-scale) amounts of diverse data from multiple platforms, modeling the cross-correlations among vision, language, and action (Li et al., 2024). This offers increased generalization, robust zero-shot capability,

and accurate multi-modal mapping across words, visual concepts, and physical actions (Kawaharazuka et al., 2025).

The most common learning paradigms adopted during the pre-training stage are:

- Supervised learning: Supervised learning equips a model with a foundational understanding of the world from high-quality, diverse, labeled data. PaLM-E (Driess et al., 2023) is jointly trained on web-scale multi-modal data and embodied experiences, while RT-2 (Zitkovich et al., 2023) is constructed using web and robot manipulation data.

- Self-supervised learning: SSL enables robots to operate beyond narrow, task-specific programming, towards generalized intelligence capabilities. DINOv2 (Oquab et al., 2024) and VideoMAE (Tong et al., 2022) learn perception priors from unlabeled data, while R3M (Nair et al., 2023) extends this by incorporating egocentric features.

- Imitation learning: IL learns a prior distribution of successful behaviors directly from expert demonstrations. RT-1 (Brohan et al., 2023a) treats robot control as next-token prediction over multi-modal streams to inherit semantic and sensorimotor skills, while Octo (Mees et al., 2024) uses the Open-X-Embodiment trajectories to derive a generalist policy.

## 8.2 Offline fine-tuning

Offline fine-tuning bridges the knowledge gap between the general-purpose representations learned during pre-training and the specificities of a given physical-world application, targeting task and embodiment specialization (Hu et al., 2023). This reduces the need for training data, increases training stability, and enables knowledge distillation of only the necessary general-purpose representations (Firoozi et al., 2025).

The most common learning paradigms adopted during the offline fine-tuning stage are:

- Imitation learning: IL equips pre-trained models with the low-level precision skills for a specific application, learning specific motor commands by mimicking expert demonstrations. Octo (Mees et al., 2024) and OpenVLA (Kim et al., 2025) employ large-scale Open-X-Embodiment pretraining and then focus on new platforms using LoRA-style adapters, while LiteVLA (Williams et al., 2025) shows NF4 quantized LoRA can be tuned on CPU-only hardware.

- Reinforcement learning: RL enables robots to learn from a reward signal, focusing on actions that lead to successful task executions. A recurrent tracker is trained with conservative offline RL on VFM-annotated trajectories (Zhong et al., 2024), while FLaRe (Hu et al., 2025) applies large-scale RL fine-tuning to transformer-based policies for long-horizon mobile manipulation.

- Generative learning: GL specializes pre-trained models to specific environments, especially under sparse target-task constraints. ReBot (Fang et al., 2025) replays real trajectories in simulation and composes them into inpainted real backgrounds to adapt to new domains, while RoboDreamer (Zhou et al., 2024a) uses compositional WMs to generate imagined video plans as additional training data.

## 8.3 Online adaptation

Online adaptation equips robots with routines for learning in real-time from their own experiences, handling the distribution shift between offline training data and what is encountered during online deployment (Firoozi et al., 2025). This offers high-precision performance in the adapted environments, continuous improvement, and robustness to distribution shifts (Yuan et al., 2025).

The most common learning paradigms adopted during the online adaptation stage are:

- Domain adaptation: This handles the physical-world constraints and sensorial noise of a specific deployment scenario, re-calibrating the model's knowledge structures to the perceived environment. TTA-Nav (Piriyajitakonkij et al., 2024) adds a reconstruction decoder on a pre-trained policy to

denoise corrupted frames without gradient updates, while Phys2Real (Wang et al., 2025b) bridges sim-to-real gaps by combining FM priors with interaction-based estimations.

- Reinforcement learning: RL allows the robot to perform micro-adjustments to its general-purpose knowledge, based on sensory feedback and exploration. Self-improving embodied FMs refine pre-trained policies from reward and success estimation across a robot fleet (Ghasemipour et al., 2025), while RL-VLM-F (Wang et al., 2024c) estimates rewards using a VLM that compares trajectory snippets with language goals.

- In-context/prompt learning: This enables zero- or few-shot specialization at deployment time, without updating model weights. ICRT (Fu et al., 2025) uses in-context imitation policies conditioned on a few recent demonstration trajectories, while LLM-based control stacks interleave reasoning and acting through ReAct-style prompting to monitor progress and to revise plans (Yao et al., 2023).

## 8.4 Continuous learning

Continuous Learning (CL), often termed lifelong learning, enables robots to acquire new skills or to adapt to new environments incrementally, without full retraining from scratch, employing conventional paradigms (e.g., IL, RL) in slower outer loops (Xiao et al., 2025). This offers increased adaptability for long-term deployment, improved scalability across a fleet, and reduced downtime (Firoozi et al., 2025).

The most common learning paradigms adopted during the continuous learning stage are:

- Domain adaptation: DA enables a model to continuously adjust its internal knowledge to dynamic, real-world settings. Action Flow Matching for Lifelong Learning (Murillo-González & Liu, 2025) incrementally aligns robot dynamics across sequential tasks for safe continual adaptation, while VR-Robo (Zhu et al., 2025) builds photorealistic digital twins from logged data to retrain and transfer navigation or locomotion policies.

- Imitation learning: IL constantly bridges the gap between a model's general knowledge and the high-precision requirements of a real-world case, given few expert demonstrations. SkillsCrafter (Wang et al., 2026a) realizes lifelong language-conditioned learning across sequential manipulation skills while reducing forgetting via symbolic skill distillation, while LOTUS (Wan et al., 2024) refines manipulation skills from demonstration streams.

- Reinforcement learning: RL supports continuous learning by enabling policies to improve through repeated interaction, autonomous practice, and reward-driven post-training over extended deployment horizons. Self-improving embodied FMs refine pretrained policies via autonomous practice from self-predicted rewards across a robot fleet (Ghasemipour et al., 2025), while LiReN (Stachowicz et al., 2024) shows navigation FMs can improve lifelong learning through online RL in open-world settings.

## 8.5 Comparative analysis and key insights

Having discussed in detail the different learning stages (Sections 8.1-8.4), this section systematically examines the literature methods, providing a comparative analysis and critical insights for each stage. In this respect, Table 6 summarizes for each learning stage its: a) Primary function, b) Data requirements, c) Learning paradigms, d) Computational requirements, e) Generalization capability, f) Key strengths, g) Critical limitations, and h) Indicative models.

Based on Table 6, the following main observations can be drawn: a) Pre-training builds general-purpose, zero-shot world knowledge from internet-scale data, but at extreme computational cost and with limited real-world grounding, b) Offline fine-tuning grounds these representations to specific tasks and embodiments with moderate data, yet lacks online exploration and it is sensitive to distribution shift, c) Online adaptation adjusts policies in real-time on edge hardware, trading low compute for sensitivity to noise and catastrophic forgetting, d) Continuous learning enables lifelong skill retention and fleet-level scalability, but faces the

Table 6: Learning stages: Comparative analysis and key insights.

| Learning stage | Pre-training | Offline fine-tuning | Online adaptation | Continuous learning |
|---|---|---|---|---|
| Primary function | • World-knowledge learning | • Task/embodiment grounding | • Real-time adjustment | • Lifelong retention |
| Data requirements | • Internet-scale data | • Expert demonstrations | • Interaction history | • Sequential streams |
| Learning paradigms | • Supervised learning
• Self-supervised learning
• Imitation learning | • Imitation learning
• Reinforcement learning
• Generative learning | • Domain adaptation
• Reinforcement learning
• In-context/prompt learning | • Domain adaptation
• Imitation learning
• Reinforcement learning |
| Computational requirements | • Extreme GPU | • Moderate-to-high GPU | • Low (edge) | • Moderate (incremental) |
| Generalization capability | • Zero-shot cross-domain | • Task-specific precision | • Local adjustment | • Cross-task transfer |
| Strengths | • Generalization
• Zero-shot capability
• Multi-modal mapping | • Data efficiency
• Training stability
• Knowledge distillation | • High precision
• Continuous improvement
• Shift robustness | • Adaptability
• Scalability
• Reduced downtime |
| Limitations | • High compute cost
• Weak real-world performance
• Safety concerns | • Distribution shift
• Data-quality dependence
• No online exploration | • Catastrophic forgetting
• Latency
• Noise sensitivity | • Catastrophic forgetting
• Stability-plasticity dilemma
• Memory overhead |
| Indicative models | • Open X-Embodiment/RT-X (O'Neill et al., 2024), GR00T N1 (Bjorck et al., 2025), PaLM-E (Driess et al., 2023), CLIP (Radford et al., 2021) | • Octo (Mees et al., 2024), OpenVLA (Kim et al., 2025), BC-Z (Jang et al., 2022), RT-2 (Zitkovich et al., 2023) | • TTA-Nav (Piriyajitakonkij et al., 2024), Phys2Real (Wang et al., 2025b), RL-VLM-F (Wang et al., 2024c), ICRT (Fu et al., 2025) | • DrEureka (Ma et al., 2024b), VR-Robo (Zhu et al., 2025), LOTUS (Wan et al., 2024), Self-Improving Embodied FMs (Ghasemipour et al., 2025) |

stability-plasticity dilemma and memory overhead, and e) Overall, the stages trade off general-purpose knowledge acquisition against specialized, real-time precision refinement.

The full details of the learning stages discussed in this section are provided in Section F of the supplementary document. In particular, for each learning stage (i.e., pre-training, offline fine-tuning, online adaptation, and continuous learning), the latter contains the complete category definition, the corresponding main advantages and limitations, as well as the extensive list of representative methods per adopted learning paradigm.

## 9 Robotic tasks

The introduction of FMs has led to transformative effects in the materialization and execution of all core robotic tasks, mainly by shifting the field from task-specific programming to general-purpose, multi-task agents. In this context, the main robotic tasks identified in the literature, where FM-based solutions have been applied, are: a) Perception, b) Planning, c) Navigation, d) Manipulation, and e) Human-robot interaction, as discussed in Section 4 and further detailed below.

### 9.1 Perception

Perception creates rich, semantic maps of the surrounding environment that enable robots to execute individual actions, realizing semantic grounding, object-affordance discovery, and contextual awareness (Kawaharazuka et al., 2024). The incorporation of FMs offers open-vocabulary recognition, zero-shot generalization, multi-modal fusion, and robustness to noise (Hu et al., 2023).

The main categories of perception methods are:

- Language-grounded detection and segmentation: This identifies (detection) and precisely outlines (segmentation) objects in the robot's environment based on natural language prompts. Grounding DINO (Liu et al., 2024c) provides phrase-based detections robust to clutter and in the zero-shot

setting, while SAM-style promptable segmenters (Kirillov et al., 2023) incorporate box, click, or text prompts into control pipelines interactively.

- Open-vocabulary 3D semantic mapping: This allows robots to perceive and localize objects of previously unseen categories in 3D space using natural language inputs. ConceptFusion (Jatavallabhula et al., 2023) builds open-set, language-searchable maps supporting multi-modal queries, while ConceptGraphs (Gu et al., 2024) estimates object nodes and relations so that planners can operate on a semantic scene graph instead of raw pixels.

- Pose estimation and affordance prediction: Pose estimation aligns an object's local coordinate system to the world frame, while affordance prediction detects the ways an object can be manipulated, with FMs linking semantics to spatial geometry. OV9D (Cai et al., 2024a) estimates category-agnostic 9-DoF pose without CAD models, while OpenAD (Nguyen et al., 2023) models zero-shot 3D affordances in a shared vision-language embedding space.

- Contact-centric and visuotactile perception: This analyzes and models the physics of robot interactions, with contact-centric approaches, using junction points as state and visuotactile ones integrating vision with tactile sensing. Tactile-VLA (Huang et al., 2025b) combines tactile streams with vision and language for insertion and assembly, while NeuralFeels (Suresh et al., 2024) enhances in-hand pose and shape estimation, when visual cues are uncertain.

- Long-term object tracking: This locates a given object or point across video frames, maintaining stability over time and recalling objects after occlusion, as required for long-horizon tasks. OVTrack (Li et al., 2023b) employs language and diffusion priors to generalize multi-object tracking to unseen categories, while DINO-MOT (Lee et al., 2024a) combines DINOv2 features with a memory mechanism for robust pedestrian tracking.

## 9.2 Planning

Planning serves as the fundamental bridge between high-level semantic reasoning and low-level motor control, with its primary usefulness lying in long-horizon task decomposition of a goal into sequential sub-goals prior to execution (Hu et al., 2023). The incorporation of FMs offers increased interpretability, generalization, reduced need for training data, and safety-constraint integration into the planning loop (Firoozi et al., 2025).

The main categories of planning methods are:

- Language-driven task decomposition: This breaks down high-level, long-horizon goals into logical sequences of primitive actions, often as structured, executable code. SayCan (Brohan et al., 2023b) grounds each action step on an affordance map so that abstract sub-goals map to concrete objects, while Code-as-Policies (Liang et al., 2023) generates short Python-like programs that render planning easier to inspect, to test, and to modify.

- Neuro-symbolic closed-loop reasoning: This combines the general-purpose knowledge of a FM with formal, logical checking to guarantee successful real-world operation. ISR-LLM (Zhou et al., 2024b) converts instructions to PDDL and iteratively refines plans via symbolic validation, while AutoTAMP (Chen et al., 2024b) translates instructions into TAMP representations using autoregressive re-prompting to correct errors.

- Multi-modal policy generation: This generates high-level plans or actions, by integrating language, vision, and embodied state. PaLM-E (Driess et al., 2023) combines language, vision, and proprioception for embodied reasoning, while RT-2 (Zitkovich et al., 2023) shows that language-aligned visual representations transfer web-scale semantic knowledge to real-world control.

- Execution-time validation and failure recovery: This monitors plans during deployment, verifying preconditions, detecting failures, and triggering corrective re-planning. Code-as-Monitor (Zhou et al., 2025a) introduces constraint-aware visual programming for reactive and proactive failure detection, while VLM-based monitoring frameworks, such as Guardian (Pacaud et al., 2025), support execution-time failure recovery in manipulation.

- Semantic multi-robot coordination: This capitalizes on the broad knowledge of FMs to coordinate multi-robot setups, reasoning about task dependencies, resources, and scheduling. LiP-LLM (Obata et al., 2024) builds a skill list and dependency graph, and uses linear programming to allocate tasks, while SMART-LLM (Kannan et al., 2024) assigns role-based tasks from a single high-level instruction.

### 9.3 Navigation

The usefulness of FMs in robot navigation lies in providing the necessary spatial common-sense knowledge in embodied AI settings, by processing the environment as a high-level semantic space instead of rigid spatial maps (Pan et al., 2025b). The incorporation of FMs offers open-world navigation, cross-embodiment transfer, and semantic reasoning that combines vision with language for instruction interpretation (Firoozi et al., 2025).

The main categories of navigation methods are:

- Semantic spatial grounding: This registers the objects in the environment, their relative position, and a concrete action plan for reaching them in natural-language form. VLMaps (Huang et al., 2023a) builds CLIP-indexed spatial memories that query objects and rooms without task-specific retraining, while open-vocabulary mapping methods, such as One Map to Find Them All (Busch et al., 2025), support multi-object navigation, dynamic environments, and functional queries.

- Instruction-following policies: This leverages the capability of FMs to interpret natural language instructions without a pre-defined map or hard-coded scripts. Generalist navigation models, such as ViNT (Shah et al., 2023c), formalize navigation as sequence prediction over images and poses across robots, while NaviLLM (Zheng et al., 2024a) unifies instruction following and embodied QA with schema-tuned prompts.

- End-to-end policies: This maps raw sensorial data directly to motor commands, instead of the conventional modular design of separate mapping, localization, and path planning. ViNT (Shah et al., 2023c) learns a generalizable visuomotor navigation policy across robots and environments, while DriveGPT-4 (Xu et al., 2024b) predicts low-level control signals directly from visual- and language-conditioned inputs.

### 9.4 Manipulation

The utility of FMs in robot manipulation lies in translating high-level, semantic, human-like instructions into the precise forces and movements needed to manipulate an object, with cross-embodiment learning as their main contribution (Li et al., 2024). The incorporation of FMs offers increased generalization to unseen objects, improved robustness via real-time feedback, and enhanced understanding of objects' physical properties (Sapkota et al., 2025).

The main categories of manipulation methods are:

- Language-to-action models: This interprets the semantic meaning of a natural language command and maps it to the physical world, without task-specific programming. RT-2 (Zitkovich et al., 2023) approaches manipulation as sequence modeling over multi-modal tokens, while OpenVLA (Kim et al., 2025) is a large vision-language-action policy that adapts to new platforms via small-scale fine-tuning.

- Retrieval-augmented imitation learning: This receives guidance from a large database of relevant previous demonstrations to predict future actions. DINOBot (Di Palo & Johns, 2024) detects similar demonstrations, via DINO feature correspondence, to estimate dense trajectories for one- and few-shot generalization, while STRAP (Memmel et al., 2025) retrieves sub-trajectories to augment few-shot imitation learning.

- Constraint-aware policy synthesis: This employs a FM to generate high-level control code or objectives bounded by physical, safety, and environmental constraints. CoPa (Huang et al., 2024a)

detects task-relevant parts with a multi-modal LLM and estimates spatial constraints translated into 6-DoF actions, while ReKep (Huang et al., 2025c) represents tasks as relational keypoint constraints optimized hierarchically for assembly.

- Semantic spatial maps: This generates representations combining 3D spatial geometry with semantic information about the objects. VoxPoser (Huang et al., 2023c) estimates constraints and affordances from language, creates 3D value maps, and applies zero-shot motion planning, while AdaRPG (Zhang et al., 2025f) leverages VLMs to infer part affordances guiding articulated-object manipulation.

## 9.5   Human-robot interaction

The usefulness of FMs in HRI is grounded on their ability for semantic, human-like reasoning and interpreting human intent, enabling robots to react to conversational instructions, to ask clarification questions, and to explain their actions (Zhao et al., 2025b). The incorporation of FMs offers rapid generalization, intuitive control, increased safety alignment, and efficient error recovery from user feedback (Xiao et al., 2025).

The main categories of HRI methods are:

- Conversational policy alignment: This uses natural language dialogue to dynamically adjust a robot's behavior in real-time to match a human's intent, preferences, and safety boundaries. DRAGON (Liu et al., 2024d) is a dialogue-based navigation framework that grounds free-form commands in visual landmarks and asks clarification questions, while PlanCollabNL (Izquierdo-Badiola et al., 2024) translates spoken instructions into editable collaborative plans.

- Reciprocal social tuning: This develops a semantic communication framework where humans and robots continuously adjust their behaviors, social cues, and expectations to harmonize. TidyBot (Wu et al., 2023a) learns user-specific clean-up preferences from a few examples and generalizes them to new scenes, while LAMS (Tao et al., 2025) extends this to assistive teleoperation, by switching control modes from user feedback.

- Active alignment and mitigation: This keeps a robot synchronized with human intent, while proactively handling errors or deviations. RoboVQA (Sermanet et al., 2024) queries egocentric video to check preconditions and to request human intervention on failures, while embodied LLM controllers incorporate human feedback to adjust plans when the state drifts from the goal (Bärmann et al., 2024).

## 9.6   Comparative analysis and key insights

Having discussed in detail the different robotic tasks (Sections 9.1-9.5), this section systematically examines the literature methods, providing a comparative analysis and critical insights for each task. In this respect, Table 7 summarizes for each robotic task its: a) Primary function, b) FM type, c) Input types, d) Output types, e) Key strengths, f) Critical limitations, and g) Indicative models. Moreover, representative literature methods per robotic task are illustrated in Fig. 5.

Based on Table 7, the following main observations can be drawn: a) Across all tasks, a fundamental transition occurs from isolated, task-specific modules to integrated, generalist architectures leveraging internet-scale pretraining, b) Perception achieves robust open-vocabulary recognition and semantic fluency, but suffers spatial imprecision and hallucinations, c) Planning attains interpretable long-horizon reasoning, yet depends on grounding to keep plans physically feasible, d) Navigation and manipulation emphasize embodiment, facing a trade-off between semantic-reasoning latency and real-time motor control, and e) HRI enables intuitive communication, though hindered by performance-latency trade-offs and the need to adhere to safety and social norms.

The full details of the robotic tasks discussed in this section are provided in Section G of the supplementary document. In particular, for each robotic task (i.e., perception, planning, navigation, manipulation, and human-robot interaction), the latter contains the complete category definition, the corresponding main advantages and limitations, as well as the extensive list of representative methods per sub-category.

Table 7: Robotic tasks: Comparative analysis and key insights.

| Robotic task | Perception | Planning | Navigation | Manipulation | Human-robot interaction |
|---|---|---|---|---|---|
| Primary function | • Semantic scene recognition
• Object detection
• 3D spatial grounding | • Goal decomposition
• Long-horizon reasoning
• Constraint-aware sequencing | • Goal-conditioned path-finding
• Obstacle avoidance
• Topological exploration | • Fine-motor control
• Grasping actions
• Object relocation | • Context-aware communication
• Social collaboration
• Command interpretation |
| FM type | • VFMs
• VLMs | • LLMs
• VLMs | • LLMs
• VFMs
• VLMs
• VLAs | • VLMs
• VLAs | • LLMs
• VLMs
• VLAs |
| Input | • RGB-D images
• 3D point clouds
• Textual prompts | • Natural-language goals
• Scene graphs
• Symbolic state
• Execution feedback | • RGB images
• Spatial maps
• Spatial memory
• Odometry measurements
• Language instructions | • Natural-language goals
• Proprioception/gripper state
• Video demonstrations
• Scene descriptions
• Tactile/force data | • Human speech
• Human gestures
• Interaction history
• Social context |
| Output | • Segmentation masks
• Scene graphs
• Semantic labels
• 3D maps
• Poses/affordances | • Plans/sub-goals/code
• Constraints/preconditions
• Task decompositions | • Velocity commands
• Spatial trajectories
• Target locations
• Route descriptions | • Action sequences
• Grasps/placements
• 6-DOF poses
• Gripper states
• Adapted policies | • Speech output
• Socially-aware motion
• Goal corrections
• Explanations/reports
• User preferences |
| Strengths | • Open-vocabulary recognition
• Zero-shot generalization
• Multi-modal fusion
• Noise robustness | • Interpretability
• Generalization
• Data efficiency
• Safety-constraint integration | • Open-world navigation
• Cross-embodiment transfer
• Semantic reasoning | • Generalization
• Robustness
• Embodiment capability | • Rapid generalization
• Intuitive control
• Safety alignment
• Error recovery |
| Limitations | • High latency
• Low explainability
• Hallucinations
• Spatial imprecision | • Logical gaps
• Limited grounding
• High latency
• Closed-loop complexity | • Sim-to-real gap
• Data scarcity
• High latency
• Safety/ethical concerns | • Data scarcity
• Safety concerns
• Low action precision | • Data scarcity
• Human bias
• Semantic drift |
| Indicative models | • CLIP (Radford et al., 2021), SAM (Kirillov et al., 2023), Grounding DINO (Liu et al., 2024c), DINOv2 (Oquab et al., 2024), ConceptGraphs (Gu et al., 2024) | • SayCan (Brohan et al., 2023b), Code-as-Policies (Liang et al., 2023), SayPlan (Rana et al., 2023), AutoTAMP (Chen et al., 2024b) | • LM-Nav (Shah et al., 2023a), GNM (Shah et al., 2023b), ViNT (Shah et al., 2023c), DRAGON (Liu et al., 2024d) | • RT-1 (Brohan et al., 2023a), OpenVLA (Kim et al., 2025), GR00T N1 (Bjorck et al., 2025), Diffusion Policy (Chi et al., 2025) | • PaLM-E (Driess et al., 2023), Gemini Robotics (Team et al., 2025), RoboVQA (Sermanet et al., 2024), TidyBot (Wu et al., 2023a), LAMS (Tao et al., 2025) |

# 10 Application domains

The incorporation of FMs in robotic solutions has greatly boosted multiple aspects of their core technologies (e.g., autonomy, complex decision-making, semantic reasoning, etc.). This has in turn facilitated their widespread use in a wide range of challenging real-world application domains, including: a) Agentic mobility, b) Industrial manipulation, c) Supply operations, d) Service robots, e) Medical robots, f) Cognitive agrisystems, g) Crisis agents, h) Maritime robotics, and i) Space robotics, as discussed in Section 4 and further detailed below. Additionally, representative literature methods per application domain are illustrated in Fig. 6. The full details of the application domains discussed in this section are provided in Section H of the supplementary document. In particular, for each application domain, the latter contains the complete domain definition, as well as the extensive list of representative methods.

## 10.1 Agentic mobility

The integration of FMs has induced a paradigm shift in autonomous movement, transitioning from rigid path-following routines to autonomous agentic solutions, where robots leverage high-level reasoning to understand mission objectives and to adapt to environmental changes. For semantic spatial grounding, VLMaps (Huang et al., 2023a) constructs CLIP-indexed spatial memories by ingesting visual-language features into 3D maps, enabling navigation to specific rooms or objects via natural language queries without task-specific retraining. For cross-embodiment and cross-site generalization, ViNT (Shah et al., 2023c) formalizes navigation as a sequence prediction problem over images and poses, enabling a single visual backbone to drive diverse mobile

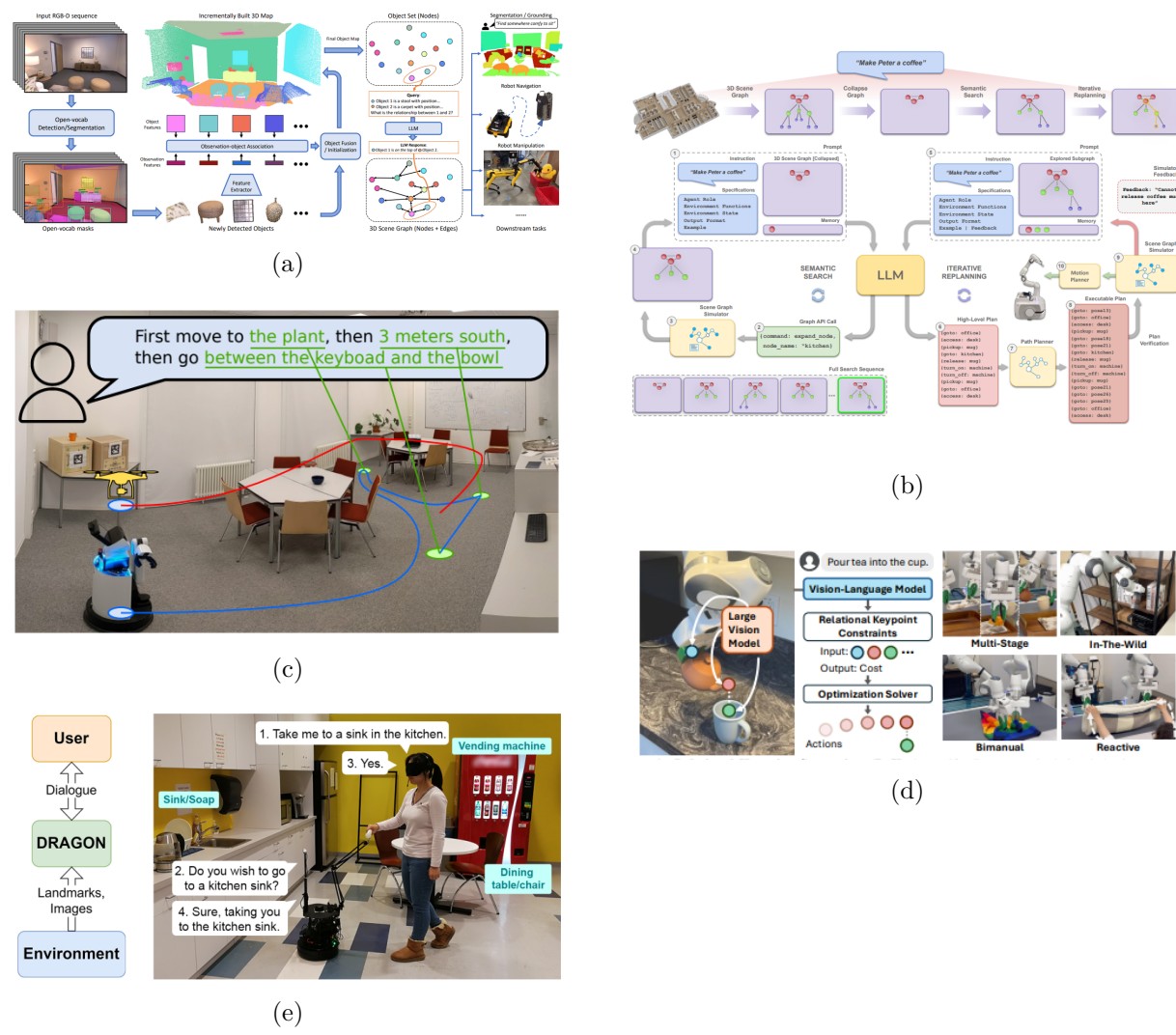

Figure 5: Representative literature methods per robotic task: (a) Perception (ConceptGraphs (Gu et al., 2024)), (b) Planning (SayPlan (Rana et al., 2023)), (c) Navigation (VLMaps (Huang et al., 2023a)), (d) Manipulation (ReKep (Huang et al., 2025c)), and (e) Human-robot interaction (DRAGON (Liu et al., 2024d)).

platforms across varied environments. In road environments, DriveGPT-4 (Xu et al., 2024b) conditions planning on linguistic reasoning to produce human-readable rationales and intermediate decisions that are easier to inspect, to debug, and to align with safety standards.

## 10.2 Industrial manipulation

FMs are fundamentally redefining industrial automation by replacing rigid, task-specific routines with flexible, general-purpose agents that excel in unstructured manufacturing environments. RFM-1 (Sohn et al., 2024) is deployed within high-throughput production cells, leveraging visuo-motor backbones trained on millions of real-world pick actions to achieve zero-shot generalization to novel objects. In order to address hardware variability, OpenVLA (Kim et al., 2025) utilizes parameter-efficient fine-tuning to simplify transfer across robotic platforms, reducing per-line retraining, while enabling operators to issue high-level, language-conditioned tasks. Moreover, RT-2 (Zitkovich et al., 2023) capitalizes on web-to-robot knowledge transfer to boost robustness against long-tail objects and complex instructions on assembly and kitting lines.

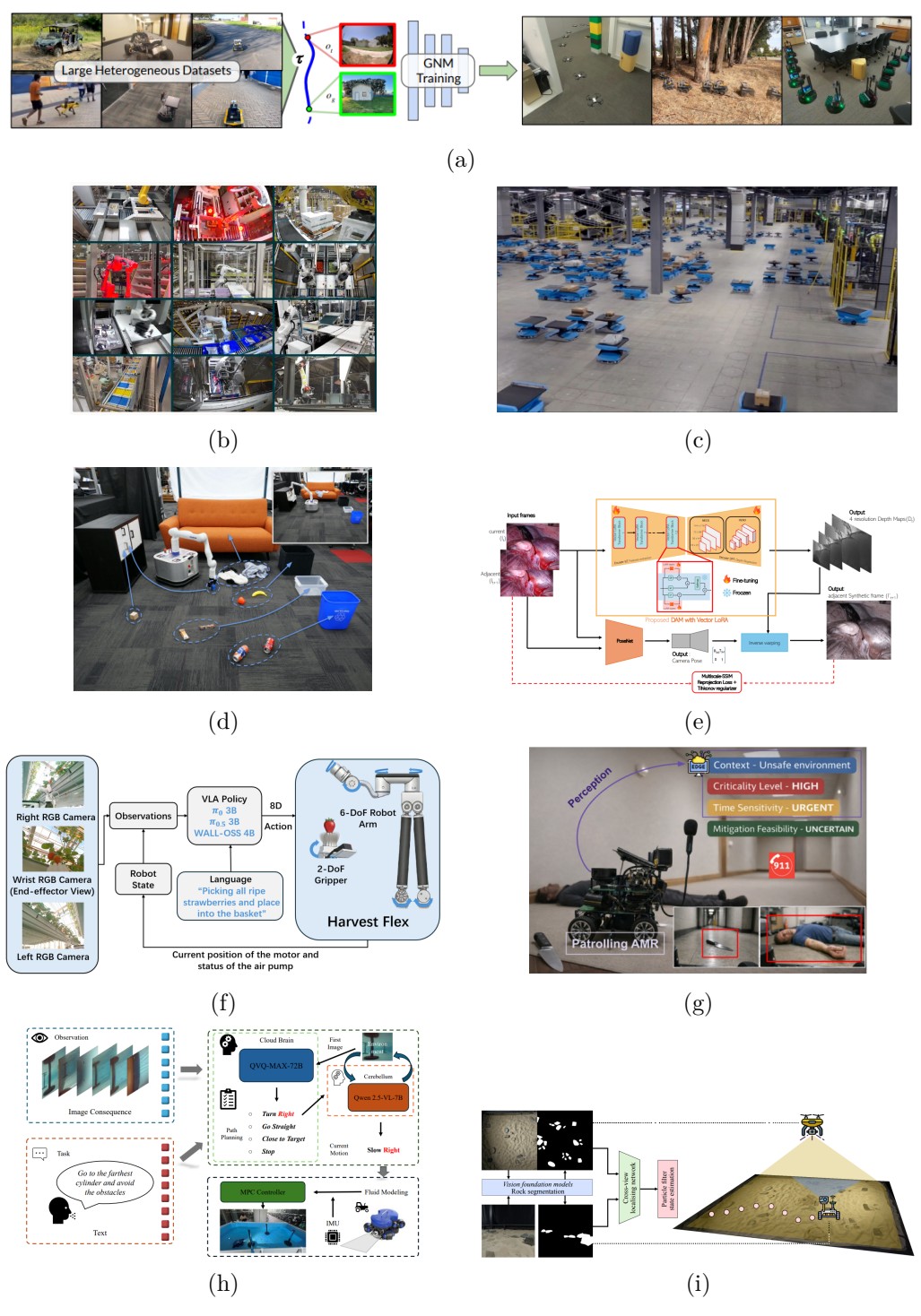

Figure 6: Representative literature methods per application domain: (a) Agentic mobility (GNM (Shah et al., 2023b)), (b) Industrial manipulation (RFM-1 (Sohn et al., 2024)), (c) Supply operations (DeepFleet (Agaskar et al., 2025)), (d) Service robots (TidyBot (Wu et al., 2023a)), (e) Medical robots (DARES (Zeinoddin et al., 2024)), (f) Cognitive agrisystems (HarvestFlex (Zhao et al., 2026b)), (g) Crisis agents (See Something, Say Something (Cancelli et al., 2026)), (h) Maritime robotics (UnderwaterVLA (Wang et al., 2025f)), and (i) Space robotics (Planetary cross-view localisation (Holden et al., 2026)).

## 10.3 Supply operations

FMs are drastically evolving autonomous supply operations, by providing robots with the flexibility and intelligence needed to handle unstructured, dynamic, and complex logistics tasks, adapting to new items, facility layouts, and operational disruptions without manual reprogramming. At a broader scale, logistics operations utilize FMs for sustainable planning, multi-agent coordination, and end-to-end decision support, targeting 'Logistics 5.0' goals, such as greener routing (Nicoletti & Appolloni, 2024). By integrating multi-agent systems with foundation backbones, autonomous supply chains can link demand forecasting, resource allocation, and transport routing through shared representations and agentic reasoning (Xu et al., 2024a; Nicoletti, 2025).

## 10.4 Service robots

The integration of FMs represents a major paradigm shift for service robots, transforming them into adaptable agents capable of navigating unstructured domestic environments and interpreting natural language instructions. TidyBot (Wu et al., 2023a) uses few-shot LLM reasoning to learn personalized tidying preferences directly from user conversations, grounding abstract 'put-away' commands into specific physical actions across novel scenes. Extending these capabilities to multi-step procedures, ELLMER (Mon-Williams et al., 2025) performs tasks, such as coffee brewing, by integrating visual, force, and linguistic feedback with a retrieval-augmented memory mechanism that adapts its plans on the fly under uncertainty.

## 10.5 Medical robots

FMs in the medical domain serve as a critical bridge between high-level clinical knowledge and low-level robotic execution, increasingly integrated into surgical perception and decision-making pipelines for high-stake, high-variability tasks. A key application is monocular depth estimation: Surgical-DINO (Cui et al., 2024) adapts DINOv2 using a LoRA adapter for endoscopic imagery, while DARES (Zeinoddin et al., 2024) tailors a Depth Anything Model using self-supervised Vector-LoRA to align with surgical scene statistics. Additionally, EndoDDC (Lin et al., 2026) addresses sparse-to-dense depth reconstruction for endoscopic robotic navigation through diffusion-based depth completion, supporting reliable 3D reconstruction and safe instrument guidance.

## 10.6 Cognitive agrisystems

The integration of FMs into cognitive agrisystems marks a transition towards adaptive, embodied intelligence in field operations, where VLA architectures ingest high-level semantic instructions and translate them into precise motor outputs in real-time. Unlike traditional agricultural robots that rely on hard-coded rules, FMs bridge the semantic gap by reasoning through the variability of biological environments, such as shifting light, overlapping foliage, and irregular crop shapes (Yin et al., 2025b). For instance, HarvestFlex (Zhao et al., 2026b) employs VLA policies for real greenhouse strawberry harvesting, a long-horizon task challenged by occlusion and specular reflections, while FM-based reasoning also supports task planning in crop monitoring and field management (Cuaran et al., 2026).

## 10.7 Crisis agents

FMs are fundamentally reforming disaster response and public safety, by enabling crisis agents to transition from rigid, remote-controlled setups to autonomous systems, capable of high-level reasoning in unpredictable and hazardous environments. In particular, SafeGuard ASF (Canh et al., 2026) combines multi-modal hazard perception with agentic reasoning for real-time fire-risk detection and disaster recovery. Additionally, a robotic fire-risk detection system based on dynamic knowledge graphs and LLM-enhanced multi-modal reasoning is presented in (Pan et al., 2025a), demonstrating how FM-based reasoning can support emergency response in safety-critical settings.

## 10.8   Maritime robotics

The incorporation of FM intelligence has significantly bolstered the capabilities of autonomous maritime systems, allowing them to perceive, to reason, and to act more effectively within complex aquatic environments. These models are specifically engineered to navigate typical underwater challenges, such as high turbidity, limited visibility, and severe communication constraints that often degrade traditional robotic sensors. In particular, UnderwaterVLA (Wang et al., 2025f) introduces a dual-brain VLA architecture for autonomous underwater navigation, combining multi-modal reasoning with embodied control for improving robustness under degraded visual and communication conditions. Additionally, MarineInst (Zheng et al., 2024b) and MarineGPT (Zheng et al., 2023) demonstrate FM capabilities in bridging raw marine visual data, semantic understanding, and domain-specific natural-language knowledge, thereby supporting richer perception and reasoning modules for maritime robotic platforms.

## 10.9   Space robotics

The integration of FMs is critically transforming the field of astro-embodied intelligence, enabling robotic agents to reason, to adapt, and to perceive within unstructured, off-world environments, where human intervention is physically impossible. These models provide strong priors and zero-shot generalization capabilities that are crucial for operating under the extreme conditions and data scarcity typical of planetary missions. A primary application involves the usage of SAM for universal crater detection (Giannakis et al., 2024), which utilizes promptable segmentation to identify features across diverse planetary imagery without requiring domain-specific retraining. Beyond basic detection, FMs are being extended to facilitate autonomous terrain understanding and complex geological analysis (Giannakis et al., 2024; Zhao & Ye, 2024; Holden et al., 2026), allowing robots to make high-stake decisions independently in remote and hazardous space settings.

# 11   Public datasets

Having systematically investigated the robotic FM literature using different criteria (Sections 4-10), this section outlines the main public datasets that have been introduced so far for developing and evaluating robotic FM methods. Rather than reproducing an exhaustive catalog, the discussion below is organized around the main dataset families that recur in the robotic FM literature; for each family, its scope/definition, its typical uses, a set of representative resources, and its main current gaps are summarized.

- Large-scale cross-embodiment trajectory corpora: These comprise massive collections of real-world robot interaction trajectories aggregated across many robotic platforms, tasks, and environments, typically pairing synchronized RGB-D observations, natural-language instructions, and low-level action tokens. Their main use lies in pre-training generalist VLA policies that transfer robustly across different kinematic structures and operating conditions, effectively serving as the backbone training data for cross-embodiment models. Representative resources include Open X-Embodiment (O'Neill et al., 2024), AgiBot World (Bu et al., 2025), DROID (Khazatsky et al., 2024), BridgeData V2 (Walke et al., 2023), and RoboMIND (Wu et al., 2025a). Their main current gaps comprise the scarcity of real-world physical interaction signals (particularly tactile and force/torque sensing), the near-absence of failure and recovery recordings, and a strong skew towards table-top manipulation settings.

- Simulation environments and benchmark suites: These comprise photo-realistic or game-engine simulators and curated task suites that algorithmically generate environments, tasks, and ground-truth supervision for embodied agents. Their main usefulness lies in enabling reproducible, low-cost, large-scale training and standardized evaluation, while also supporting long-horizon, language-conditioned policy learning and the systematic study of knowledge transfer across tasks. Representative resources include AI2-THOR (Kolve et al., 2017), RLBench (James et al., 2020), CALVIN (Mees et al., 2022), BEHAVIOR-1K (Li et al., 2023a), and LIBERO (Liu et al., 2023a). Their main current limitations comprise the persistent sim-to-real gap, the limited fidelity in modeling contact-rich physics, and a narrow object and scene diversity relative to the real world.

- Vision and language navigation datasets: These comprise benchmarks that pair natural language instructions or goals with embodied navigation episodes in 3D environments. Their main use lies in training and evaluating language-guided navigation, open-vocabulary goal finding, and fine-grained instruction following. Representative resources include REVERIE (Qi et al., 2020), RxR (Ku et al., 2020), VLN-CE (Krantz et al., 2020), ScaleVLN (Wang et al., 2023), and GOAT-Bench (Khanna et al., 2024). Their main current gaps comprise a heavy reliance on static photorealistic scans, the scarcity of dynamic, human-populated, and outdoor settings, and limited modeling of social/interaction constraints.

- Real-world perception and scene-understanding datasets: These comprise large, richly annotated real-world sensor collections (RGB-D, LiDAR, and multi-camera) targeting 3D indoor reconstruction and outdoor autonomous-driving perception. Their main use lies in pre-training and benchmarking visual and 3D perception backbones (e.g., semantic and instance segmentation, detection, and mapping) that subsequently feed downstream robotic policies. Representative resources include nuScenes (Caesar et al., 2020), Waymo Open (Sun et al., 2020), Semantic-KITTI (Behley et al., 2019), ScanNet++ (Yeshwanth et al., 2023), and Matterport3D (Chang et al., 2017). Their main current gaps comprise their predominantly passive nature (lacking embodiment and action labels), a domain concentration on driving and indoor scenes, and the limited presence of contact or manipulation context.

- Egocentric human-video corpora: These comprise large-scale first-person (egocentric) and closely related fixed-view videos of humans performing everyday activities, capturing natural manipulation, interaction, and navigation behavior. Their main use lies in providing web-scale human priors for representation pre-training, affordance and activity understanding, and learning from human demonstration to bootstrap robot policies. Representative resources include Ego4D (Grauman et al., 2022), EPIC-KITCHENS (Damen et al., 2018), Ego-Exo4D (Grauman et al., 2024), and COM Kitchens (Maeda et al., 2024). Their main current gaps comprise the still comparatively small number of corpora available at this scale, the absence of explicit robot action/control labels, and the embodiment gap between human and robot morphologies that complicates direct transfer.

- Human-robot interaction and dialogue datasets: These comprise datasets that capture multimodal human-robot communication, including dialogue, clarification questions, gestures, and speech grounded in embodied tasks. Their main use lies in training and evaluating conversational grounding, interactive task completion, clarification-seeking behavior, and reasoning over interleaved vision-language-action streams. Representative resources include RoboVQA (Sermanet et al., 2024), TEACh (Padmakumar et al., 2022), CVDN (Thomason et al., 2020), DialFRED (Gao et al., 2022), and NatSGD (Shrestha et al., 2025). Their main current gaps comprise their limited scale, the largely simulated or scripted nature of the recorded dialogues, and the scarcity of real-world, synchronized multimodal (e.g., speech plus gesture) interaction data.

The full details of the public datasets discussed in this section are provided in Section I of the supplementary document. In particular, the latter contains the complete dataset catalog, organized as detailed per-task tables (grouping the various benchmarks by main robotic task and reporting, for each entry, its year, scale, semantic classes, modalities, annotation type, domain, environment type, temporality, embodiment, and a short description), as well as the extensive list of global observations regarding the main current trends across robotic FM datasets.

## 12 Current challenges

Despite the large body of works that have recently been introduced in the field of robotic FM-based methods and the tremendous advancements accomplished, significant challenges and open research problems still remain, which if robustly addressed will further increase the efficiency, reliability, acceptance, and adoption of such solutions in real-world deployment scenarios. In the remaining of this section, the main challenges identified in the literature are systematically examined and outlined.

## 12.1 Data aspects

Unlike benchmark requirements and availability in fields like NLP and CV, robotic data is physically-grounded, high-dimensional, multi-modal, and typically expensive to collect. In this respect, the following specific challenges related to data aspects in robotic FM research are present:

- Scarcity of physical-world data (C1.1): Despite the availability of internet-scale visual/text benchmarks, robotic solution development lacks comparable datasets of high-quality, diverse, real-world robot trajectories (Brohan et al., 2023a). The main factor for the latter comprises the data collection cost, which requires physical hardware, human teleoperation/demonstration/instruction, and significant time. Additionally, the added difficulty of collecting 'long-tail scenarios' (i.e., rare and unexpected events with huge impact on the system performance) makes the learning of safe recovery behaviors even more challenging (Park et al., 2025).

- Embodiment heterogeneity (C1.2): Robotic data is generally not uniform, since it is produced by different robots with diverse hardware and physical configurations (e.g., degrees of freedom, sensor suites, kinematics, etc.) (O'Neill et al., 2024). Additionally, there is no single format for continuous action data across diverse embodiments, without losing physical meaning/properties (Reed et al., 2022). The above result into an inherent difficulty to transfer knowledge between different robotic platforms (Liu et al., 2025).

- Maintaining high data quality (C1.3): Increasing the data scale usually leads to improved performance, conditioned on the fact of maintaining a high-quality in the captured data (Zitkovich et al., 2023). However, human-captured teleoperation data often includes suboptimal movements, hesitation, or outright failures (Hu et al., 2023). Additionally, manual inspection of immense amounts of expert demonstration videos is tremendously expensive.

- Domain transfer gap (C1.4): Typical approaches to overcome the data scarcity problem is to make use of simulation or data from another (similar) domain. However, simulation engines often fail to accurately reproduce complex physics (e.g., friction, object deformations, etc.), leading to low performance in real-world settings (Xiao et al., 2024). On the other hand, incorporation of data from multiple, diverse environments is shown to have a more crucial impact on training, than simply scaling the available datasets (O'Neill et al., 2024; Zitkovich et al., 2023).

- Multi-modal and temporal alignment (C1.5): Robots need to efficiently integrate vision, language, and proprioception streams to accomplish robust performance in real-world settings. However, the high and different frequency of the various information sequences, along with the inherent difficulty in mapping continuous physical-world actions to discrete information tokens, results into precision degradation (e.g., in dexterous tasks) (Zhou et al., 2025d).

## 12.2 Computation aspects

In any real-world robotic application case, time performance and resource requirements constitute a cornerstone regarding fundamental safety and stability specifications. In this respect, the following specific challenges related to computation aspects in robotic FM research are present:

- Need for real-time inference (C2.1): Robot control loops typically operate at high frequencies (e.g., often more than 50Hz), in order to achieve stability and safe execution (Chi et al., 2025). However, the inherently extreme scale of FMs naturally introduces significant inference latencies, ranging from some hundreds of milliseconds up to multiple seconds (Firoozi et al., 2025; Zitkovich et al., 2023). This is highly likely to result into FM-based solutions not to be applicable in multiple cases or to lead to inaccurate policy executions (Ameperosa et al., 2025).

- Need for safety bound interval (C2.2): Apart from the inference time itself, safe robot operation requires the application of certain safety routines or the execution of corrective actions (Sinha et al., 2024). The latter impose additional time constraints during execution, i.e. a strict upper bound for overall end-to-end latency (Firoozi et al., 2025; Sinha et al., 2024).

- Constrained onboard resources (C2.3): Robotic platforms typically pose specific and strict size, weight, and energy specifications, setting particular limitations to the onboard embedded GPUs and their operation. Contrary to FM execution on cloud, resource-abundant environments, edge devices exhibit limitations in terms of available memory, processing time, and energy/thermal tolerance, prohibiting the deployment of full-scale and best-performing versions of robotic FMs (O'Neill et al., 2024; Yue et al., 2024).

## 12.3 Safety and security aspects

The fundamental advantage of FMs in robotic applications relies on their ability to combine high-level semantic reasoning with low-level, physical planning/execution procedures. The latter though raise particular safety and security concerns, which extend beyond conventional/traditional robotic settings. In this respect, the following specific challenges related to safety and security aspects in robotic FM research are present:

- Semantic-physical space mismatch (C3.1): It is very common for FMs to generate policies that are linguistically, logically, and semantically sound, but can likely lead to dangerous or even physically impossible scenarios (with the extreme case being that of the presence of hallucinations) (Lin et al., 2025b; Yin et al., 2025a). This is mainly due to the typical limitation of most FMs to account for physical-world parameters (e.g., friction, torque limits, material properties, etc.) in their reasoning process (Firoozi et al., 2025). To make matters worse, FMs typically lack the ability to estimate the consequences of their (erroneous) actions in the physical environment (Zitkovich et al., 2023; Lin et al., 2025b).

- Adversarial vulnerabilities (C3.2): As the number of modality streams and data scale that a single FM can process rises, the respective (cyber) attack surface of the (often cloud-hosted) model itself increases proportionally (Radanliev et al., 2026). In particular, even relatively minor perturbations in the data can result into significant behavior deviations or incorrect/hazardous actions (Wang et al., 2025d).

- Inaccurate uncertainty quantification (C3.3): Especially in human-robot collaboration settings, the robotic agent needs to constantly maintain a precise estimation of its own state/policy uncertainty (Wang et al., 2025e). Current FMs though do not reliably balance their reactions with respect to aleatoric (environmental noise) and epistemic (model ignorance) uncertainty (Marques & Berenson, 2024). This bottleneck becomes even more evident in onboard deployment settings, where increased inference latency or compressed models are used (due to resource constraints) (Zitkovich et al., 2023).

## 12.4 Embodiment aspects

FMs have equipped robots with unprecedented perception, reasoning, and execution capabilities under real-world, dynamic environments. However, the critical issue that arises comprises that of grounding robot tasks on a physical platform, which poses specific sensor, actuator, and hardware constraints. In this respect, the following specific challenges related to embodiment aspects in robotic FM research are present:

- Heterogeneity of robot action spaces (C4.1): Unlike the case of other AI deployment scenarios, robots exhibit a vast variety in terms of physical forms, morphologies, and capabilities. The latter renders difficult to deploy a model trained on a specific hardware setup (e.g., dual-arm, mobile platform, etc.) to another (e.g., quadruped, drone, etc.), due to difference in key robotic specifications (e.g., varying degrees-of-freedom) (O'Neill et al., 2024; Mees et al., 2024). Additionally, the absence of a universally applicable, standardized control interface across different robots, makes the development of robust, generalized, platform-independent FMs particularly difficult (Zheng et al., 2025b).

- Sim-to-real gap (C4.2): In an attempt to alleviate from the need for large-scale, high-quality robotic interaction data, simulation engines are often used for data generation. However, employing simulation environments does not always result into the assembly of benchmarks with sufficient diversity in both task execution and environment settings (Jonnarth et al., 2025). Moreover, simulation engines are

typically prone to not model accurately complex physical-world dynamics (e.g., friction, deformation, fluid interactions, etc.), which in turn results into reduced robot performance (Makoviychuk et al., 2021; Ai et al., 2025).

- Limited physical space grounding (C4.3): Despite the unprecedented semantic capabilities of FMs, the mapping of such high-level, semantic representations to low-level, real-world physics is not guaranteed. In particular, current FMs exhibit limitations in precise spatial, geometric, and physical interactions (Qi et al., 2025). Additionally, the lack of sufficient contextual, common-sense knowledge in FMs may result into the generation of failure modes in the real-world (Chen et al., 2024a; Huang et al., 2023c).

- Constrained haptic capabilities (C4.4): The so-called 'final frontier' problem of physical intelligence comprises the difficulty of robots to replicate nuanced sensorial feedback, like humans do in the real world (Yang et al., 2024). The latter constitutes inherently a multi-faceted problem, including perspectives related to the scarcity of large-scale haptic/tactile data, robot hardware heterogeneity, sim-to-real variance, and high-frequency temporal reaction demands (Zhang et al., 2025b).

## 12.5 Reasoning aspects

While inference capabilities of FMs in other AI domains (e.g., CV, NLP, etc.) has successfully achieved the incorporation of rich, abstract logic in most cases, their application to physically-grounded, robot agents exhibits significant obstacles. In this respect, the following specific challenges related to reasoning aspects in robotic FM research are present:

- Lack of physical common sense knowledge (C5.1): The usual embodiment-agnostic understanding of the world by FMs (e.g., gravity, object permanence, material properties, etc.) makes their real-world deployment difficult (Firoozi et al., 2025; Kawaharazuka et al., 2024). Additionally, the lack of sufficient causal reasoning capabilities hinders robots to predict the physical consequences of their actions (Töberg et al., 2024).

- Constrained long-horizon planning (C5.2): The reasoning performance of robotic FMs tends to degrade exponentially, as the number of required steps increases (Driess et al., 2023). The latter has a great impact, especially in cases of large-scale operational environments (Lisondra et al., 2026). Moreover, the training process itself poses difficulties to FMs to connect long-term goals to specific sequences of decision steps (Brohan et al., 2023b).

- Imprecise explanations of robot behaviors (C5.3): FMs in robotics comprise by nature massive architectures that integrate multi-modal information streams, while connecting high-level reasoning with low-level, physical control (Brohan et al., 2023a). In contrast to traditional modular robotic solutions, where failure modes can be traced to specific components, FMs implement end-to-end generalist policies that make it particularly difficult to specify the factors that have led to the execution of a specific physical action (Kawaharazuka et al., 2024).

## 12.6 Evaluation aspects

Provided that FMs bridge the entire gap between high-level, semantic reasoning and low-level, physical execution in dynamic environments, the assessment of their performance and robustness exhibits particular characteristics, compared to other conventional AI solutions. In this respect, the following specific challenges related to evaluation aspects in robotic FM research are present:

- Lack of a unified evaluation framework (C6.1): Despite the availability of a series of standardized, task-oriented, intuitive performance metrics, no holistic and integrated evaluation protocol is currently present to assess the multi-factored robotic failure cases (Firoozi et al., 2025). In particular, the available (and typically binary) metrics are often proven to be coarse, failing to clearly highlight the underlying factors leading to low performance (e.g., inefficient bi-manual coordination, asymmetric

arm usage, etc.) (Jiang et al., 2023; O'Neill et al., 2024). Additionally, the development of independent, domain-specific metrics across different fields makes even more difficult to assess robot performance with respect to different architectures or motion parameters (Brohan et al., 2023a).

- Imprecise generalization ability assessment (C6.2): Despite the key driving force of FMs to exhibit increased (especially zero-shot) generalization ability, its comprehensive, accurate, and robust assessment is especially difficult (Zitkovich et al., 2023; Liang et al., 2023). In particular, the introduction of even minimal distribution shifts in task descriptions or observation domains can have a great impact on the resulting robot behavior (or even failure) (Kube et al., 2026).

## 13 Future research directions

The introduction of FMs in the field of robotics has led to unprecedented accomplishments and advances in all core robotic technologies, as discussed in Sections 3-10. Despite these tremendous developments, several open challenges are still present, which pose restrictions in the wider deployment of robotic solutions in real-world scenarios, as outlined in Section 12. In this respect, this section discusses the main and most promising future research directions towards achieving the goal of developing efficient, robust, and general-purpose robotic agents, in correspondence to the challenges described in Section 12.

### 13.1 Architectural evolution

VLAs comprise one of the most promising types of NN architectures, which implement a unified neural function for mapping visual observations, linguistic instructions, and proprioceptive states directly to low-level, robot control policies. Their core characteristic is their end-to-end integration nature, which enables robotic agents to perceive/reason about their surrounding environment and, subsequently, to react upon the input stimuli in a single forward/inference pass. In this respect, the following promising research directions emerge:

- Heterogeneous action spaces: An ambitious goal of current research efforts focuses on the development of universal robotic FMs, capable of operating across different robotic platforms. This requires the robust addressing of the action heterogeneity (or cross-embodiment generalization, stated alternatively) problem (O'Neill et al., 2024). In this context, future research could emphasize on the development of embodiment-agnostic action spaces, which would enable FMs to predict desired end-effector trajectories/forces, while their eventual mapping to particular, platform-specific, low-level operations could be controlled by a hardware-specific modulation component (Mees et al., 2024). This direction mainly aims to address the challenges of 'Embodiment heterogeneity' (C1.2) and 'Heterogeneity of robot action spaces' (C4.1), as outlined in Section 12.

- Sophisticated action sequence tokenization: One of the main challenges in the design of VLA transformer architectures comprises the discrete, tokenized representation of continuous space robotic actions. The conventional approach of binning each dimension to a fixed number of values inherently leads to loss of precision, especially for dexterous tasks (Zitkovich et al., 2023; Reed et al., 2022). In this context, future research could emphasize on more sophisticated tokenization methods capable of capturing the detailed dynamics of continuous actions, while maintaining the efficiency of autoregressive decoding. This direction mainly aims to address the challenge of 'Multi-modal and temporal alignment' (C1.5) (Section 12), particularly the precision loss incurred when mapping continuous physical actions to discrete tokens.

- Diffusion and flow-based action modeling: When a robot attempts to perform an action, there is a set of practically infinite trajectories to select from. Common architectures (e.g. transformers) tend to learn the average of such possible motions, leading to performance degradation or failure (Florence et al., 2022). On the contrary, flow- and diffusion-based objectives facilitate the modeling of temporal dynamics in a continuous latent space. In this context, future research could emphasize on the latter aspects, targeting to equip the robots with the ability to predict the consequences of their actions in a continuous space representation (Chi et al., 2025). This direction mainly aims to address the challenges of 'Multi-modal and temporal alignment' (C1.5) and 'Lack of physical common sense

knowledge' (C5.1) (Section 12), by enabling robots to anticipate the physical consequences of their candidate actions.

## 13.2 Multi-modal embodied intelligence

The main driving-force of current FMs lies on the incorporation of vision and language information, while true embodied intelligence requires a holistic understanding of the physical world through the integration of touch, sound, and force sensing. The latter is essential especially for tasks requiring dexterity and delicate interaction. In this respect, the following promising research directions emerge:

- Tactile information: Vision is often insufficient for performing contact-rich tasks and reaching human-level dexterity capability, which relies on incorporating feedback regarding the surface texture, slip, and force (Yu et al., 2024). For achieving the latter, the robust integration of tactile information is essential, capitalizing on the current/early-works on developing tactile FMs. In this context, future research could emphasize on the latter aspects, aiming at enhancing robots to efficiently manipulate deformable objects (Wu et al., 2020). This direction mainly aims to address the challenges of 'Constrained haptic capabilities' (C4.4) and 'Scarcity of physical-world data' (C1.1) (Section 12), the latter regarding the under-representation of large-scale tactile data.

- Proprioception and force control: Beyond tactile sensing, the incorporation of proprioception (i.e., the robot's sense of its own body state) and force control is crucial for realizing contact-rich manipulaton tasks. Current solutions usually focus only on position control, while exploratory works, like the integration of joint torque sensors, emerges as a promising approach (Kumar et al., 2021). In this context, future research could emphasize on further exploiting dense proprioception and force technologies, in order to boost more fine-grained manipulation and efficient human-robot collaboration (Liu et al., 2023b). This direction mainly aims to address the challenges of 'Limited physical space grounding' (C4.3) and 'Constrained haptic capabilities' (C4.4), as outlined in Section 12.

- Auditory feedback: The sound signal can often bear significant information regarding the occurrence of critical events during robot execution, complementing the corresponding visual and force feedback streams (Dimiccoli et al., 2022). Additionally, audio processing requires significantly less resources than the respective visual stream. In this context, future research could emphasize on further investigating the exploitation of auditory feedback, enabling robots to operate more accurately and reliably in dynamic and noisy environments (Mejia et al., 2024). This direction mainly aims to address the challenge of 'Multi-modal and temporal alignment' (C1.5) (Section 12), by enriching the set of complementary sensory streams that robots can integrate.

## 13.3 Reasoning and long-horizon autonomy

Current VLA-based solutions have demonstrated reliable performance on immediate reactive tasks. However, the implementation of multiple real-world activities requires the robust handling of complex, long-horizon, multi-stage missions. In this respect, the following promising research directions emerge:

- Long-horizon memory frameworks: Due to constraints in the memory capacity of current FMs, robots often tend to repeat unsuccessful policies, due to the relatively limited context window that they maintain and which might not include information about the previous policy failure (Firoozi et al., 2025). To this end, extending the memory capabilities of robotic solutions would have a great impact on long-horizon goal achievement. In this context, future research could emphasize on enhancing long-horizon autonomy, by incorporating structured summaries of past interactions (instead of full-sequence replays) (Wu et al., 2026), latent-graph memory formalisms to maintain experiences over longer contexts (Chen et al., 2026b), and stage-aware reward signals for more efficient policy learning in multi-step setups (Chen et al., 2026a). This direction mainly aims to address the challenge of 'Constrained long-horizon planning' (C5.2), as outlined in Section 12.

- Hierarchical semantic representations: In case of robot operation in cluttered environments, the usage of hierarchical abstractions (e.g., scene graphs) of the processed semantic information is proven to be

efficient in simplifying and strengthening decision-making (Gu et al., 2024). In this context, future research could emphasize on realizing task planning and reasoning on such hierarchical abstractions, so as to both reduce computational cost (i.e., avoid intense computations at the pixel or joint angle level) and to achieve more robust decision making (Gao et al., 2025a; Hughes et al., 2022). This direction mainly aims to address the challenges of 'Constrained long-horizon planning' (C5.2), 'Need for real-time inference' (C2.1), and 'Constrained onboard resources' (C2.3), as outlined in Section 12.

### 13.4 World foundation models

The deployment of FMs in robotic applications requires high-quality, diverse, and large-scale embodied interaction data. Given the fact that the collection of the latter datasets is typically expensive and time-consuming, world foundation models are being investigated as an alternative solution for trajectory generation. In this respect, the following promising research directions emerge:

- Physics-informed generative models: WMs have already been proven to generate photorealistic data of high quality. However, there still exists a gap between simulated physics and real-world dynamics, which would enable robots to learn and to act more reliably in actual operational settings (Ai et al., 2025; Lee et al., 2025a). In this context, future research could emphasize on explicitly incorporating constraints like gravity, friction, and fluid dynamics into the video generation process (Xie et al., 2024). Another closely related aspect concerns the integration of sophisticated physics-grounded predictive pipelines, which would serve as value functions for model-based planning, especially in long-horizon tasks (Zhou et al., 2024a; Yang et al., 2023). This direction mainly aims to address the challenges of 'Scarcity of physical-world data' (C1.1), 'Domain transfer gap' (C1.4), 'Sim-to-real gap' (C4.2), and 'Lack of physical common sense knowledge' (C5.1), as outlined in Section 12.

- Action-conditioned scenario generation: Research in world models regarding action-conditioned scenario generation has advanced from plain video prediction towards the creation of unified models, which function as physically grounded, interactive cognitive engines (Bruce et al., 2024). In particular, significant efforts are devoted on integrating differentiable physics and unified geometric representations for ensuring the spatio-temporal consistency and accuracy of the generated predictions over long horizons (Lee et al., 2025a; Tu et al., 2025). In this context, future research could emphasize on further extending current capabilities, especially focusing on cross-embodiment generalization and 'long-tail' scenarios, so as to boost the translation of human-centric video data into physically plausible robotic trajectories across multiple hardware platforms (O'Neill et al., 2024; Mees et al., 2024). This direction mainly aims to address the challenges of 'Scarcity of physical-world data' (C1.1), 'Embodiment heterogeneity' (C1.2), 'Maintaining high data quality' (C1.3), and 'Imprecise generalization ability assessment' (C6.2) (Section 12), since generated long-tail and cross-embodiment scenarios can also serve to systematically probe model generalization.

### 13.5 Safety and verification

Given the increasing deployment of robots in dynamic, unstructured environments, their interaction and collaboration with humans requires the robust addressing of safety risks. For achieving this, FM-based solutions need to incorporate a combined approach, consisting of both high-level semantic understanding and low-level, deterministic safety aspects. In this respect, the following promising research directions emerge:

- Adaptive safety: The incorporation of adaptive safety measures in robotic FMs is gradually shifting from a reactive post-processing approach towards an integrated one, where constraints are directly projected to the model's training and reasoning loops (Ren et al., 2023). This enables robots to predict and to mitigate physical risks through grounded reasoning, prior to action execution (Zha et al., 2024). In this context, future research could emphasize on further enhancing safety alignment approaches through constrained learning and process reward models for evaluating individual reasoning steps and environmental affordances in real-time (Yu et al., 2025a; Anand et al., 2026). This direction mainly aims to address the challenges of 'Semantic-physical space mismatch' (C3.1), 'Inaccurate uncertainty quantification' (C3.3), and 'Need for safety bound interval' (C2.2), as outlined in Section 12.

- Formal verification: Research in formal verification of robotic FMs is currently translating from conventional offline proofs towards modular, runtime-based assurance frameworks. In particular, current efforts largely focus on control barrier functions and reachability analysis, in order to intercept model outputs in real-time (Miyaoka & Inoue, 2025; Wang & Wen, 2025). In this context, future research could emphasize on investigating neuro-symbolic integration approaches for mapping neural outputs to formal logics (Cunnington et al., 2024) and automated specification mining pipelines that employ LLMs for translating natural language instructions into precise mathematical formulas (Rabiei et al., 2025). This direction mainly aims to address the challenges of 'Semantic-physical space mismatch' (C3.1), 'Adversarial vulnerabilities' (C3.2), 'Imprecise explanations of robot behaviors' (C5.3), and 'Lack of a unified evaluation framework' (C6.1) (Section 12), the latter through standardized, formally grounded assessment criteria.

## 14    Conclusion

**Summary**: The introduction of Foundation Models (FMs) has resulted into transformative effects in the field of robotics, transitioning the current practice from rigid, single-task solutions towards adaptive, multi-sensory, and generalist agents, capable of operating in complex, dynamic, open-world environments. The current review has provided a holistic, thorough, systematic, and in-depth analysis of the research landscape by delineating five distinct evolution phases, starting from early Natural Language Processing (NLP) and Computer Vision (CV) model integration to the current frontier of multi-sensory generalization and real-world deployment. Through a highly-granular, multi-criteria, taxonomic literature investigation, this work has analyzed the interplay between different foundation model types (LLMs, VFMs, VLMs, and VLAs), underlying neural network architectures, adopted learning paradigms, learning stages for skill acquisition, robotic tasks (perception, planning, navigation, manipulation, and human-robot interaction), and real-world application domains. The above discussion has been accompanied by a methodical comparative analysis of the various categories of approaches and critical insights per defined criterion. Moreover, an overview of the publicly available datasets used for model training and evaluation was provided, organized around the main recurring dataset families and their respective scope, typical uses, representative resources, and current gaps. Furthermore, a detailed and hierarchical discussion on the current open challenges and promising future research directions in the field was incorporated.

**Concluding synthesis**: Whereas Section 12 systematically enumerates the open challenges (C1.1-C6.2) and Section 13 proposes corresponding research directions, the purpose of this concluding discussion is to connect the two more in-depth, by elaborating on the most critical open problems and explicitly relating them to the body of works reviewed throughout this survey. Specifically, for each problem family (Sections 12.1-12.6), a distinction is made between aspects that existing works already address (at least partially) and those that remain genuinely unsolved, in order to highlight where the field has made tangible progress and where future effort is needed the most.

Open problems in data: Although internet-scale pre-training endows robotic FMs with unprecedented perceptual and reasoning priors, the field still lacks corpora of high-quality, diverse, real-world robot trajectories comparable to those available in NLP and CV (C1.1-C1.5). Existing works partially mitigate this gap through large-scale cross-embodiment aggregation (O'Neill et al., 2024; Mees et al., 2024; Liu et al., 2025) and, more recently, through world models that synthesize interaction data and serve as model-based planners (Zhou et al., 2024a; Yang et al., 2023; Bruce et al., 2024). Nevertheless, the systematic generation of safety-critical 'long-tail' scenarios (Park et al., 2025) and the curation of high-quality demonstrations from suboptimal teleoperation data (Hu et al., 2023) remain largely open, keeping data the most fundamental bottleneck of the field.

Open problems in computation: A second cluster of unresolved problems concerns the balance between the scale of FMs and the strict real-time, resource, and safety-bound constraints of robotic platforms (C2.1-C2.3). Robot control loops typically require frequencies above 50Hz (Chi et al., 2025), whereas FM inference latencies range from hundreds of milliseconds to several seconds (Firoozi et al., 2025; Zitkovich et al., 2023), and onboard edge devices impose tight memory, energy, and thermal limits (Yue et al., 2024). Despite progress

in model compression and efficient inference, the deployment of full-scale, best-performing FMs under hard latency and onboard-resource constraints remains, to a large extent, an open problem.

Open problems in safety and security: The defining strength of robotic FMs, namely the coupling of high-level semantic reasoning with low-level physical execution, is also the source of their most pressing safety and security concerns (C3.1-C3.3). Existing efforts increasingly shift from reactive post-processing towards safety constraints embedded in the reasoning loop (Ren et al., 2023; Zha et al., 2024) and runtime assurance, via control barrier functions, reachability analysis, and neuro-symbolic verification (Miyaoka & Inoue, 2025; Wang & Wen, 2025; Cunnington et al., 2024). However, these approaches are still at early-stage and reliable uncertainty quantification, together with formal guarantees in high-stake human-robot interaction settings, remains an open challenge.

Open problems in embodiment: A further set of open problems stems from the difficulty of grounding generalist policies on heterogeneous physical platforms (C4.1-C4.4). The heterogeneity of robot morphologies and action spaces is partially addressed by embodiment-agnostic action representations and hardware-specific modulation (O'Neill et al., 2024; Mees et al., 2024), while the replication of nuanced sensorial feedback, often referred to as the 'final frontier' of physical intelligence (Yang et al., 2024), is being approached through early tactile FMs (Yu et al., 2024; Zhang et al., 2025b). Nonetheless, the scarcity of large-scale haptic data, the absence of standardized control interfaces, and the persistent sim-to-real gap maintain cross-embodiment generalization far from solved.

Open problems in reasoning: The application of FM inference to physically-grounded agents exposes open problems in physical common sense and long-horizon autonomy (C5.1-C5.3). Existing works seek to extend reasoning over longer contexts through structured memory summaries, latent-graph memory formalisms, and stage-aware reward signals (Wu et al., 2026; Chen et al., 2026b;a), as well as through reasoning over hierarchical semantic abstractions, such as scene graphs (Gu et al., 2024). So far, the lack of causal, physically-grounded common-sense knowledge and the exponential degradation of reasoning performance with task horizon remain among the least understood and most impactful open problems.

Open problems in evaluation: Acting as an element affecting all the above, the field still lacks a unified protocol for assessing FM-based robotic systems (C6.1-C6.2). Current metrics are typically coarse and binary, failing to expose the underlying factors of multi-factored failure cases (Firoozi et al., 2025), while the reliable, comprehensive assessment of (especially zero-shot) generalization under distribution shift is particularly difficult (Kube et al., 2026). The absence of holistic, standardized evaluation frameworks therefore constitutes an under-addressed open problem that hampers fair comparison and reproducibility across the works reviewed in this survey.

**Unifying semantic-physical grounding gap**: It is critical to highlight that all the above problem clusters are indeed not independent, but rather facets of a single overarching open problem, namely the gap between high-level semantic reasoning and low-level physical grounding (most evidently manifested by challenges C3.1, C4.3, and C5.1) (Firoozi et al., 2025; Lin et al., 2025b). At the same time, these problems are also strongly interdependent: Synthesizing data through world models (C1.1) simultaneously bears on the sim-to-real gap (C4.2) and on physical common sense (C5.1), whereas hard latency constraints (C2.1) limit safety verification (C3.1-C3.3) levels that can be executed online. In terms of prioritization, latency reduction and unified evaluation appear comparatively tractable in the near term, whereas physical common-sense reasoning and human-level haptic intelligence constitute longer-horizon goals; robustly bridging the semantic-physical grounding gap, however, remains the central question on which the convergence of the field ultimately depends.

**Limitations of current survey**: In line with the comparative positioning summarized in Table 1 and the corresponding discussion in Section 1, the most significant limitation of this survey is that it provides a qualitative, taxonomic, and comparative analysis rather than a comprehensive quantitative benchmarking of the examined methods; reproducible, head-to-head empirical evaluation is deliberately left outside its scope and the reported insights rely on results as stated by the original works. Additionally, as also acknowledged in Table 1, the current review inevitably constitutes a snapshot of a rapidly evolving field, so relevant works will continue to appear after its cutoff. Beyond these main limitations, the survey covers partly certain connecting aspects, namely hardware design and low-level control theory, under-represented sensory modalities (such as

tactile and auditory feedback), and a number of more specialized topics (e.g., multi-robot/swarm systems and the legal, ethical, and societal dimensions of deployment).

**Scope for future surveys**: Building upon the above, the identified boundaries directly suggest concrete avenues for subsequent similar reviews. Initially, dedicated empirical/quantitative benchmarking surveys would complement the present qualitative analysis with reproducible, standardized, head-to-head comparisons, particularly for cross-embodiment generalization. Additionally, focused surveys on world foundation models for robotics could consolidate the rapidly growing literature on physics-informed and action-conditioned generation. Moreover, dedicated reviews on the safety and formal verification of embodied FMs would deepen the treatment of the safety-related challenges that this work necessarily covers at a higher level. Moreover, deeper/targeted domain-specific surveys (e.g., on surgical/medical, agricultural, space, or maritime robotics) and horizontal surveys on the under-covered topics noted above (i.e., hardware co-design, under-represented modalities, sustainability/energy efficiency, and the societal and trust dimensions of deployment) would further facilitate developments in the field.

**Outlook**: Overall, while FMs have already redefined what robotic agents can perceive, reason about, and accomplish, their transition from impressive demonstrations to dependable real-world deployment hinges on the robust addressing of the open problems outlined above. Towards this goal, by explicitly relating these problems to the works reviewed throughout this survey, the present analysis aims to offer a useful map for prioritizing future research towards more autonomous, reliable, and trustworthy robotic systems.

## Acknowledgments

This work has received funding from the European Union's Horizon Europe research and innovation programme under Grant Agreement No. 101189557 project TORNADO (foundaTion mOdels for Robots that haNdle smAll, soft and Deformable Objects) and No. 101168042 project TRIFFID (auTonomous Robotic aId For increasing First responders Efficiency). The views and opinions expressed in this paper are those of the authors only and do not necessarily reflect those of the European Union or the European Commission.

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
