# Foundation Models in Robotics: A Comprehensive Review of Methods, Models, Datasets, Challenges and Future Research Directions

## - Supplementary Material -

**Aggelos Psiris**                                                                 *aggelospsiris@hua.gr*
*Department of Informatics and Telematics, Harokopio University of Athens, Athens, Greece*

**Vasileios Argyriou**                                                   *Vasileios.Argyriou@kingston.ac.uk*
*Department of Networks and Digital Media, Kingston University, London, United Kingdom*

**Evangelos K. Markakis**                                                              *emarkakis@hmu.gr*
*Department of Electrical and Computer Engineering, Hellenic Mediterranean University, Heraklion, Greece*

**Panagiotis Sarigiannidis**                                                     *psarigiannidis@uowm.gr*
*Department of Electrical and Computer Engineering, University of Western Macedonia, Kozani, Greece*

**Efstratios Gavves**                                                                     *e.gavves@uva.nl*
*University of Amsterdam, Amsterdam, Netherlands*
*Archimedes, Athena Research Center, Athens, Greece*

**Kostas Bekris**                                                                     *kb572@cs.rutgers.edu*
*Computer Science Department, Rutgers University, New Brunswick, NJ, USA*

**Arash Ajoudani**                                                                  *Arash.Ajoudani@iit.it*
*Istituto Italiano di Tecnologia, Genova, Italy*

**Georgios Th. Papadopoulos**                                              *g.th.papadopoulos@hua.gr*
*Department of Informatics and Telematics, Harokopio University of Athens, Athens, Greece*
*Archimedes, Athena Research Center, Athens, Greece*

**Reviewed on OpenReview:** *https://openreview.net/forum?id=qF7vdPrpPk*

This supplementary document provides additional technical details and analyses, in order to complement and to support the main paper entitled "Foundation Models in Robotics: A Comprehensive Review of Methods, Models, Datasets, Challenges and Future Research Directions".

## A    Comparative analysis with recent surveys

This section provides the detailed comparison of the present survey with recent ones in the field, complementing the condensed summary presented in Table 1 of the main paper. In particular, for each surveyed work, Table 1 reports the following aspects: a) Survey scope, b) Review methodology, c) Primary contributions, and d) Main limitations.

Table 1: Detailed comparative analysis of recent surveys in the field of foundation models in robotics.

| Article | Scope | Methodology | Primary contributions | Limitations |
|---|---|---|---|---|
| Hu et al. (2023) (arXiv) | Broad survey and meta-analysis on the use of FMs towards general-purpose robots | • Literature analysis per robotic task
• Performance analysis per robotic task
• Analysis of 301 papers | • Separation between CV/NLP and native robotic FMs
• Identification of trends based on experimental results
• Discussion on open challenges and future research directions | • Discussion of early developments in robotic FMs
• No systematic and theoretical comparison between different approaches
• No discussion on recent developments (e.g., world models, diffusion policies, etc.)
• No discussion on application domains |
| Xu et al. (2024c) (arXiv) | Task-oriented survey focusing on robotic manipulation | • Literature categorization to high-level planning and low-level control approaches
• Analysis of 64 papers | • Discussion on form and assistant perspectives of planning
• Analysis of focused components in the learning process
• Discussion on open challenges and future research directions | • Investigation of only robotic manipulation approaches
• Limited literature coverage
• No systematic and theoretical comparison between different approaches
• No empirical/experimental benchmarking of the surveyed methods
• No discussion on application domains |
| Ma et al. (2024c) (arXiv) | VLA-oriented survey emphasizing on embodied AI aspects | • Taxonomic analysis of VLAs based on individual components, control policies, and high-level task planning
• Analysis of 785 papers | • Systematic comparison of methods
• Analysis of strengths and limitations per category
• Discussion on open challenges and future research directions | • No explicit discussion on other FM types (i.e., LLMs, VFMs, VLMs)
• No systematic analysis and comparisons across multiple criteria (i.e., NN architecture, learning paradigm, learning stage, robotic task)
• No empirical/experimental benchmarking of the surveyed methods
• No discussion on application domains |
| Jang et al. (2024) (IJCAS) | Application-oriented survey focusing on robotic autonomy | • Literature categorization based on perception, task planning, and control
• Analysis of environmental setups
• Analysis of 255 papers | • Discussion of impact of individual FM components on robot autonomy
• Analysis of robotic platforms and simulation environments
• Discussion on future research directions | • No systematic analysis across multiple criteria (i.e., NN architecture, learning paradigm, learning stage)
• No theoretical comparison between different approaches
• No empirical/experimental benchmarking of the surveyed methods
• No discussion on application domains |
| Kawaharazuka et al. (2024) (AR) | Application-oriented survey focusing on component replacement with FMs | • Literature categorization based on perception, planning, and data augmentation
• Analysis of 225 papers | • Analysis of input-output relationships in FMs
• Discussion on the role of FMs in perception, motion planning, and control
• Discussion on future research directions | • No systematic analysis across multiple criteria (i.e., FM type, NN architecture, learning paradigm, learning stage)
• No theoretical comparison between different approaches
• No empirical/experimental benchmarking of the surveyed methods
• No discussion on application domains |
| Firoozi et al. (2025) (IJRR) | Broad survey emphasizing on the use of FMs in robotics and embodied AI | • Literature analysis regarding decision-making, planning, and control
• Analysis of 233 papers | • Literature investigation regarding FMs that are native and relevant to robotics
• Analysis of embodied AI aspects
• Discussion on open challenges and future research directions | • No systematic analysis across multiple criteria (i.e., FM type, NN architecture, learning paradigm, learning stage)
• No theoretical comparison between different approaches
• No empirical/experimental benchmarking of the surveyed methods
• No discussion on application domains |
| Xiao et al. (2025b) (Neurocomputing) | Robot learning-oriented survey for generalist robots | • Analysis of robot learning in manipulation, navigation, task planning, and reasoning
• Analysis of 464 papers | • Systematic and hierarchical analysis of robot learning techniques
• Discussion on open challenges and future research directions | • No systematic analysis across multiple criteria (i.e., FM type, NN architecture, learning stage)
• No theoretical comparison between different approaches
• No empirical/experimental benchmarking of the surveyed methods
• No discussion on application domains |
| Tayyab Khan & Waheed (2025) (arXiv) | Integration-oriented survey focusing on real-world deployment | • Literature analysis regarding integrated, system-level strategies
• Feasibility analysis in real-world environments
• Analysis of 175 papers | • Literature categorization across simulation-driven design, open-world execution, sim-to-real transfer, and adaptable robotics
• Discussion on open challenges and future research directions | • No systematic analysis across multiple criteria (i.e., FM type, NN architecture, learning paradigm, learning stage, robotic task)
• No theoretical comparison between different approaches
• No empirical/experimental benchmarking of the surveyed methods
• No discussion on application domains |
| Kawaharazuka et al. (2025) (IEEE Access) | VLA-oriented survey emphasizing on full-stack aspects | • VLA literature analysis regarding both software and hardware perspectives
• Analysis of 427 papers | • Systematic analysis of architectural designs and learning paradigms
• Discussion on practical implementation aspects
• Discussion on future research directions | • No explicit discussion on other FM types (i.e., LLMs, VFMs, VLMs)
• Coarse categorization of all learning paradigms and robotic tasks
• No theoretical comparison between different approaches
• No empirical/experimental benchmarking of the surveyed methods
• No discussion on application domains |
| Sapkota et al. (2025) (arXiv) | VLA-oriented survey detailing research evolution aspects | • Literature analysis covering FM evolution, key progress areas, and application domains
• Analysis of 299 papers | • Analysis of research evolution
• Discussion on architectural innovations, training strategies, and real-time inference
• Analysis of application domains
• Discussion on open challenges and future research directions | • No explicit discussion on other FM types (i.e., LLMs, VFMs, VLMs)
• No systematic analysis across multiple criteria (i.e., NN architecture, learning paradigm, learning stage)
• No theoretical comparison between different approaches
• No empirical/experimental benchmarking of the surveyed methods |
| **Current survey** | Thorough and systematic analysis of the landscape of FMs in robotics | • Structured and systematic literature analysis, querying 6 major databases, using specific inclusion/exclusion criteria, and applying iterative screening
• Analysis of 438 papers | • Investigation of research evolution in 5 distinct phases
• Highly-granular multi-criteria (6) taxonomic analysis of literature, examining FM types, NN architectures, learning paradigms, learning stages, robotic tasks, and application domains
• Per criterion methodical comparative analysis of different approaches and insights reporting
• Comprehensive and hierarchical discussion on current challenges and future research directions | • No empirical/quantitative benchmarking of the surveyed methods
• Constitutes a snapshot of a rapidly evolving field |

# B    Details of literature review methodology

This section provides the full details of the structured literature review methodology that was followed in order to efficiently and thoroughly identify/map the robotic FM literature, while at the same time detecting key concepts and trends (complementing the condensed presentation of Section 2 of the main paper). The adopted systematic approach, which ensures comprehensiveness and relevance of the selected research works, consists of the iterative main steps described below.

## B.1    Scope and objectives formulation

The fundamental goal of the performed survey study was to review the robotic FM literature, i.e., approaches for various robotic tasks (namely, perception, planning, navigation, manipulation, and human-robot interaction) whose execution relies on the use of an underlying FM. In particular, the focus was on identifying the most recent advancements and works with substantial contribution to the field, emphasizing the following objectives: a) The main categories of methods based on multiple criteria, b) The utilized datasets, c) Current challenges, and d) Future research directions.

## B.2    Literature search

In order to ensure broad and thorough coverage of the relevant literature, the search strategy involved querying several major scientific databases, including IEEE Xplore, Google Scholar, Scopus, DBLP, arXiv, and Web of Science. The actual search was performed by combining targeted keywords/terms (e.g., "foundation model", "robotics", "vision-language-action", "large language model", etc.) with Boolean operators (i.e., "AND", "OR", "NOT"). To guarantee contemporary relevance, the search primarily focused on research works published within the last five years; however, certain earlier seminal studies were also included. An example of the query used in the Scopus database is provided below. It needs to be emphasized that this Scopus-style string comprises only an illustrative example of one search instance and does not constitute the complete literature search protocol.

TITLE-ABS-KEY("foundation model" OR "vision-language model" OR "visual foundation model" OR "large language model" OR "vision-language-action" OR "robotic foundation model")
AND TITLE-ABS-KEY("robot" OR "robotics" OR "manipulation" OR "navigation" OR "human-robot interaction" OR "embodied")
AND PUBYEAR > 2020
AND PUBYEAR <= 2026
AND (SRCTYPE(j) OR SRCTYPE(p))
AND EXCLUDE(DOCTYPE, "bk")

Multiple search steps were performed iteratively, involving refinements to the employed keywords so as to retrieve more relevant works. Moreover, the list of references of each research article was also analyzed in order to identify additional relevant studies. The combination of the above elements (i.e., the use of six complementary databases, the iterative keyword refinement, and the extensive backward reference-checking/chaining procedure) renders the overall literature review process relatively robust against missing critical/important works in the field. In particular, this design is capable of capturing seminal studies whose original terminology predates the currently established FM one and which, therefore, would be missed by a single keyword-based query.

Regarding the choice of the search mechanism, the adopted keyword plus reference search strategy was deliberately favored over a pure embedding-based semantic retrieval over paper embeddings. The main rationale is that the keyword-based approach offers improved reproducibility (the exact query strings, databases, and date bounds can be reported and re-executed), transparency (the inclusion of each work can be traced to explicit criteria rather than to an opaque similarity score), and precision/recall control (the Boolean structure allows for the systematic tuning of the retrieved set). In contrast, embedding-based semantic retrieval depends on a particular embedding model, a seed corpus, and a similarity threshold, none of which are easily auditable or reproducible across the six heterogeneous databases considered, most of which do not natively expose such a functionality. Nevertheless, it is acknowledged that embedding-based semantic search can serve as a useful complementary tool, particularly for further improving recall under

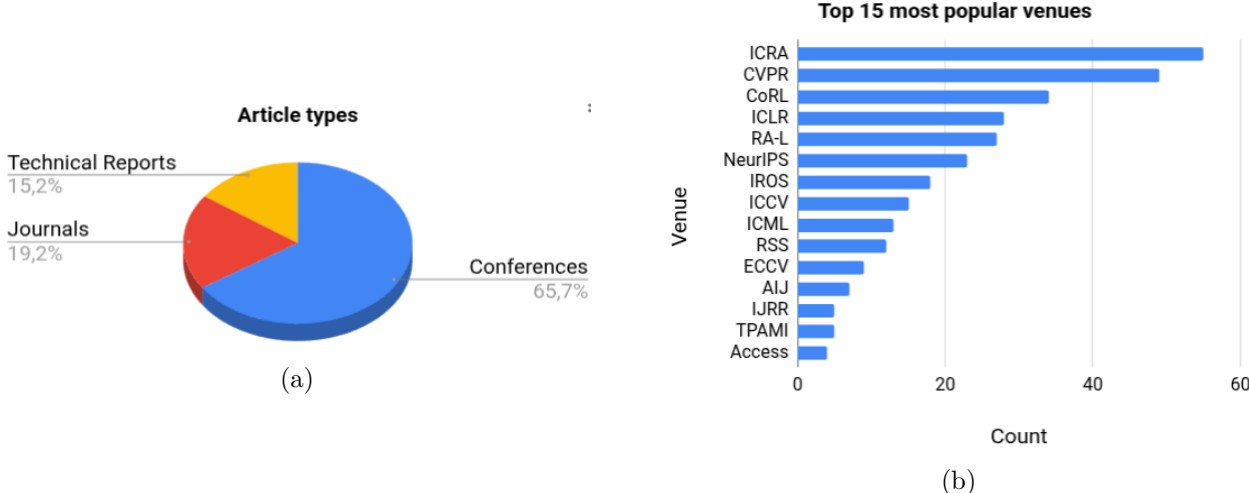

Figure 1: Key bibliometric analytics regarding robotic FM literature: (a) Article types, and (b) Top-15 most popular venues.

terminology drift, a role that in the current study was effectively fulfilled by the iterative refinement and the reference-chaining steps described above.

### B.3 Screening

Initial screening relied on excluding duplicate records, non-English papers, and articles without full-text access, in order to maintain the integrity of the review study. Then, article selection was performed taking into account title and abstract information, so as to eliminate irrelevant works. Subsequently, in-depth full-text review was performed for considering only research studies that: a) Focus on demonstrating approaches for implementing various robotic tasks (i.e., perception, planning, navigation, manipulation, and human-robot interaction), where a FM comprises a key algorithmic component, and b) Exhibit substantial theoretical and/or experimental contributions. Additionally, priority was given to research works originating from prominent robotics and AI/ML publication venues. Eventually, a total of 438 articles were selected for analysis and were included as references in the main manuscript.

Key bibliometric analytics regarding the performed literature review study are illustrated in Fig. 1.

### B.4 LLM assistance

Regarding the use of automated tools during the preparation process of the manuscript, LLM assistance was employed only on specific occasions and solely as a complementary aid for cross-checking potential gaps in the literature search described above (i.e., as a supplementary means of identifying relevant works that might have been missed by the database queries). However, all substantive scholarly work was carried out manually by the authors; in particular, the taxonomy design, the paper inclusion/exclusion decisions, all comparative methodical analyses, the study of individual works, and the inclusion and verification of all references were carefully performed and validated by the authors. The authors take full responsibility for the entire content of the manuscript.

## C   Foundation model types

This section provides the full details of the different types of FMs that are utilized in robotics, complementing the condensed presentation of Section 5 of the main paper. For each FM type (namely, LLMs, VFMs, VLMs, and VLAs), the complete category definition, the corresponding main advantages and limitations, as well as the extensive list of representative methods per sub-category of supported functionality are provided below.

### C.1 Large Language Models (LLMs)

In the context of robotics, LLMs are primarily used as high-level, cognitive task planners and reasoning engines that generate the sequence of operations that are necessary for accomplishing a stated goal (Brohan et al., 2023b; Liang et al., 2023; Chen et al., 2024b). So far, they have been shown to be particularly effective for tasks that require sophisticated, logical reasoning and complex decision-making. In practice, LLMs translate high-level natural language inputs to low-level robot behaviors, by turning free-form instructions and short state summaries into typed goals, multi-step plans, executable code, constraints, and run-time feedback. Their main advantages are (Li et al., 2025a): a) Robustness in instruction translation, where free-form, flexible, human-like natural language instructions are transformed into formal, actionable representations, b) Efficiency in task decomposition and sequencing, where complex, long-horizon natural language instructions are mapped to concrete sequences of robot sub-tasks, and c) Increased generalization ability, where the large amount of general-purpose knowledge incorporated in an LLM can boost the generation of robust action plans in diverse, complex, real-world environments. On the contrary, LLMs exhibit the following critical limitations (Tayyab Khan & Waheed, 2025): a) Lack of embodiment and grounding, where LLMs operate in a semantic space that does not incorporate connections to the physical environment; hence, leading to action plans that may not be feasible, b) Presence of hallucinations, where the LLM inference procedure may result into semantically/syntactically correct outcomes that, however, correspond to irrational actions, c) Increased latency, where the inherently high computational complexity of LLMs may not be suitable for real-time operational settings, and d) Input bias and sensitivity, where obscure or ambiguous input instructions may lead to unpredictable or biased action plans.

In terms of supported functionality/operation, LLM-based systems can be classified into the following main categories:

- Goal/constraint grounding and context: LLMs serve as a reasoning and translation interface that maps high-level, abstract human instructions (i.e., goal and context) to low-level, physical robot actions (i.e., grounding). In particular, SayCan (Brohan et al., 2023b) employs a base LLM mechanism (PaLM (Kim et al., 2024)) for filtering using value-based affordances, so that the selected skills to be maintained within capabilities and context. Additionally, LM-Nav (Shah et al., 2023a) converts instructions into visually grounded way-points that a planner can subsequently follow in a navigation setting. For larger spaces, SayPlan (Rana et al., 2023) associates language with 3D scene graphs, so that plans can remain consistent across different rooms and floors.

- Command interpretation and code synthesis: LLMs in principle act as natural language to code translators, enabling robots to receive high-level, human-like flexible instructions, instead of requiring the writing of conventional code. Specifically, Code-as-Policies (Liang et al., 2023) compiles instructions into robot Application Programming Interface (API) code using code-generating LLMs, which renders robot behavior easy to inspect and to reuse. Similarly, AutoTAMP (Chen et al., 2024b) translates requests into task-and-motion planning (TAMP) algorithmic specs that a symbolic planner checks for geometric and kinematic feasibility. Additional works focus on generating behavioral trees or decomposing tasks into formalized subproblems (Ao et al., 2025; Kwon et al., 2025; Liu et al., 2025c), synthesizing policy code from video-plus-text prompts (Xie et al., 2025), and constructing language-conditioned 3D value maps to guide placement and grasping (Huang et al., 2023c).

- Task planning and long-horizon reasoning: LLMs implement a high-level, cognitive mechanism that decomposes abstract human goals into sequential, grounded actions and maintains contextual awareness over multiple subsequent steps. In particular, SELP (Wu et al., 2025b) performs the mapping of language instructions to temporal logic representations, while combining constrained decoding with domain tuning, so that the generated plans to eventually satisfy safety and efficiency constraints. Similarly, LLM-GROP (Zhang et al., 2025f) cross-checks language-driven task instructions with motion feasibility and perceptual information for improving execution in cluttered settings. Moreover, hierarchical goal decomposition can speed up planning and robustness of long-horizon task execution (Kwon et al., 2025; Liu et al., 2025c; 2024a).

- Perception-aware and multimodal integration: The LLM core language processing functionality is enhanced by taking into account the robot's surrounding visual and physical world. In this context, PaLM-E (Driess et al., 2023) incorporates visual and proprioceptive information streams into the inference process, so that the model decisions to better correspond to real-world observations and actions. In a similar way, Chain-of-Modality (Wang et al., 2025a) employs prompts originating from both human-performing videos and auxiliary, associated signals (i.e., muscle or audio), in order to define both a task plan and corresponding control parameters from a single demonstration. Moreover, associating language inputs with visual grounding and 3D value maps can also facilitate more accurate robot manipulation (Huang et al., 2023c).

- Navigation and spatial understanding: LLMs translate abstract navigation commands into physically grounded, actionable paths within a map, primarily through the estimation of structured textual representations of the surrounding environment. For instance, LM-Nav (Shah et al., 2023a) links human instructions with spatial landmarks and routes, via a CLIP-based grounding mechanism, and passes way-points to a low-level navigation module. Additionally, SayPlan (Rana et al., 2023) partitions a plan over a 3D scene graph into layers and, subsequently, performs re-planning when an action step is infeasible, keeping in this way long-horizon missions on track in large-scale environments.

- Conversational interfaces and teleoperation: LLMs create intuitive, conversational interfaces and enable shared-control teleoperation, making robots accessible to non-expert users. In particular, TidyBot (Wu et al., 2023a) learns user-specific tidying conventions through conversation and transfers them to novel domestic scenarios. Additionally, LAMS (Tao et al., 2025) predicts human intent and automatically switches teleoperation modes in assistive settings, aiming at lowering cognitive load during extended task executions.

- Execution-time validation, error handling, and recovery: LLMs translate technical run-time failures into semantic problems, enabling the definition of dynamic, common-sense solutions, instead of relying solely on pre-defined, baseline, brittle recovery routines. Specifically, STATLER (Yoneda et al., 2024) collects and interprets robot state and tool feedback information, allowing the suggestion of targeted repairs without restarting the overall execution pipeline. Similarly, CAPE (Raman et al., 2024) performs re-prompting and proposes concrete fixes in case of preconditioned failures, while CoPAL (Joublin et al., 2024) realizes re-planning when divergence incidents are detected. Moreover, uncertainty estimation over planning proposals can provide additional mitigation measures, prior to acting on hardware (Yin et al., 2025a).

- Adaptation, efficiency, and safety: LLMs facilitate robotic systems to be adaptive, efficient, and safe during task execution, by robustly handling novel situations and operational disruptions. In particular, Eureka (Ma et al., 2024a) writes and refines reward code using GPT-4 (Achiam et al., 2023), accelerating in this way skill acquisition across diverse robot platforms. Additionally, DrEureka (Ma et al., 2024b) co-designs rewards and domain randomization for efficient sim-to-real transfer. AutoRT (Ahn et al., 2024) implements simultaneously instruction handling, task assignment, and safety checking across multiple robots. Moreover, planners can encode safety rules directly (e.g., SELP's use of temporal logic) (Wu et al., 2025b), while recent works study jailbreak-style exploits and propose corresponding defense measures (Ravichandran et al., 2025).

- Knowledge retrieval and memory: LLMs leverage their ability to access, synthesize, and store massive amounts of information from external knowledge and past execution episodes, allowing systems to overcome limitations of their immediate sensorial data or within a single planning session. For instance, ELLMER (Mon-Williams et al., 2025) combines GPT-4 with a retrieval-augmented memory mechanism, so that a mobile manipulator can incorporate context, adapt plans on the fly, and complete multi-step household tasks as conditions change. Similar architectures are capable of performing inference over planning route and tool knowledge for on-demand lookup (Temiraliev et al., 2026; Anwar et al., 2025).

## C.2 Vision Foundation Models (VFMs)

The ultimate goal of VFMs is to satisfy the perceptual requirements of embodied AI systems, by providing generalized, high-quality visual representations necessary for interaction with the physical world (Kirillov et al., 2023; Oquab et al., 2024). In the robotics setting, VFMs aim at distilling raw pixel information into rich, transferable visual features or embeddings. The latter enables a robust and generalized visual understanding that serves as the input information stream for modulating downstream policies, enabling the direct mapping of visual observations to specific control actions (Shang et al., 2024). Their main advantages are (Tayyab Khan & Waheed, 2025): a) Increased transfer learning capabilities, where visuomotor robot policies can efficiently generalize and acquire new skills, without requiring task-specific perception module learning from scratch, b) Open-world perception, where robots are enabled to recognize and to process previously unseen object and scene categories, c) Improved spatial awareness, where especially the incorporation of depth perception is crucial for enabling the execution of intricate actions with precision, and d) Increased robustness, where VFMs are shown to be less susceptible to visual distortions (e.g., presence of noise, variations in lighting conditions, etc.), compared to previous visual information processing modules. On the other hand, the main limitations of VFMs are (Awais et al., 2025): a) Domain specificity, where VFMs can often perform sufficiently well in a relatively narrow range of domains, b) Incomplete physical world dynamics modeling, where VFMs often exhibit limitations to generalize robustly to subtle physical dynamics, long-range temporal correlations, causal coherence, and geometric properties, and c) Increased computational cost, which may result into critical constraints regarding real-time, uninterrupted operation.

In terms of supported functionality/operation, VFM-based systems can be classified into the following main categories:

- Object recognition: VFMs enable generalized visual recognition, by providing transferable representations that reduce the need for task-specific perception training. In particular, SAM (Kirillov et al., 2023) provides class-agnostic, promptable segmentation masks that facilitate isolating objects and parts, while DINOv2 (Oquab et al., 2024) provides robust dense visual features that transfer well across scenes and support recognition and retrieval under domain shift (Kirillov et al., 2023; Oquab et al., 2024). Additionally, 3D-MVP (Qian et al., 2025a) uses a multi-view encoder to learn object and part-level representations that are useful for recognition and downstream manipulation. Moreover, ZeroGrasp (Iwase et al., 2025) couples recognition with reconstruction, estimating object geometry and predicting grasp poses from a single RGB-D observation in near real-time.

- Localization: VFMs enhance robot localization by estimating robust, semantic, and globally consistent visual representations, outperforming conventional geometric methods. Specifically, DINO-VO (Azhari & Shim, 2025) utilizes DINOv2-generated features and a ViT-based keypoint estimation scheme for improving robustness and generalization in monocular visual odometry at high throughput rates. Additionally, LiteVLoc (Jiao et al., 2025) and ZeroVO (Lai et al., 2025) support long-range re-localization for image-goal navigation and zero-shot cross-camera visual odometry, respectively.

- Object tracking: VFMs provide trackers with rich, semantic understanding and robust long-term memory, increasing robustness with respect to occlusion, viewpoint change, and category drift. In particular, Zhong et al. (2024) use a pre-trained VFM to extract semantic segmentation masks with text prompts, while a recurrent policy network with offline reinforcement learning is subsequently trained from the collected demonstrations. Additional approaches make use of open-vocabulary or frozen-backbone cues for instance tracking and segmentation, which is particularly useful for handling novel objects over time (Guo et al., 2025a; Fang et al., 2025a).

- Depth perception: VFMs enable robust, generalized, and high-fidelity depth estimation, especially in scenarios where specialized sensors or traditional methods tend to under-perform. Specifically, Metric3D v2 (Hu et al., 2024) employs geometric priors (namely, canonical camera transformations and joint depth-normal optimization) and supports zero-shot metric depth estimation across diverse camera settings. Additionally, Prompt-Depth-Anything (Lin et al., 2025) demonstrates that a small LiDAR 'metric prompt' can steer a FM to accurate, high-resolution metric depth estimation.

Similarly, DepthCrafter (Hu et al., 2025b) generates temporally consistent long depth sequences with intricate details for open-world videos.

- Semantic map creation: VFMs provide robust, semantic-aware features that enhance accuracy, robustness, and informativeness of the generated maps. In particular, Busch et al. (2025) create reusable open-vocabulary feature maps, capable of supporting probabilistic-semantic updating for informed multi-object exploration. In a similar way, VFM feature representations can be combined with Gaussian-splatting or factor-graph mechanisms for robust long-horizon missions in dynamic environments (Zheng et al., 2025; Yugay et al., 2025).

- Visual-inertial fusion: VFMs enhance the visual part of visual-inertial odometry (VIO) systems, which is crucial for drift correction and estimation of metric scale. Specifically, features that improve VO (Azhari & Shim, 2025) or metric priors from depth FMs (Hu et al., 2024) are combined with Inertial Measurement Unit (IMU) data in standard VIO estimators. Additionally, depth-foundation priors are injected to stereo/VIO pipelines for stabilizing scale and short-horizon pose, while being complemented by IMU FMs for cross-platform generalization (Jiang et al., 2025a; Zhao et al., 2025b).

- Environment mapping: VFMs enable map building by producing dense visual embeddings that can be fused into persistent, semantically enriched scene representations. FMGS (Zuo et al., 2025) combines FM features with 3D Gaussian splatting for semantic 3D scene reconstruction and open-vocabulary scene understanding. Similarly, OpenGS-SLAM (Yang et al., 2025a) and FeatureSLAM (Thirgood et al., 2026) extend Gaussian-splatting-based mapping with FM-derived semantic features, enabling open-set scene understanding and improving the robustness of real-time tracking and mapping.

## C.3 Vision-Language Models (VLMs)

VLMs combine computer vision with natural language processing capabilities for establishing a coherent, concrete semantic understanding of the world, enabling interpretation and generation of language descriptions of the observed visual entities (Radford et al., 2021). In the context of robotics, VLMs enable robots to simultaneously interpret visual data and natural language commands, allowing for intuitive human-robot interaction and robust task execution in unstructured environments (Zhou et al., 2025c). Their main advantages are (Tayyab Khan & Waheed, 2025): a) Richer semantic information and environment understanding, where VLMs generate detailed, interpretable semantic outputs that boost robots to handle complex and rare/novel scenarios more efficiently, b) Open-vocabulary object recognition, where the use of the VLMs' shared visual-language embedding space significantly facilitates open-world perception, c) Flexible perception interface, where embodied agents are capable of supporting nuanced queries and generalized reasoning, and d) Enhanced generalization capability, where VLMs allow for efficient handling of novel visual and linguistic combinations. On the contrary, the main limitations of VLMs are (Tayyab Khan & Waheed, 2025): a) Inability to define precise actions, where VLMs are capable of interpreting instructions and identifying visual entities, but they lack the intrinsic capability to translate this semantic understanding into precise, executable motor control commands, b) Incomplete semantic grounding, where VLMs exhibit difficulties in connecting abstract, human-like commands to concrete, actionable physical locations and poses required for robot manipulation tasks in real-world environments, and c) Dependency on external policy generators, where VLMs inherently require a dedicated external policy or low-level planner to provide their semantic output for implementing robot actions.

With respect to supported functionalities/operations, VLM-based systems can be classified into the following main categories:

- Manipulation grounding and control signals: VLMs map high-level, semantic intents into concrete, physically-actionable constraints in the proximity of the robot's operating environment. In particular, OmniManip (Pan et al., 2025c) translates VLM reasoning outcomes to object-centric interaction primitives and, subsequently, implements dual closed-loop procedures (namely, planning and execution) to produce precise 3D spatial constraints. Additionally, RoboGround (Huang et al., 2025a) provides grounded masks for both targets and placement regions into a low-level policy for improving generalization across different types of scenes. KUDA (Liu et al., 2025d) poses queries to a VLM for

task keypoints and, subsequently, converts them into optimization costs for model-based planning, enabling open-vocabulary manipulation over rigid, deformable, and granular objects. Moreover, chain-of-modality-style schemes prompt a VLM on human videos and auxiliary signals for extracting step-wise plans and control parameters (Wang et al., 2025a). In a similar way, IKER (Patel et al., 2024) and ReWiND (Zhang et al., 2025c) refine visually-grounded rewards or costs over long-horizon tasks for stabilizing execution, respectively.

- Semantic mapping, referring expressions, and navigation: VLMs are capable of creating human-readable maps, understanding complex spatial language, and grounding navigation goals in the physical world, materializing the transition from purely geometric to semantic navigation. In particular, One-Map-to-Find-Them-All (Busch et al., 2025) creates a reusable open-vocabulary feature map for real-time, zero-shot, multi-object-oriented navigation, while supporting probabilistic semantic updates. Additionally, functional and relational reasoning at scene level is enabled by the use of open-vocabulary functional 3D scene graphs (Zhang et al., 2025b) and open-scene graphs (Loo et al., 2025) for open-world object-based navigation. OpenVIS (Guo et al., 2025a) incorporates open-vocabulary video instance segmentation and tracking for pursuing tools or novel objects. Moreover, Vision-Language Fly (VLFly) (Zhang et al., 2025i) supports grounded vision-language navigation for Unmanned Aerial Vehicles (UAVs) with open-vocabulary goal understanding, without requiring localization or active ranging sensors.

- Execution-time check and progress verification: VLMs enable robot, semantic, self-monitoring within a closed-loop control framework, materializing the translation of low-level, sensor feedback to high-level, human-understandable checks. Specifically, ExploreVLM (Lou et al., 2025) deploys a closed-loop task planning framework for real-time integration of perception, planning, and execution validation. Additionally, Ahmad et al. (2025) introduce a unified framework for real-time failure recovery, where a VLM acts as a monitoring tool for verifying pre- and post-conditions of individual skills, inferring missing pre-conditions, and suggesting new skills for recovery. Moreover, Guardian (Pacaud et al., 2025) enhances the robot abilities to detect manipulation planning and execution errors, by identifying fine-grained failure modes (e.g., object slippage, incorrect action sequencing, etc.).

- Closed-loop mobile manipulation: VLMs provide the necessary cognitive capabilities regarding continuous feedback and adaptation, so as to directly address the inherent long-horizon execution challenge in unstructured, large-scale, and dynamic environments. For instance, COME-robot (Zhi et al., 2025) uses GPT-4 for situated reasoning and iterative feedback, in order to recover from failures. Additionally, HomeRobot (Yenamandra et al., 2023) relies on an agent that is capable of navigating through household environments for grasping novel objects and placing them on target receptacles.

## C.4 Vision-Language-Action models (VLAs)

VLAs aim at integrating multi-modal understanding with direct physical execution, targeting to serve as the basis for autonomous embodied task execution (Ma et al., 2024c). In particular, a VLA model receives multi-modal inputs (typically, vision, language, and robot state) and generates real-world physical actions or control policies in real-time, often designed in an end-to-end way. Their main advantages are (Sapkota et al., 2025): a) End-to-end implementation and operational simplicity, where VLAs perform the direct mapping from perception signals to control actions, eliminating the need for complex interconnection of multiple, distinct planning modules, b) Robust generalization ability, where VLAs are shown to demonstrate increased generalization performance across different operating environments and robotic platforms, and c) Integration of action and reasoning, where VLAs by design address the multi-modal, semantic grounding problem internally. On the other hand, the main limitations of VLAs are (Sapkota et al., 2025): a) High demand for large-scale training data, where VLAs require outstanding amounts of high-quality, heterogeneous action data, b) Decreased efficiency across multiple robotic platforms, where unavailability of not always sufficiently-broad robotic training datasets leads to suboptimal task performance across different robotic setups, c) Increased operational latency, where the large-scale nature of the underlying VLA model architectures leads to critical challenges in real-time control settings, and d) Increased complexity in failure management, where errors in end-to-end VLA systems may affect significantly the model's overall behavior.

In terms of supported functionality/operation, VLA-based systems can be classified into the following main categories:

- Scaling and web-to-robot transfer: VLAs target the transferring of vast semantic, visual, and common sense knowledge acquired from the internet to the robots' physical-world control policies, while reinforcing generalization across different tasks and embodiments. In particular, RT-1 (Brohan et al., 2023a) scales-up imitation learning capabilities using tokenized action sequences corresponding to language formalized goals, improving robustness in long-tail household task execution. Its successor RT-2 (Zitkovich et al., 2023) incorporates web-scale vision-language pretraining in the control pipeline, so that open-vocabulary knowledge transfer to be performed to real-world robots, through the use of a language-aligned visual encoder. Additionally, OpenVLA (Kim et al., 2025) integrates a vision-language encoder and a relatively small action head, while supporting different robotic platforms and maintaining increased zero/few-shot generalization capabilities. Similar approaches incorporate stronger inductive biases and specific 3D spatial priors (Li et al., 2025c), object-centric adapters for few-shot tuning (Li et al., 2025b), standardized cross-embodiment corpora (O'Neill et al., 2024), multiple low-rank adaptation modules (Zhao et al., 2025a), and policy distillation schemes (Xu et al., 2025).

- Fusion and action parameterization: VLAs aim at unifying perception, reasoning, and control by combining a VLM with an action decoder. In particular, GR00T N1 (Bjorck et al., 2025) tightly couples a VLM for interpreting the environment, through vision and language instructions, with a subsequent diffusion transformer for generating motor actions in real-time. Additionally, $\pi_{0.5}$ (Black et al., 2025) formalizes action generation as a flow matching process on top of a VLM, supporting stable, continuous control and increased generalization at moderate computing requirements. Moreover, similar systems further improve the interaction between the VLM and the action decoder through several complementary design choices, including diffusion models for enhanced dexterity (Wen et al., 2025b), unified autoregressive-diffusion heads (Liu et al., 2026a), rectified-flow policies (Reuss et al., 2025), state-space formulations (Liu et al., 2024b), short-horizon video prediction (Hu et al., 2025c), and embodied reasoning mechanisms (Team et al., 2025).

- Specialization, adapters, and mixture-of-experts: VLAs accomplish to leverage massive pre-trained backbones without the need/cost of parsing/running the entire network for every predicted action. Specifically, MoRE (Zhao et al., 2025a) makes use of sparse Low-Rank Adaptation (LoRA) experts, selecting only a few of them per step for expanding the model's capacity without increased inference cost. Additionally, OpenVLA (Kim et al., 2025) combines a Llama 2 language model with a visual encoder and implements efficient fine-tuning for new tasks, boosting robust, generalizable policies for visuo-motor control. Moreover, RLDG (Xu et al., 2025) combines task-specific Reinforcement Learning (RL) with generalist policy distillation for more efficient robotic manipulation.

- Navigation and locomotion: VLAs serve as semantic, navigation planners that translate abstract, natural-language instructions into physically executable movements by mobile and legged robots, often adopting a hierarchical design. In particular, NaVILA (Cheng et al., 2025) initially generates mid-level actions with spatial information in the form of language and, subsequently, utilizes this as input to a visual locomotion RL policy generator for execution. Additionally, VAMOS (Castro et al., 2025) comprises a hierarchical VLA that decouples semantic planning from embodiment grounding, where a generalist planner learns from diverse, open-world data and a specialist affordance model encodes the robot's physical constraints and capabilities. Moreover, Humanoid-VLA (Ding et al., 2025) integrates language-motion pre-alignment (using non-egocentric human motion data paired with textual descriptions), and egocentric visual context (through parameter efficient video-conditioned fine-tuning).

- Operations, deployment, and safety: VLAs aim at combining their high-level, flexible, and generalized reasoning capabilities with conventional safety guarantees for real-world deployment. Specifically, SafeVLA (Zhang et al., 2025a) formalizes safety alignment as a constraint learning problem, targeting the VLA operation to respect task rules and safety measures, and not relying only on post-hoc filtering.

VLATest (Wang et al., 2025f) comprises a framework designed to generate robotic manipulation scenes for testing VLAs. Moreover, fleet orchestration frameworks combine language interfaces with guardrails and human-in-the-loop supervision, while mechanistic steering implements zero-shot behavior shaping (Ahn et al., 2024; Häon et al., 2025).

## D  Neural network architectures

This section provides the full details of the different types of neural network (NN) architectures that are utilized in robotic FM methods, complementing the condensed presentation of Section 6 of the main paper. For each architecture type (namely, transformers, state-space models, diffusion models, convolutional and hybrid encoders, and graphical models), the complete category definition, the corresponding main advantages and limitations, as well as the extensive list of representative methods per sub-category are provided below.

### D.1  Transformers

In the context of robotics, transformers are widely used for different/diverse tasks (e.g., high-level task planning, low-level policy learning, perception, human-robot interaction, etc.), relying on the fundamental principle of formalizing them as a sequence modeling problem (Firoozi et al., 2025). The latter is grounded on converting input data (e.g., states, actions, images, etc.) to numerical tokens, which are then processed as a sequence for estimating a prediction. Their main advantages are (Sanghai & Brown, 2024): a) Increased long-range dependency modeling, where transformers enable policies to capture complex, long-range dependencies between states, actions, and rewards over long time horizons, b) Architectural homogenization, where a single, well-understood transformer architecture can be used as the backbone for diverse robotic tasks and input modalities, and c) Increased efficiency in high-level planning and reasoning, where the primary origin of the transformer architecture in language modeling also makes it suitable for high-level, language-conditioned task planning. On the other hand, the main limitations of transformers are (Firoozi et al., 2025): a) Increased computational complexity, where the transformer self-attention mechanism compute and memory requirements scale quadratically with the input sequence length; hence, often making pure transformers unsuitable for low-latency, real-time control loops, b) Requirement for discrete tokenization, where transformers inherently operate on discrete tokens, which in turn introduces quantization error and is a fundamentally unnatural representation of physical dynamics, and c) Infinite context contradiction, where a pure-form transformer exhibits outstanding reasoning performance over a finite context, but it is architecturally unsuited for infinite-horizon, continuous processing required by embodied agents.

With respect to the input data modality, transformer-based systems can be classified into the following main categories:

- Vision Transformers (ViTs): ViTs comprise the fundamental architecture of multiple VFMs in robotics. They use self-attention to capture global image relationships, unlike traditional local convolutional methods (Yuan et al., 2021). Models like DINOv2 use self-supervised learning on web-scale datasets to estimate general-purpose visual features that transfer across tasks without extensive fine-tuning (Oquab et al., 2024). As a consequence, ViTs have a central role in several perception pipelines in robotics. One such key area is scene understanding, which focuses on the immediate, local environment (including 3D scene understanding) for realizing manipulation, localization, and visual odometry (Azhari & Shim, 2025; Martins et al., 2025a). Additionally, in the context of semantic mapping, ViT features facilitate the construction/maintenance of long-term, global, semantic maps, supporting open-vocabulary representations for multi-object navigation, SLAM, and targeted exploration (Busch et al., 2025; Laina et al., 2025; Martins et al., 2025b; Jiang et al., 2025b; Deng et al., 2025b). Moreover, ViTs also provide essential geometric cues for low-level control, by generating dense spatial signals (such as zero-shot metric depth and surface normals) from a single image (Hu et al., 2024; Yang et al., 2024b; Guo et al., 2025b); these signals, along with grounded spatial constraints, support closed-loop execution and grasp planning in manipulation systems (Huang et al., 2025a).

- Text transformers: Text transformers act as the language interface, planner, programmer, memory, and supervisor across robotic pipelines, providing a common semantic layer that connects high-level intent to grounded perception and control. In particular, bidirectional encoders (e.g., BERT) learn transferable semantics for downstream modules (Devlin et al., 2019) and decoder-only models (e.g., PaLM (Kim et al., 2024) and GPT-4 (Achiam et al., 2023)) benefit from scale, improving robotic reasoning and planning performance. Also open-weight LLaMA backbones enable practical on-device deployment (Touvron et al., 2023). In terms of exhibited functionality, text transformers can translate free-form language into structured plans and formal artefacts, by decomposing long-term goals, filling behavioural trees, and generating PDDL templates that pass feasibility checks (Ao et al., 2025; Liu et al., 2025c; Zhang et al., 2025f). Similarly, the same models can generate or adapt policy code from language or video inputs, enabling low-level controllers to remain auditable and closing the gap between high-level intent and executable actions (Liang et al., 2023; Xie et al., 2025; Ji et al., 2026). Additionally, retrieval-augmented setups can ground the planner in external memory and past context, in order to improve long-horizon behaviours, to sustain states across sub-goals, and to reduce drift during environmental changes (Mon-Williams et al., 2025; Gu et al., 2024; Anwar et al., 2025). In case of action execution divergence from the planned one, text transformers are able to propose corrections, to reason about failed preconditions, and to maintain explicit state estimations to prevent cascading errors/effects (Yoneda et al., 2024; Raman et al., 2024; Joublin et al., 2024). Moreover, even in generalist vision-language-action frameworks, the textual component remains on top of the formed reasoning layer, efficiently modulating the respective perception and control ones (Zitkovich et al., 2023; Mees et al., 2024; Bjorck et al., 2025; Kim et al., 2025; Team et al., 2025).

- Multi-modal transformers: These models transform each individual sensor stream into a sequence of tokens, project them to a common/shared latent space, and subsequently align them using a cross-attention or a gated fusion mechanism. The training process combines alignment and generative objectives, so that a single formed backbone network to be capable of understanding scenes, following instructions, and selecting actions (Kim et al., 2025; Zitkovich et al., 2023; Driess et al., 2023; Team et al., 2025; Mees et al., 2024; Bjorck et al., 2025). Multi-modal transformers are also capable of combining vision with proprioception information, where the fusion of visual and robot state tokens allows the generation of embodiment-aware policies that can scale across multiple/diverse robots and tasks (Wang et al., 2024b; Mees et al., 2024; Zitkovich et al., 2023). An additional capability comprises the enrichment of vision with geometric cues, where the integration of metric depth and surface normals can strengthen mapping and grasp planning with dense geometry that can generalize to novel scenes without requiring task-specific labels (Hu et al., 2024; Yang et al., 2024b; Huang et al., 2025a). Moreover, multi-modal transformers enable the generation of unified tactile-vision embeddings that can boost reasoning capabilities regarding contact, texture, and stability; hence, improving manipulation under uncertainty and occlusion (Yang et al., 2024a; Feng et al., 2025). On another direction, joint audio-visual tokens can support navigation in noisy, multi-source settings and improve robustness in the presence of distractions (Shi et al., 2025; Park et al., 2026). Furthermore, transformer-based fusion of thermal and RGB information streams can improve perception and localization, especially under low light and adverse weather conditions (Puttagunta et al., 2024; Skorokhodov et al., 2026). More recently, lightweight adapter modules are widely used for enabling the incorporation of additional sensors without retraining the full model, which comprises a common pattern shared across VLA and heterogeneous pre-training schemes (Kim et al., 2025; Wang et al., 2024b; Team et al., 2025).

## D.2 State-space models

SSMs are increasingly adopted in robotics for realizing real-time control and long-horizon reasoning, by treating the sensorial input streams as latent states and subsequently producing output predictions in a step-by-step way (Liu et al., 2024b). The latter is in practice performed by learning end-to-end system matrices with diagonal-plus-low-rank parameterizations and hardware-aware modeling (Gu & Dao, 2024; Dao & Gu, 2024). Their main advantages are (Gu et al., 2022; Smith et al., 2023): a) Linear scaling of complexity, where computation and memory requirements grow linearly with sequence length, b) Stable long-horizon memory, where the latent state maintains temporal context without large activation caches,

and c) Deployment efficiency, where low latency and steady throughput are suitable for embedded robotic solutions. On the contrary, the main limitations of SSMs are (Lenz et al., 2025): a) Reduced cross-token check, since explicit/direct correlation between tokens is not inherently supported, b) Reduced global context modeling, compared to full attention-based counterparts, and c) Frequent need for hybrid designs, which typically integrate attention or retrieval blocks for implementing tool use and long-range planning.

With respect to the input modalities, SSM-based systems can be classified into the following main categories:

- Visual SSMs: Visual SSMs can replace attention mechanisms with corresponding selection-based ones, resulting into linear-complexity encoders that can be plugged into recognition, dense prediction, and tracking heads (Liu et al., 2024e; Xiao et al., 2025a). Additionally, scene understanding and object tracking can benefit from long temporal context modeling at constant cost, which is particularly suitable for long video streams and multi-camera setups (Park et al., 2024). Moreover, event-driven perception can be boosted to handle irregular sampling and rapid environmental dynamics, which is beneficial for agile navigation and manipulation in low light or high motion settings (Zubic et al., 2024).

- Policy/control SSMs: SSMs can be used for realizing data-driven nonlinear reduction in complex systems, such as modeling hysteresis and memory effects. In particular, table-top manipulation can adopt region-aware selective-state policies with flow-matching objectives to learn precise, real-world skills from limited demonstrations (Wang et al., 2025c). Additionally, hybrid selective-state diffusion policies may reduce parameters while maintaining performance, in order to improve sample efficiency under multi-view inputs and long horizons (Cao et al., 2025a). Moreover, vision-driven locomotion can incorporate depth and proprioception information, through stacked selective-state formalisms and end-to-end reinforcement learning (Wang & Tao, 2026).

- Multimodal SSMs: A single SSM trunk can be simultaneously pretrained on long videos, robot logs, and demonstrations, in order to support perception, planning interfaces, and action heads. In particular, SSM-based VLA models can fuse vision, language, and proprioception inputs as token streams and, subsequently, generate actions with lower latency and memory than attention-only implementations (Tsuji, 2025), which makes the application of long-horizon policies more efficient and practical (Liu et al., 2024b). When explicit/direct lookup, tool use, or long-range queries are required, attention or retrieval layers can be integrated on top of the main SSM model (Lenz et al., 2025; Wang et al., 2025d).

## D.3 Diffusion models

DMs can generate robot behaviours by reversing a gradual noising process, where a forward pass increasingly adds Gaussian noise to the input data and, subsequently, a learned reverse model denoises back to the original input space (e.g., actions, trajectories, sub-goals, etc.) (Ho et al., 2020; Song et al., 2021). In particular, DMs implement control policies as a conditional denoising process and have been shown robust across different manipulation tasks (Chi et al., 2025), while they can also serve as generative heads on top of pretrained vision and multimodal backbones (Kapelyukh et al., 2024; Zeng et al., 2024). Their main advantages are (Liang, 2025): a) Multi-modal feature modelling, where sampling can produce diverse, uncertainty-aware candidates that can facilitate planning under partial observability and contact variability (Janner et al., 2022; Chi et al., 2025), b) Composite conditioning, where a single denoising process can employ frozen vision, language, depth, and proprioceptive backbones to modulate goals and action chunks without task-specific labels (Kapelyukh et al., 2024; Zeng et al., 2024; Ze et al., 2024), and c) Trajectory-level decision making, where value or constraint guidance can steer full roll-outs towards safe and feasible plans, while improving long-horizon behavior (Janner et al., 2022). On the contrary, the main limitations of DMs are (Wolf et al., 2025): a) Sampling latency and energy cost, stemming from the inherent iterative denoising process (Dong et al., 2024), b) Lack of built-in safety guarantees, where constraint checks or guided objectives are needed to avoid collisions and dynamics violations (Janner et al., 2022; Wolf et al., 2025), and c) Sensitivity to conditioning drift, where errors/noise in visual or language features can mislead the sampling process (Chi et al., 2025; Ze et al., 2024).

With respect to the conditioning type, DM-based systems can be classified into the following main categories:

- Vision-conditioned DMs: DMs can translate visual goals into usable, structured information for control purposes. In particular, image-goal generation and rearrangement of priors can estimate object- and scene-level targets that downstream controllers can subsequently follow (Kapelyukh et al., 2023; Zeng et al., 2024). Additionally, pretrained image-editing DMs can generate sub-goal images from language instructions and current camera views, guiding goal-conditioned policies in real-world settings (Black et al., 2024). Moreover, compact 3D visual tokens can be employed for improving spatial grounding and robustness across different viewpoints for manipulation planning (Chi et al., 2025; Ze et al., 2024; Kapelyukh et al., 2024).

- Proprioception-, force-, and haptic-conditioned DMs: Visuomotor diffusion policies treat action sequences as denoised samples conditioned on images and robot states, which facilitates in handling multi-modal actions and improving stability for manipulation tasks (Chi et al., 2025). When force or contact requirements are present, conditioning on haptics and force signals can be incorporated, for example, in visual-tactile slow-fast policies for contact-rich skills (Shukla et al., 2025; Xue et al., 2025). In order to maintain control loops short, progressive refinement can increase prediction rates up to real-time performance (Dong et al., 2024).

- Language-conditioned DMs: Textual inputs can serve as a guiding signal, where the denoising mechanism can modulate goals, trajectories, or sub-goals, simplifying task setup and execution (Bjorck et al., 2025). Additional works demonstrate the growing use of language prompts for manipulation and planning tasks based on diffusion backbones (Wolf et al., 2025; Liang, 2025).

- Human behaviour-conditioned DMs: Diffusion objectives can be defined so that early-stage human motion detection can facilitate the accurate prediction of intent, improving intuitiveness and comfort in human-robot interaction without changing the controller structure. In this context, the Legibility Diffuser (Bronars et al., 2024) demonstrates that a policy trained on offline demonstrations can generate intent-expressive collaborative motions that humans find easier to understand, while still completing a given task more efficiently (Ng et al., 2023).

## D.4 Convolutional and hybrid encoders

Visual encoders typically comprise the main perception module of any robotic solution, translating raw pixel data into latent representations that a robot can use for subsequent planning procedures (Nair et al., 2023). The selection between CNNs and hybrid CNN-transformer implementations essentially comprises a decision on the trade-off between local spatial precision and global context modeling, respectively. Their main advantages are (Brohan et al., 2023a; Tan & Le, 2019): a) Zero-shot generalization, where due to the fact that the models are trained on internet-scale datasets, previously unseen objects can often be recognized, b) Robustness to noise, exhibiting increased resilience to changes in lighting, shadows, or cluttered backgrounds, and c) Reduced need for training data, where 'frozen' pre-trained encoders often exhibit increased performance, without re-training. On the contrary, their main limitations are (Ma et al., 2023): a) High computational latency, due to the typical high-scale of FMs, b) Loss of fine-grained details, where encoders often divide processing of images in patches for reducing memory requirements, c) Prone to distribution shifts, where there might be a significant discrepancy between the internet-scale training data and the specific application images, and d) Lack of temporal consistency, due to many visual encoders processing video streams in an independent frame-by-frame fashion.

With respect to the encoder and integration type, the following main categories can be identified:

- CNN encoders: CNNs excel at capturing low-level spatial details like edges, textures, and object boundaries, due to their local receptive fields. In particular, ResNet-series encoders are used for high-fidelity state representations in diffusion-based exploration tasks in (Cao et al., 2025b). Additionally, EfficientNet-B3 is employed as a visual encoder to facilitate real-time, goal-conditioned navigation and exploration in (Sridhar et al., 2024). Additionally, R3M (Nair et al., 2023) freezes a ResNet-50 trained on Ego4D for improving manipulation for both simulation and real-world scenarios. Moreover,

language-reasoning segmentation masks generated by internet-scale trained encoders are leveraged to condition robot manipulation tasks (Yang et al., 2025b).

- CNN-transformer hybrids: Hybrid architectures often combine a CNN for handling pixel-level information and a transformer one for addressing context and action aspects. In particular, RT-1 (Brohan et al., 2023a) encodes frames with a FiLM-conditioned EfficientNet (Tan & Le, 2019), compresses them with TokenLearner, and then predicts discrete actions using a transformer, enabling real-world task control. Additionally, BC-Z/PaLM-SayCan (Jang et al., 2022; Brohan et al., 2023b) employ a lightweight ResNet coupled with a shallow attention network for supporting instruction-conditioned policies. In a similar way, diffusion-transformer policies may also adopt convolutional components as image tokenizers prior to respective transformer layers (Dasari et al., 2025).

- CNN tokenizers inside generalist agents: Generalist agents typically convert high-resolution images into compact tokens, prior to sequence modeling. In particular, Gato (Reed et al., 2022) employs a small ResNet image tokenizer and feeds visual, text, and proprioception information to a single transformer for controlling multiple skills. RoboCat (Bousmalis et al., 2024) makes use of a pretrained VQ-GAN image tokenizer and a transformer network, in order to adapt across robots and tasks according to a sequence of self-improvement cycles. Moreover, CNN tokenizers achieve to maintain low information bandwidth and to preserve an efficient interface to large sequence models for various navigation and manipulation tasks (Shah et al., 2023c).

- CNN-conditioned diffusion policies: Diffusion policies often condition a temporal U-Net on CNN features for generating action chunks that are diverse, yet feasible. In particular, Diffusion Policy (Chi et al., 2025) employs ResNet-18 features for robot manipulation, while preserving low added latency. DiffuserLite (Dong et al., 2024) demonstrates that progressive refinement with a frozen MobileNet-V3 encoder can increase prediction rates towards real-time performance on embedded platforms. Additional works concentrate on similar CNN-based diffusion frameworks for imitation and reinforcement learning purposes (Liang et al., 2025; Dasari et al., 2025).

### D.5 Graphical models

In the context of robotic FM methods, the use of graphs introduces additional capabilities towards the goal of connecting low-level, raw sensorial data and high-level, structured reasoning (Maggio et al., 2024; Booker et al., 2024). Unlike other NN architectures that process input data as matrices (e.g., images) or sequences (e.g., text tokens), graphs enable the modeling and interpretation of the surrounding environment as a set of interconnected entities. For example, scene graphs often decompose the visual world into nodes (e.g., objects, parts, or agents) and edges (e.g., spatial, semantic, or functional relationships), essentially aiming at modeling their inter-dependencies. Their main advantages are (Gu et al., 2024): a) Combinatorial generalization, where the learning of relationships among entities (instead of specific instances) can boost the generalization to previously unseen cases, b) Permutation invariance, due to the inherent ability of graphs to learn the structure (and not the order) of the data, c) Sample efficiency, which derives from the strong inductive bias natively incorporated in a graph model, and d) Increased explainability, due to the improved efficiency in interpreting the reasoning process of a graphical model. On the contrary, their main limitations are (Rana et al., 2023): a) Computational overhead, where the complexity of the message-passing algorithm in large graphs can introduce critical latency for real-time control applications, b) Dynamic topology, which relates to the mathematical difficulty in modeling real-world, dynamic environments in a stable way, and c) Need for integration with latent spaces, where graphs typically need to connect and to operate on precomputed, high-dimensional vector spaces.

With respect to the graph and function type, the following main categories can be identified:

- Scene graphs: Open-vocabulary 3D scene graphs associate vision-language features with real-world entities (often in a hierarchical way), while remaining compact compared to dense map representations. This structure enables robots to query targets, to reason about relationships, and to define sub-goals to planners in large-scale environments (Gu et al., 2024; Rana et al., 2023; Yan et al., 2025). More recently, graphs are constructed online directly from RGB-D streams, using hierarchical structures

for language-grounded navigation and adding functional links to the incorporated entities (Werby et al., 2024; Yin et al., 2024; Zhang et al., 2025b).

- Shared graphs: Compressed-form scene graphs allow bandwidth-limited sharing and map merging, while maintaining open-vocabulary query capabilities. In particular, decentralized visual FMs can estimate peer poses and can produce local Bird's-Eye View (BEV) maps on embedded hardware, while reducing communication requirements without losing key semantic information (Blumenkamp et al., 2025; Gu et al., 2025). Such designs make robot-team perception and planning feasible in large-scale environments.

- Graph neural networks: Graph Neural Networks (GNNs) enable message passing over task, object, and agent graphs for performing allocation, scheduling, and policy conditioning in a data-driven way. Recent hybrid, cognitive pipelines couple GNN-based scene graphs with LLM or symbolic planners, achieving to maintain plans physically feasible, while at the same time still following language goals (Tong et al., 2026; Strader et al., 2025).

- Embodiment graphs: Embodiment graphs encode robot joint information, as well as links between them, allowing a single learned policy to adapt across different platforms. In this respect, attention or message passing algorithms follow the learned graph connectivity, which in turn boosts zero-shot transfer to new morphologies and supports reusable controllers across different robots (Patel & Song, 2025).

## E   Learning paradigms

This section provides the full details of the different learning paradigms that are utilized in robotic FM methods, complementing the condensed presentation of Section 7 of the main paper. For each learning paradigm (namely, supervised learning, self-supervised learning, fine-tuning, domain adaptation, imitation learning, reinforcement learning, in-context/prompt learning, world model learning, and generative learning), the complete category definition, the corresponding main advantages and limitations, as well as the extensive list of representative methods per sub-category are provided below.

### E.1   Supervised learning

Supervised learning in robotic FM development trains a model on large-scale, labeled datasets, directly fitting a mapping from inputs (e.g., images, language instructions, and proprioceptive states) to target outputs (e.g., class labels, tokens, or action commands) under explicit human-provided supervision. Its main advantages are (Xiao et al., 2025b): a) High task accuracy, since explicit targets provide a direct and well-defined optimization signal, b) Stable and efficient optimization, where labeled objectives yield reliable convergence, c) Strong semantic transfer, where supervision on internet-scale labeled corpora injects rich semantic priors that transfer to downstream tasks, and d) Straightforward evaluation, where labeled targets make performance directly measurable. On the contrary, its main limitations are (Zitkovich et al., 2023): a) Dependence on costly human annotation, which is difficult to scale for embodied robot data, b) Limited coverage, where supervised models generalize poorly beyond the distribution of the labeled set, c) High computational cost when paired with internet-scale corpora, and d) Safety concerns, where the presence of hallucinations in the model behavior leads to lacking of formal safety guarantees. In this context, PaLM-E (Driess et al., 2023) is jointly trained using web-scale labeled multi-modal data and embodied experiences, while RT-2 (Zitkovich et al., 2023) and Gemini Robotics (Team et al., 2025) couple web-scale vision-language supervision with labeled robot manipulation data for open-vocabulary, instruction-following control.

### E.2   Self-supervised learning

SSL techniques employ a set of 'pretext tasks' (e.g., predicting the next video frame, reconstructing a masked image patch, etc.) and large quantities of unlabeled data streams (e.g., visual, depth, force, audio, robot logs, etc.) for enabling FMs to acquire common sense knowledge regarding physics, object permanence, and spatial relationships (He et al., 2022; Tong et al., 2022; Oquab et al., 2024). Their main advantages are

(Nair et al., 2023; Wu et al., 2023b; Assran et al., 2025): a) Massive scalability, where robots can learn from internet-scale datasets, without requiring human supervision, b) Sample efficiency, where only a handful of training samples is needed for adapting to new tasks, c) Zero-shot generalization, where SSL-trained models often exhibit increased performance in novel settings, and d) Autonomous improvement, where SSL methods can infer reward signals for training directly from the data itself. On the contrary, their main limitations are (Nair et al., 2023; Wu et al., 2023b): a) Embodiment gap, where most large-scale SSL-employed sources lack proprioceptive data (e.g., joint torques and forces), b) High computational cost, which typically requires massive GPU resources, c) Presence of hallucinations, where the learned models may infer physically impossible actions, and d) Evaluation difficulty, where it is inherently challenging to measure the success of SSL-learned representations, until actual model deployment.

The most common SSL techniques used for developing robotic FM solutions are:

- Masked Autoencoder (MAE): The model learns by reconstructing missing or corrupted parts of the input data, aiming at modeling robust spatial and temporal features. Image MAE (He et al., 2022) and its video counterpart VideoMAE (Tong et al., 2022) are widely used for producing robust visual backbone networks. Additionally, robotics-specific masked-pretraining approaches, such as 3D-MVP (Qian et al., 2025a), adapt the MAE paradigm to robot learning for manipulation.

- Contrastive learning: The fundamental principle relies on aligning matched views (and separating mismatched ones) for creating discriminative features and supporting open-vocabulary grounding, so that robots can link language to perception and retrieval skills. CLIP (Radford et al., 2021) establishes vision-language alignment at scale, while egocentric robot features, like R3M (Nair et al., 2023), can improve manipulation sample-efficiency using recorded human-performing videos.

- Autoregressive sequence modeling: The ultimate goal is grounded on predicting the next token in vision, language, or action streams, in order to capture long-horizon structure and to enable unified perception-to-policy modeling. Generalist agents, such as Gato (Reed et al., 2022), demonstrate how one sequence model can condition on images, text, and proprioception to produce actions across many tasks. Additionally, LLMs, like GPT-3 (Brown et al., 2020) and GPT-4 (Achiam et al., 2023), showcase the scalability and cross-domain reasoning capabilities of autoregressive transformers in embodied pipelines (Turcato et al., 2025; Mon-Williams et al., 2025).

- World model learning: Also often reported as an individual learning paradigm (Section E.8), WMs aim at learning to predict how the world will change in response to specific actions, essentially encoding causal relations. Various individual WM-based methods are introduced for varying environmental settings, including Dreamer-style agents on physical robots (Wu et al., 2023b), JEPA-based designs (such as AdaWorld (Gao et al., 2025) and ACT-JEPA (Vujinovic & Kovacevic, 2025)), and compositional video world models (like RoboDreamer (Zhou et al., 2024a)).

### E.3   Fine-tuning

Fine-tuning aims at adapting the internet-scale acquired knowledge to specific physical-world execution settings. In particular, it targets to adjust the rich, common-sense knowledge structures of a pretrained FM to a specific robot, sensor suite, or task, making use of a smaller, in-domain, annotated dataset. Its main advantages are (Yu et al., 2025): a) Sample efficiency, where only a relatively reduced set of training data is needed for the new/specific robot task, b) Domain adaptation, allowing the pretrained FM to handle different/specific application settings that are not present in the original training set, and c) Improved precision, where the learned policies are enabled to become more accurate and robust. On the contrary, its main limitations are (Mees et al., 2024): a) Catastrophic forgetting, related to the risk of the FM losing part of its general-purpose skills, b) Overfitting, which may occur when the fine-tuning dataset is relatively small and the FM may extensively adapt to specific operational settings, and c) Need for annotated data, where the requirement for high-quality robotic data is still present.

The most common fine-tuning techniques in robotics are:

- Full fine-tuning: The goal is to update all FM weights using a relatively small, in-domain robot dataset, in order to re-target a pretrained policy to a new robot, sensor setup, or task. Recent generalist policies report fast adaptation to new observation and action spaces on standard GPUs, using full fine-tuning as the baseline learning method (Mees et al., 2024; Kim et al., 2025).

- Low-Rank Adaptation (LoRA): The fundamental aim relies on maintaining the original FM weights unchanged and adding two low-rank matrices that learn/incorporate the required model modifications. In particular, OpenVLA (Kim et al., 2025) demonstrates LoRA-based tuning on the large-scale Open-X-Embodiment dataset (O'Neill et al., 2024). Additionally, more recent approaches (including quantized LoRA versions) target model adaptation in resource-constrained robotic platforms (Williams et al., 2026; Kim et al., 2026).

- Quantized parameter-efficient fine-tuning: This extends the original Parameter-Efficient Fine-Tuning (PEFT) approach, by combining low-precision weights with LoRA-style adapters for maintaining latency and memory requirements low on embedded or field hardware, while at the same time retaining most of the full-precision performance. In particular, LiteVLA (Williams et al., 2026) and similar approaches (Williams et al., 2025) report that an 4-8 bit quantization plus adapters can preserve high recognition rates, while enabling real-time control on smaller platforms.

- Action space remapping: This relies on integrating lightweight encoders/decoders or tokenizers, so that the core policy can be fine-tuned to new sensors or actuators without retraining the FM from scratch. Generalist policies, such as Octo (Mees et al., 2024) and RT-2 (Zitkovich et al., 2023), rely on this technique to re-target a given FM to multiple robots and grippers with modest additional data.

### E.4 Domain adaptation

Domain Adaptation (DA) aims at bridging the critical gap between FMs being pre-trained on massive internet-scale data and their subsequent deployment in real-world environments. In practice, DA targets to equip FMs with the required physical-world grounding or specific environmental awareness capabilities for given application scenarios, which is typically termed as the 'sim-to-real' transfer challenge. Its main advantages are (Da et al., 2025): a) Reduced need for real-world samples, where the use of simulation data accommodates the need for extensive amounts of real-world ones, b) Robustness to sensor shifts, where models are boosted to remain stable in case of robot hardware changes, and c) Safety guarantee, where, since training is performed in simulation environments, the risk of robots damaging their surroundings during the learning phase is reduced. On the contrary, its main limitations are (Tayyab Khan & Waheed, 2025): a) Risk of negative transfer, which may occur when the difference between the simulation and the real-world environments is large, b) Training instability, where the process of defining multiple hyper-parameters in the simulation environment renders the whole training process sensitive to their selection, and c) Lack of predictability, where it is difficult to assess the real-world situations where the trained robot might succeed or fail.

The most common DA techniques in robotics are:

- Sim-to-real transfer: This aims at training a FM model using large corpora of simulated/synthetic data, evaluating their performance in such simulation suites, and eventually calibrating them on real-world hardware (typically employing some real-world data). In particular, humanoid and manipulation solutions demonstrate the efficiency of this approach (Luo et al., 2026; Deng et al., 2025a).

- Real-to-sim-to-real transfer: This scenario involves the replay of real-world robot trajectories in high-fidelity simulations, the diversification of the depicted scenes and objects, the synthesis of new training data, the evaluation of the developed models in simulation (often employing domain randomization techniques), and the eventual calibration of the FM model to the real-world specifications. Various works demonstrate the validity of this approach in diverse operational settings (Zhu et al., 2025; Fang et al., 2025b).

### E.5 Imitation learning

The fundamental consideration of Imitation Learning (IL) relies on enabling a model to learn directly from a (human) expert via teleoperation or video demonstrations of the desired skills. The main usefulness of IL lies on its efficient multi-modal alignment, which allows the mapping of high-level language instructions and/or visual inputs directly to low-level motor commands. Its main advantages are (Zare et al., 2024): a) No reward signal definition, where the robot is only required to mimic the expert behavior, b) Reduced training data, where IL is shown to require a reduced number of expert demonstrations to achieve robust performance, and c) Easy training supervision, which is performed directly through teleoperation and/or expert demonstrations. On the contrary, its main limitations are (Kawaharazuka et al., 2025): a) High dependency on data quality, where the quality of the observed demonstrations has a direct impact on the robot learning process, b) Covariate shift, where the robot is highly likely to fail if its actions deviate from the observed demonstrations, and c) Causal confusion, where the model might incorrectly learn unintended factors being present in the demonstrations.

The most common IL techniques in robotics are:

- Behavioral cloning: This serves as a particular type of Supervised Fine-Tuning (SFT), by imitating human expert demonstrations. In particular, RT-1 (Brohan et al., 2023a) learns a direct mapping of observations and language goals to actions, using large-scale transformer networks trained on diverse demonstration data. RT-2 (Zitkovich et al., 2023) extends this with web-scale vision-language pretraining to open-vocabulary, instruction-following control. Moreover, vision FMs for embodiment- and environment-agnostic scene representation further decouple perception from control to facilitate cross-robot transfer (Riou et al., 2024).

- Diffusion-based IL: The fundamental consideration relies on representing a robot's behavior as a conditional denoising process. In this way, diffusion policies treat actions as a data distribution that can be iteratively refined from random noise. In particular, Diffusion Policy (Chi et al., 2025) and Diff-Dagger (Lee et al., 2025) generate action sequences that match expert behavior and handle multi-modal inputs, while improving stability for long-horizon manipulation.

- In-context IL: The main goal is for the robot to learn novel tasks on the fly, i.e., to enable zero- or few-shot task adaptation, but without updating the model weights. ICRT (Fu et al., 2025) instantiates this idea by using next-token prediction over sensorimotor streams for real-robot in-context imitation, extending a similar previous approach that is based on a sequence modeling formalism. Similarly, prompt demonstrations are augmented using explicit visual reasoning traces (Nguyen et al., 2026), allowing the model to infer task intent more reliably in complex and ambiguous environments, while jointly predicting reasoning and low-level actions in an autoregressive way.

- Continual IL: This focuses on addressing the long-term memory and evolution challenges of robotic FMs. The core functionality is to enable a robot to sequentially acquire new skills over time, without forgetting previously learned ones. In this context, LOTUS (Wan et al., 2024) introduces a continual imitation learning framework for skill acquisition by a real robot.

### E.6 Reinforcement learning

Reinforcement Learning (RL) serves as the optimization formalism that aims at bridging the gap between high-level, semantic reasoning (supported by FMs) and low-level, physical robot actions. In practice, the goal of RL is to involve the FM in a continuous, self-improvement cycle of subsequent perception, action, and evaluation steps. Its main advantages are (Tang et al., 2025): a) Self-improvement capability, where RL allows a robot to explore and to surpass the quality of its training data, b) Increased generalization ability, where RL encourages the model to investigate alternative strategies, making it more robust to novel situations, and c) Efficient sim-to-real implementation, where the RL component can be extensively trained in simulation, prior to be deployed in real operational settings. On the contrary, its main limitations are (Ter et al., 2025): a) Sample inefficiency, where RL often requires a very large number of training trajectories to converge, b) Careful reward engineering, which may require detailed definition of the reward function for

avoiding misleads during training, and c) Difficulty in credit assignment, which corresponds to the inherent challenge of identifying incorrect robot behaviors over long-horizon tasks.

The most common RL techniques in robotics are:

- SFT-to-RL: This involves a two-stage process, where BC is initially applied for learning a policy and subsequently RL is employed for further improving it. In this context, RT-1 (Brohan et al., 2023a) and RT-2 (Zitkovich et al., 2023) demonstrate how large-scale BC may produce powerful priors that can be refined further through interaction. Additionally, ExploRLLM (Ma et al., 2025) combines an LLM-guided exploration policy with a residual RL head to improve sample efficiency and robustness.

- LLM-guided reward design: This leverages the capabilities of LLMs for writing/refining the RL reward code and tuning domain randomization procedures, in order to improve the robustness of learning and knowledge transfer. In particular, Eureka (Ma et al., 2024a) automates reward design and outperforms expert rewards on multiple robotic tasks, while DrEureka (Ma et al., 2024b) extends this approach to the sim-to-real setting, by jointly optimizing rewards and randomization for locomotion and dexterous manipulation. Moreover, Gen2Sim (Katara et al., 2024) increases the application scope, by using generative models and LLMs to synthesize tasks, scenes, and reward functions for large-scale RL in simulation.

- Preference-based RL: This relies on replacing detailed/numeric RL rewards with preferences produced by VLMs (or adapted ones, by involving small-scale human intervention). RL-VLM-F (Wang et al., 2024d) models rewards from VLM comparisons over image observations and task-related text, in order to improve manipulation without human guidance. Additionally, VARP (Singh et al., 2025) regularizes VLM-derived preferences with the agent's own rollouts for reducing misalignment and hallucinations in vision-language feedback.

- Offline-to-online RL: This involves a 3-step process, where a) The model is initially trained offline on massive, heterogeneous datasets, b) It subsequently undergoes an offline RL refinement step using an appropriate reward function, and c) Eventually, it follows online RL for real-world deployment. Indicatively, embodied visual tracking is combined with a text-promptable encoder and offline RL for improved perception in (Zhong et al., 2024). Additionally, FLaRe (Hu et al., 2025a) applies large-scale RL fine-tuning on a pre-trained VLA for adaptive manipulation across diverse tasks.

- World-model RL: World-Model Reinforcement Learning (WM-RL) aims at learning a generative world model with language-aware structure and, subsequently, using it for RL-based policy improvement. In this respect, RoboDreamer (Zhou et al., 2024a) factors video generation into compositional parts, conditioned by language and visual goals, and exhibits robust performance on long-horizon tasks.

### E.7 In-context/prompt learning

ICL and prompt learning enable the adaptation of FMs at inference time by conditioning their behavior on demonstrations, examples, or task-specific instructions, without requiring updates to the underlying model weights. While closely related, the two are not identical: ICL relies on task-relevant examples provided within the context window, whereas prompt learning more broadly focuses on steering a pre-trained model through textual or multimodal instructions. Their main advantages are (Fu et al., 2025; Yin et al., 2025c): a) Generalization to novel settings, where a small number of demonstrations or prompts can enable adaptation to previously unseen tasks and environments, b) High-level task planning, where natural-language guidance can be translated into structured reasoning steps and, subsequently, into low-level physical actions, and c) Multimodal task specification, where context can be expressed through language, visual observations, or sensorimotor demonstrations. On the contrary, their main limitations are (Yao et al., 2023): a) Prompt sensitivity, where even minor changes in the input context may lead to unstable or suboptimal behavior, b) Limited grounding, where inference-time reasoning does not inherently guarantee consistency with real-world constraints, and c) Inference overhead, where long context windows and multi-step prompting strategies may increase latency and computational cost during deployment.

The most common in-context/prompt learning techniques in robotics are:

- Language prompting: This relies on the use of natural language instructions for guiding a robot's behavior, decision-making, and physical actions. In particular, few-shot language prompts can encode demonstrations or templates, so that an LLM can output low-level actions or to acquire new skills (Yin et al., 2025c; Liang et al., 2023; Huang et al., 2023b).

- Reason-act prompting (ReAct): This interleaves natural language reasoning with physical actions, allowing a robot to decompose complex goals, to validate its progress, and to dynamically adjust its plan. In this respect, planning, execution, and re-planning can be performed in a single loop, also involving an LLM-based verification checking step (Yao et al., 2023; Grigorev et al., 2025).

- In-context imitation: This enables a FM to perform a novel task by observing a few videos or sensorimotor demonstrations, without applying any permanent changes to its internal weights. In this context, a causal policy can parse short teleoperation trajectories as a prompt and, subsequently, to predict the next action for new tasks without the need for fine-tuning (Fu et al., 2025).

## E.8 World model learning

The fundamental functionality of World Models (WMs) in robotic FM applications is that they allow robots to predict environmental changes in response to their actions. In particular, their primary role is to decouple perception from action, where, instead of directly operating only on pixel values, the robot can learn the underlying physics of the world. Their main advantages are (Li et al., 2025d): a) Incorporation of 'imagined' experiences, which reduces the need for physical-world trials, b) Modeling rules of physics, where the model learns the impact of factors like gravity, friction, and collisions in the real environment, and c) Reduced delays in operation, where the robot is able to predict future states and to maintain smooth actions. On the contrary, their main limitations are (Zhang et al., 2025e): a) Presence of hallucinations, where the robot may converge to misleading actions in case of slight inaccuracies in the WM, b) Cumulative errors, where small prediction errors can be accumulated in long-horizon tasks, and c) Increased computational cost, where the training of a high-fidelity WM may require excessive GPU resources.

The most common WM learning techniques in robotics are:

- Feature-space WMs: Instead of performing a prediction for each pixel, which is computationally expensive and often noisy, feature-space WMs map visual inputs to an abstract feature space and perform future predictions there. In particular, future DINOv2 patch embeddings are predicted from offline trajectories and, subsequently, action sequences are optimized in the embedding space for zero-shot planning (Zhou et al., 2025b).

- Latent-action WMs: Latent-action WMs aim at learning robots to model the underlying physics and intent of actions, instead of focusing on representing specific skills. In this respect, continuous latent actions are discovered from videos, while an auto-regressive WM is trained that conditions on those actions to transfer skills across scenes and embodiments with small-scale finetuning (Gao et al., 2025).

- Compositional video WMs: These adopt a modular approach, where the model aims at breaking down (factorizing) the surrounding environment into its constituent parts (namely, objects, relationships, and action primitives) and, subsequently, reconnecting them for generating future scenarios. In this context, videos are factorized into objects and relations so that the model can synthesize plans for unseen combinations of goals-scenes and guide long-horizon decisions (Zhou et al., 2024a).

- JEPA-style WMs: The main focus lies on predicting an abstract meaning of what will happen next, which enables robots to plan complex tasks, without getting distracted by irrelevant noise. In particular, joint predictions of short-horizon actions and abstract observations are realized (Vujinovic & Kovacevic, 2025), in order to couple imitation with predictive learning and to reduce control error accumulation.

### E.9 Generative learning

Generative Learning (GL) enables robots to imagine future states, to synthesize training data, and to propose complex action sequences. Its fundamental use lies on the capability of generative FMs of producing large quantities of data samples, alleviating from the need for extensive high-quality robotic interaction samples. Its main advantages are (Zhang et al., 2025d): a) Increased zero-shot generalization, where generative models are suitable for handling objects or environments previously unseen, b) Handling multi-modality, due to the ability of generative models of predicting missing information between different sensorial data, and c) Long-horizon planning, where generative models enable robots to perform accurate predictions for multiple future steps. On the contrary, its main limitations are (Liu et al., 2025a): a) Sim-to-real gap, where the generated data can miss subtle physical nuances, b) Presence of hallucinations, where generative robots are prone to hallucinating a physical capability, and c) Increased inference latency, where generative models typically require extensive computations.

The most common GL techniques in robotics are:

- Autoregressive sequence modeling: The main goal lies on predicting the next action or state based on previous observations. In this respect, PACT (Bonatti et al., 2023) trains a causal transformer to predict the next observation-action token, so that a single model can capture long-horizon structure across different tasks. Additionally, long-horizon manipulation is modeled through sequential generation of action tokens (Zhang et al., 2025h).

- Diffusion-based action policies: This employs a diffusion model for generating a chunk of actions at once, by gradual application of a denoising process. In particular, Diffusion Policy (Chi et al., 2025) learns a conditional denoising process that samples action sequences for handling multi-modal behaviors and supporting stable visuomotor control. Similarly, Legibility Diffuser (Bronars et al., 2024) comprises an intent-expressive variant of the latter.

- Generative video and scene synthesis: The fundamental aim comprises the creation of a model of the physical reality, which in turn enables robots to imagine, to simulate, and to plan actions prior to their execution in the real world. In this context, RoboDreamer (Zhou et al., 2024a) employs compositional video WMs, while ReBot (Fang et al., 2025b) makes use of a real-to-sim-to-real synthesis approach.

## F Learning stages

This section provides the full details of the different learning stages involved in the development of robotic FM methods, complementing the condensed presentation of Section 8 of the main paper. For each learning stage (namely, pre-training, offline fine-tuning, online adaptation, and continuous learning), the complete category definition, the corresponding main advantages and limitations, as well as the extensive list of representative methods per adopted learning paradigm are provided below.

### F.1 Pre-training

The ultimate goal of the pre-training stage is to estimate robust, general-purpose representations of robotic data. In particular, instead of training a robotic agent for a specific task or application, pre-training aims at processing massive (often internet-scale) amounts of diverse data (e.g., human demonstration videos, simulation data, sensorial data streams, etc.) from multiple robotic platforms, in order to acquire knowledge and to model the cross-correlations among vision, language, and action. Its main advantages are (Li et al., 2024a): a) Increased generalization, where pre-trained models are more likely to adapt to new environments, than undergoing training from scratch, b) Robust zero-shot capability, where pre-trained models can often perform robustly tasks that they weren't explicitly trained for, without the need for extra training samples, and c) Accurate multi-modal mapping, where due to the usual large-scale datasets used, pre-trained models are proven to robustly map across textual words, visual concepts, and physical actions. On the contrary, its main limitations are (Kawaharazuka et al., 2025): a) Excessive computational cost, where pre-training requires massive GPU resources to be implemented, b) Reduced performance in real-world, where if a model is

pre-trained largely on internet or simulation data, it may not always perform robustly in real-world, physical circumstances, and c) Safety concerns, where training using internet resources does not always take into consideration strict safety constraints.

The most common learning paradigms adopted during the pre-training stage are:

- Supervised learning: Supervised learning aims at equipping a model with a foundational understanding of the world from high-quality, diverse, labeled data. In particular, PaLM-E (Driess et al., 2023) is jointly trained using both web-scale multi-modal data and embodied experiences. Similarly, RT-2 (Zitkovich et al., 2023) and Gemini Robotics (Team et al., 2025) are constructed using web and robot manipulation data.

- Self-supervised learning: SSL targets to enable robots to operate beyond narrow, task-specific programming, aiming at accomplishing generalized intelligence capabilities. In particular, DINOv2 (Oquab et al., 2024) and VideoMAE (Tong et al., 2022) can learn perception priors from unlabeled data. Moreover, R3M (Nair et al., 2023) extends this approach, by incorporating egocentric features.

- Imitation learning: The goal of IL is to learn a prior distribution of successful behaviors directly from expert demonstrations. In particular, RT-1 (Brohan et al., 2023a) treats robot control as a next-token prediction problem over multi-modal streams, so that the learned policies to inherit both semantic and sensorimotor skills. Additionally, RT-2 (Zitkovich et al., 2023) is trained using vision-language and robot action tokens, while Octo (Mees et al., 2024) makes use of the Open-X-Embodiment trajectories (O'Neill et al., 2024) for deriving a generalist policy. Moreover, Seer (Tian et al., 2025) investigates the scaling laws of multi-task IL, aiming at drastically reducing the required target-domain data.

### F.2 Offline fine-tuning

Offline fine-tuning aims at bridging the knowledge gap between general-purpose representations (learned during the pre-training step) and the specificities of a given physical-world application. In short, the primary purpose of this stage is task and embodiment specialization. Its main advantages are (Hu et al., 2023): a) Reduced need for training data, where the need for training samples is significantly lower than the pre-training step, b) Increased training stability, where models are likely to converge to robust performance states, provided that sufficient training examples are available, and c) Knowledge distillation, where only the necessary parts of the general-purpose representations learned during the pre-training stage can be used for the given application at hand. On the contrary, its main limitations are (Firoozi et al., 2025): a) Distribution shift, where if a robot encounters novel state challenges, it is difficult for it to handle, b) Dependency on data quality, where the presence of suboptimal or erroneous demonstrations can mislead the training process, and c) Lack of online exploration, where the model can only employ skills present in the training dataset, while not being able to adapt to online challenges.

The most common learning paradigms adopted during the offline fine-tuning stage are:

- Imitation learning: IL aims at equipping pre-trained models with the necessary low-level precision skills for a specific application. In particular, by mimicking expert demonstrations, it targets to learn specific motor commands with respect to a given task or robotic platform. In particular, Octo (Mees et al., 2024) and OpenVLA (Kim et al., 2025) employ large-scale Open X Embodiment pretraining and, subsequently, focus on new robotic platforms, making use of LoRA-style adapters. Similarly, LiteVLA (Williams et al., 2025) demonstrates that NF4 quantized LoRA can be tuned on CPU-only hardware.

- Reinforcement learning: RL enables robots to learn from a reward signal, by focusing on performed actions that lead to successful task executions. In particular, a recurrent tracker is trained, using conservative offline RL, on VFM annotated trajectories, prior to deployment (Zhong et al., 2024). Additionally, FLaRe (Hu et al., 2025a) applies large-scale RL fine-tuning to transformer-based policies, in order to improve long-horizon mobile manipulation.

- Generative learning: GL facilitates the specialization of pre-trained models to specific environments, especially in the presence of sparse constraints on the target task. In particular, ReBot (Fang et al., 2025b) replays real trajectories in simulation and, subsequently, composes them into inpainted real backgrounds to adapt to new domains. Additionally, RoboDreamer (Zhou et al., 2024a) makes use of compositional WMs to generate imagined video plans, which serve as additional training data.

### F.3 Online adaptation

Online adaptation targets to equip robots with the appropriate routines for learning in real-time from their own experiences. In practice, at this stage robots aim to handle the distribution shift between the offline training data and the ones encountered during online deployment. Its main advantages are (Firoozi et al., 2025): a) High precision performance, where robots typically accomplish superior task execution accuracy for the specific environments to which they are adapted, b) Continuous improvement, where the robot continuously increases its performance as it constantly learns from its experiences, and c) Robustness to distribution shifts, where the models achieve to maintain performance in the presence of environmental changes. On the contrary, its main limitations are (Yuan et al., 2025): a) Catastrophic forgetting, which denotes the risk that might occur as the robot learns new skills to lose some of its general-purpose ones, b) Computational latency, where the algorithmic operations involved need to be performed in (near) real-time, and c) Prone to noise, where real-time, noisy sensorial data may mislead the adaptation process.

The most common learning paradigms adopted during the online adaptation stage are:

- Domain adaptation: This aims at handling the particular physical-world constraints and sensorial noise for a specific robot deployment scenario. In practice, this enables the model to re-calibrate its knowledge structures to the perceived environment. In this context, Test-Time Adaptation (TTA)-Nav (Piriyajitakonkij et al., 2024) incorporates a reconstruction decoder on top of a pre-trained policy, so that the agent can denoise corrupted frames without gradient updates, while restoring point-goal navigation under severe corruptions. Additionally, Phys2Real (Wang et al., 2025b) targets to bridge sim-to-real gaps, by combining FM priors with interaction-based estimations.

- Reinforcement learning: The aim of RL lies on allowing the robot to perform micro-adjustments to its general-purpose knowledge, based on sensory feedback and exploration. In this context, self-improving embodied FMs refine pre-trained policies, based on reward and success estimation from model predictions, across a robot fleet (Ghasemipour et al., 2025). Additionally, RL-VLM-F (Wang et al., 2024d) estimates rewards, using a VLM model that compares trajectory snippets with language goals. Similarly, VARP (Singh et al., 2025) regularizes VLM-derived preferences with the agent's own rollouts for reducing misalignment. Moreover, Eureka (Ma et al., 2024a) and DrEureka (Ma et al., 2024b) are LLM-guided RL frameworks that automate reward design and, in the case of DrEureka, domain randomization, in order to improve policies for locomotion and dexterous manipulation.

- In-context/prompt learning: This paradigm is particularly suitable for online adaptation, since it enables zero- or few-shot specialization at deployment time, without requiring updates to the underlying model weights. In particular, ICRT (Fu et al., 2025) employs in-context imitation policies that condition on a small number of recent demonstration trajectories for adapting to novel manipulation tasks. Additionally, LLM-based control stacks can interleave reasoning and acting through ReAct-style prompting, allowing robots to monitor execution progress and to revise plans when needed (Yao et al., 2023). Similarly, verification-based frameworks can check high-level task plans prior to execution, improving reliability in dynamic deployment settings (Grigorev et al., 2025).

### F.4 Continuous learning

Continuous Learning (CL) (often termed lifelong learning) aims at enabling robots to acquire new skills or to adapt to new environments incrementally, without requiring to be fully retrained from scratch. In practice, it employs conventional learning paradigms (e.g., IL, RL, etc.) in slower outer loops for improving the robot's performance. Its main advantages are (Xiao et al., 2025b): a) Increased adaptability, where robot

capabilities can continuously evolve in response to changes in the surrounding environment, making them suitable for long-term deployment, b) Improved scalability, where data from a single robot can be used to update the knowledge structures of an entire fleet, and c) Reduced downtime, where robots that continuously update their underlying models are less likely to become non-sufficiently robust/operational for long periods of time. On the contrary, its main limitations are (Firoozi et al., 2025): a) Catastrophic forgetting, where the continuous update in the robot's knowledge structures may result into overwriting previously learned skills, b) Stability-plasticity dilemma, which corresponds to a critical trade-off between the ability to integrate new skills and the capability to maintain the old ones, and c) Memory overhead, where the model needs to maintain increased past data for robustly updating its behavior in subsequent steps.

The most common learning paradigms adopted during the continuous learning stage are:

- Domain adaptation: DA enables a model to continuously adjust its internal knowledge to dynamic, real-world settings. In particular, Action Flow Matching for Lifelong Learning (Murillo-González & Liu, 2025) develops a lifelong robot-learning framework that incrementally aligns robot dynamics across sequential tasks, supporting efficient and safe continual adaptation. On another direction, VR-Robo (Zhu et al., 2025) utilizes a real-to-sim-to-real framework for constructing photorealistic digital twins from logged data, retraining navigation or locomotion policies in them, and transferring the acquired skills to the real-world.

- Imitation learning: IL aims at constantly bridging the gap between a model's general knowledge and the specific, high-precision requirements of a real-world application case, on the condition of the availability of few expert demonstrations. In particular, SkillsCrafter (Wang et al., 2026) realizes lifelong language-conditioned robot learning across multiple sequential manipulation skills, while reducing catastrophic forgetting through symbolic skill distillation. Additionally, LOTUS (Wan et al., 2024) models and refines manipulation skills from demonstration streams, supporting long-term expansion of its skill repertoire.

- Reinforcement learning: RL supports continuous learning by enabling FM robotic policies to improve through repeated interaction, autonomous practice, and reward-driven post-training over extended deployment horizons. In particular, self-improving embodied FMs refine pretrained policies through autonomous practice based on self-predicted rewards and success signals across a robot fleet, thereby enabling downstream skill acquisition with minimal human supervision (Ghasemipour et al., 2025). Similarly, LiReN (Stachowicz et al., 2024) demonstrates that navigation FMs can improve lifelong learning, through the employment of online RL pipelines, in open-world settings. More recently, VLA models have been adapted through reinforcement fine-tuning, showing improved retention and adaptation to new tasks, while mitigating catastrophic forgetting in sequential manipulation tasks (Liu et al., 2026b).

## G Robotic tasks

This section provides the full details of the different robotic tasks for which FM-based solutions have been developed, complementing the condensed presentation of Section 9 of the main paper. For each robotic task (namely, perception, planning, navigation, manipulation, and human-robot interaction), the complete category definition, the corresponding main advantages and limitations, as well as the extensive list of representative methods per sub-category are provided below.

### G.1 Perception

Perception aims at creating rich, semantic maps of the surrounding environment, which subsequently enable robots to execute individual actions. In particular, robot perception enables the realization of semantic grounding (i.e., the connection of visual stimuli with real-world entities), discovery of object affordances (i.e., the tasks that can be performed with different objects), and contextual awareness (i.e., the identification of the different types of semantic entities and their location). The main advantages of the incorporation of FMs in robot perception are (Kawaharazuka et al., 2024): a) Open-vocabulary recognition, where models

can identify entities for which they are not specifically trained for, b) Zero-shot generalization, where robots can handle novel environments or object types, c) Multi-modal fusion, where robotic agents can efficiently combine multiple information streams (e.g., visual, language, proprioception, etc.), and d) Robustness to noise, where FMs are shown to be reliable in the presence of noisy data. On the contrary, the main limitations of the integration of FMs in robot perception are (Hu et al., 2023): a) High latency, due to the typical extreme scale of the underlying models employed, b) Decreased explainability, where the main factors leading to a particular robot decision is difficult to be precisely defined, c) Presence of hallucinations, where model predictions can be misled and to result in failures in the physical world, and d) Spatial imprecision, which relates to inaccurate localization of (even correct) entity predictions.

The main categories of perception methods are:

- Language-grounded detection and segmentation: This aims at identifying (detection) and precisely outlining (segmentation) the objects present in the robot's surrounding environment based on natural language prompts. In particular, GLIP (Li et al., 2022) and Grounding DINO (Liu et al., 2024c) provide phrase-based detections that remain robust in the presence of clutter as well as in the zero-shot setting; such detections can subsequently feed language-conditioned manipulation and navigation pipelines, such as CLIPort (Shridhar et al., 2022) and similar solutions (Unlu et al., 2024; Hao et al., 2025). Additionally, SAM-style promptable segmenters (Kirillov et al., 2023; Ravi et al., 2025; Carion et al., 2026) allow the incorporation of box, click, or text prompt information into control pipelines in an interactive way, like SAM-6D (Lin et al., 2024) for 6D pose estimation and RoG-SAM (Mei et al., 2025) for instance-level robotic grasping detection.

- Open-vocabulary 3D semantic mapping: This allows robots to perceive and to localize objects of previously unseen categories in the 3D space, making use of natural language inputs. In this respect, ConceptFusion (Jatavallabhula et al., 2023) builds open-set, language-searchable maps that support uncommon and previously unseen concepts, and multi-modal queries. Additionally, ConceptGraphs (Gu et al., 2024) estimates object nodes and their relations, so that task planners can subsequently operate on top of a semantic scene graph (instead of raw pixels). Moreover, radiance-field (e.g., LERF (Kerr et al., 2023)) and 3D-Gaussian (e.g., FMGS (Zuo et al., 2025)) grounded approaches embed CLIP/DINO features into neural fields for estimating consistent, view-invariant labels. Furthermore, OpenFusion++ (Jin et al., 2025), OpenVox (Deng et al., 2025b), and MR-COGraphs (Gu et al., 2025) focus on real-time, open-vocabulary voxel mapping and multi-robot scene graphs for efficient robot exploration.

- Pose estimation and affordance prediction: Pose estimation aims at aligning an object's local coordinate system to the global, world one, while affordance prediction aims at the detection of the ways that an object can be manipulated; in both cases, FMs are particularly useful due to their inherent ability of linking semantics (language) with spatial geometry (pixels). In particular, OV9D (Cai et al., 2024a) estimates category-agnostic 9-DoF pose and object size without relying on the use of CAD models. Similarly, Oryon (Corsetti et al., 2024) aligns CLIP-guided segments across different views, in order to recover relative 6D pose for unseen objects. On the other hand, OpenAD (Nguyen et al., 2023) models zero-shot 3D affordances in a shared vision-language embedding space, while OVA-Fields (Su et al., 2025) extends this direction to weakly supervised open-vocabulary affordance fields for robot operational part detection in 3D scenes. Moreover, one-shot open affordance learning transforms a single example into dense, class-agnostic affordance masks (Li et al., 2024b).

- Contact-centric and visuotactile perception: This category of methods focuses on analyzing and modeling the physics of robot interactions for enhancing perception. In particular, contact-centric approaches consider junction points as the primary state representation, while visuotactile ones integrate exocentric vision with local tactile sensing. In this respect, TLA (Hao et al., 2026) and Tactile-VLA (Huang et al., 2025b) combine tactile information streams with vision and language, in order to improve insertion, assembly, and material reasoning tasks. Large tactile-vision-language models (e.g., TALON (Jiang et al., 2024)) further extend this idea to richer contact semantics. Moreover, visuotactile systems, such as NeuralFeels (Suresh et al., 2024), can enhance in-hand pose and shape estimation, when visual cues are uncertain.

- Long-term object tracking: This aims at locating a given object or point across a sequence of video frames, while maintaining prediction stability over time and capability to recall objects when they become occluded or exit the field of view. The latter is a particular requirement for long-horizon tasks. In this respect, OVTrack (Li et al., 2023b) employs language and diffusion priors to generalize multi-object tracking to unseen categories without explicit video pre-training. Similarly, DINO-MOT (Lee et al., 2024) combines DINOv2 features with a memory mechanism for robust pedestrian tracking, while COVTrack (Qian et al., 2025b) further strengthens open-vocabulary tracking through improved temporal association across continuous trajectories.

## G.2 Planning

Planning serves as the fundamental bridge between high-level (semantic) reasoning and low-level motor control procedures. In particular, the primary usefulness of robot planning lies in long-horizon task decomposition, where a high-level task goal needs to be broken down into multiple, sequential, individual sub-goals over an extended period, prior to their actual execution. The main advantages of the incorporation of FMs in robot planning are (Hu et al., 2023): a) Increased interpretability, where FMs enable a planned sequence of robot actions to be represented in human-like form, b) Increased generalization ability, where robots can often adapt to novel settings by leveraging general-purpose, world knowledge stored in a FM, c) Reduced need for training data, where pre-trained FMs exhibit a decreased need for large-scale, expensive, visuomotor robot data, and d) Safety contraints integration, where FMs enable the efficient incorporation of safety constraints directly into the planning loop. On the contrary, the main limitations of the integration of FMs in robot planning are (Firoozi et al., 2025): a) Logical gaps, where FM-based planners may estimate action steps that are not physically possible, b) Lack of grounding, where FM inference may result into deviations between a high-level plan and the low-level capacities of the robot at hand, c) Increased latency, where the high-computing nature of FMs may be proven restrictive for high-pace tasks, and d) Increased closed-loop complexity, where FM-based solutions face challenges in adjusting their behavior in real-time settings, as a response to constant environmental changes.

The main categories of planning methods are:

- Language-driven task decomposition: This aims at breaking down high-level, long-horizon goals into logical sequences of robot primitive actions (atomic skills), often in the form of structured, executable code (program synthesis). In particular, SayCan (Brohan et al., 2023b) and SayPlan (Rana et al., 2023) ground each action step on an affordance map or a 3D scene graph, respectively, so that abstract sub-goals to correspond to concrete objects and locations. On the other hand, Code-as-Policies (Liang et al., 2023) and similar approaches generate directly short, Python-like programs for integrating existing software libraries, rendering planning easier to inspect, to test, and to modify (Singh et al., 2023; Huang et al., 2023c).

- Neuro-symbolic closed-loop reasoning: This category of methods combines the general-purpose knowledge of a FM with formal, logical checking procedures, so as to guarantee the successful operation of robot agents in the real world. In this respect, ISR-LLM (Zhou et al., 2024b) converts natural language instructions to PDDL ones and iteratively refines the estimated plans using a symbolic validation scheme, until a feasible sequence of actions is determined. Additionally, AutoTAMP (Chen et al., 2024b) employs an LLM for generating or translating high-level language instructions into intermediate representations suitable for a task-and-motion planning (TAMP) solving mechanism, while making use of autoregressive re-prompting to correct syntactic and semantic errors. Moreover, safety- and feasibility-oriented planners, such as SELP (Wu et al., 2025b) and LLM-GROP (Zhang et al., 2025f), translate LLM proposals into explicit constraint checking and task-and-motion reasoning procedures, in order to avoid unsafe or dead-end policies.

- Multi-modal policy generation: This category focuses on generating high-level plans or robot actions by integrating multiple input modalities, such as language, vision, and embodied state information. In particular, PaLM-E (Driess et al., 2023) combines language, vision, and proprioception for embodied reasoning and action generation, while RT-2 (Zitkovich et al., 2023) and OpenVLA (Kim et al., 2025)

demonstrate that language-aligned visual representations can transfer web-scale semantic knowledge to real-world robot control.

- Execution-time validation and failure recovery: This category focuses on monitoring generated plans during deployment, verifying action preconditions, detecting inconsistencies or failures, and triggering corrective re-planning when needed. In this respect, Code-as-Monitor (Zhou et al., 2025a) introduces constraint-aware visual programming for reactive and proactive robotic failure detection, SELP (Wu et al., 2025b) incorporates explicit safety and feasibility constraints into the planning loop, while VLM-based monitoring frameworks, such as Guardian (Pacaud et al., 2025) and unified real-time failure-handling approaches (Ahmad et al., 2025), support execution-time failure detection and recovery in robotic manipulation.

- Semantic multi-robot coordination: This category of methods capitalizes on the broad knowledge base of FMs for coordinating multi-robot setups, where accurate reasoning about task dependencies, resources, and scheduling is needed. In this direction, LiP-LLM (Obata et al., 2024) employs an LLM to create a skill list and a corresponding dependency graph from language instructions, while subsequently relying on linear programming techniques to allocate tasks across robotic platforms. Additionally, SMART-LLM (Kannan et al., 2024) assigns agents task-specific roles based on structured representations of their skills and capabilities, reducing in this way instruction drift and enabling coalition formation and task allocation, based on a single high-level instruction. Moreover, large-scale orchestration systems, such as AutoRT (Ahn et al., 2024), combine language-based task assignment with human oversight and monitoring, demonstrating that FM-based planners can coordinate dozens of physical robots in real-world environments.

## G.3 Navigation

The fundamental usefulness of FMs in robot navigation lies on providing the necessary spatial common-sense knowledge in embodied AI settings. The latter is mainly accomplished due to the capacity of FMs to process the robot's surrounding environment as a high-level semantic space, instead of considering rigid, inflexible spatial maps. The main advantages of the incorporation of FMs in robot navigation are (Pan et al., 2025b): a) Open-world navigation, where robots can operate in environments including previously unseen entities, b) Cross-embodiment transfer, where a single pretrained model can be deployed in different/diverse hardware platforms, and c) Semantic reasoning capability, which relies on the inherent ability of FMs to combine vision with language understanding for instruction interpretation. On the contrary, the main limitations of the integration of FMs in robot navigation are (Firoozi et al., 2025): a) Sim-to-real gap, where FMs trained in simulation may exhibit difficulties in operating in real-world environments, b) Lack of training data, where large-scale, high-quality, diverse 3D navigation data, required for FM training, is difficult and expensive to collect, c) Increased latency, where the high computational needs of FMs can lead to challenging situations in real-world applications, and d) Safety and ethical concerns, where FMs need to be equipped with appropriate social norms for operating in human-shared spaces.

The main categories of navigation methods are:

- Semantic spatial grounding: This category of methods aims at registering the objects present in the environment, but also their relative position (with respect to the robots) and a concrete action plan for reaching them (in natural language form). In particular, VLMaps (Huang et al., 2023a) builds CLIP-indexed spatial memories that enable the system to query objects and rooms without task-specific retraining, directly ingesting visual-language features into a 3D map. Zero-shot localization schemes, such as PixNav (Cai et al., 2024b) and VLTNet (Wen et al., 2025a), guide navigation through pixel-level target cues or construct semantic navigation maps and rank exploration frontiers from language prompts. Moreover, open-vocabulary mapping methods (like One Map to Find Them All (Busch et al., 2025), DualMap (Jiang et al., 2025b), and scene graph-based approaches (Loo et al., 2025)) support multi-object navigation, dynamic environments, and functional queries, so that a single map can accommodate for many language goals and robots.

- Instruction-following policies: This is based on the increased capability of FMs in interpreting natural language instructions, without requiring a pre-defined map or hard-coded scripts for every possible object present in the environment. In this respect, generalist navigation models, such as GNM (Shah et al., 2023b) and ViNT (Shah et al., 2023c), formalize navigation as a sequence prediction problem over images and poses, relying on a single model across different robots and environments. Additionally, NaviLLM (Zheng et al., 2024a) unifies instruction following and embodied QA with schema-tuned prompts, while NavFormer (Wang et al., 2024a) learns target-driven policies in unknown, dynamic environments. Moreover, FASTNav (Chen et al., 2024c) demonstrates that compact, LoRA-adapted language models can operate in real-time on embedded hardware, offering a practical path from large offline training to on-board controllers.

- End-to-end policies: This aims at developing a unified architecture that can map raw sensorial data directly to motor commands, instead of adopting the conventional modular design of breaking down navigation into individual steps (like mapping, localization, and path planning). In particular, ViNT (Shah et al., 2023c) learns a generalizable visuomotor navigation policy across multiple robots and environments, while NavFoM (Zhang et al., 2026) extends this idea towards cross-embodiment and cross-task navigation. In driving-oriented settings, end-to-end frameworks, such as DiffusionDrive (Liao et al., 2025) and DriveGPT-4 (Xu et al., 2024b), further demonstrate that multi-modal models can predict low-level control signals directly from visual- and language-conditioned inputs.

## G.4 Manipulation

The primary utility of FMs in robot manipulation lies on providing the required knowledge for implementing the translation from high-level, semantic, human-like instructions to precise physical pressures and movements needed to manipulate an object. The main contribution of FMs for achieving the latter comprises their cross-embodiment learning capability, where the same model pretrained on data from multiple different platforms can be deployed to diverse setups. The main advantages of the incorporation of FMs in robot manipulation are (Li et al., 2024a): a) Increased generalization, where FMs are shown to be robust in handling previously unseen object types, b) Improved robustness, where FMs can make use of real-time (visual) feedback to adjust their manipulation strategy on the fly, and c) Enhanced embodiment capability, where FM solutions enable the understanding of the physical properties of the objects, prior to their manipulation. On the contrary, the main limitations of the integration of FMs in robot manipulation are (Sapkota et al., 2025): a) Limited training data, where sufficient quantities of high-quality, labeled robotic manipulation data is difficult to collect, b) Safety concerns, where the robot policies are not always guaranteed to result into feasible and safe manipulations, and c) Low action precision, where the increased generalization ability of FMs is accompanied with corresponding decrease in physical task execution for specialized domains.

The main categories of manipulation methods are:

- Language-to-action models: This category aims to interpret the semantic meaning of a natural language command and to map it to the physical world setting, without requiring task-specific programming. In particular, RT-1 (Brohan et al., 2023a) and RT-2 (Zitkovich et al., 2023) approach manipulation as a sequence modeling problem over multi-modal tokens, while PaLI-X (Chen et al., 2024a) aims at incorporating broad visual knowledge. Additionally, OpenVLA (Kim et al., 2025) comprises a large vision-language-action policy that adapts to new robotic platforms, based on small-scale fine-tuning. GR00T N1 (Bjorck et al., 2025) combines a deliberative VLM with a diffusion motor policy for bimanual humanoid skills acquisition. Moreover, state-space variants, such as RoboMamba (Liu et al., 2024b), replace the core transformer component with a Mamba-based selective state-space model for lowering latency, while maintaining increased visuomotor skill performance.

- Retrieval-augmented imitation learning: This relies on receiving guidance from a large database of relevant previous demonstrations for predicting future actions. In this context, DINOBot (Di Palo & Johns, 2024) detects similar demonstrations based on DINO feature correspondence and subsequently estimates dense manipulation trajectories for realizing one- and few-shot generalization to novel objects. Additionally, STRAP (Memmel et al., 2025) retrieves relevant manipulation sub-trajectories from

prior demonstrations and uses them to augment few-shot imitation learning, improving generalization to novel objects and tasks.

- Constraint-aware policy synthesis: This category employs a FM to generate high-level control code or mathematical objectives, which are explicitly bounded by physical, safety, and environmental constraints. In this context, CoPa (Huang et al., 2024) detects task-relevant parts using a multi-modal LLM and estimates spatial constraints that a conventional planner translates into 6-DoF actions. ReKep (Huang et al., 2025c) represents tasks as sequences of relational keypoint constraints and optimizes them hierarchically for single- and dual-arm assembly tasks. Moreover, part-centric perception methods, like PartSLIP++ (Zhou et al., 2025d), estimate part-affordance correlations that are required for robust task execution.

- Semantic spatial maps: This category aims at generating representations of the environment that combine 3D spatial geometry with semantic information about the involved objects. In this direction, VoxPoser (Huang et al., 2023c) estimates constraints and object affordances from natural language inputs, creates 3D value maps from vision-language information cues, and applies common motion planning routines in a zero-shot fashion. Additionally, AdaRPG (Zhang et al., 2025g) leverages VLMs to infer part affordances and operational constraints that guide primitive skills for articulated-object manipulation.

## G.5 Human-robot interaction

The fundamental usefulness of FMs in HRI is grounded on their increased ability for realizing semantic, human-like reasoning and interpreting human intent. In particular, robots are capable of reacting to conversational instructions, asking clarification questions, understanding contextual settings, and providing explanations for their actions, which greatly simplifies communication (especially) with non-expert users. The main advantages of the incorporation of FMs in HRI are (Zhao et al., 2025c): a) Rapid generalization, where a human user can demonstrate a new task to a robot and it can adapt instantly, b) Intuitive control, where due to the increased human behavior interpretation capabilities of FMs, robot control becomes more efficient and intuitive, c) Increased safety alignment, where user-provided feedback can boost robots to learn/incorporate social norms, and d) Efficient error recovery, where human provided feedback can be rapidly exploited by a robot for recovering from a fault state. On the contrary, the main limitations of the integration of FMs in HRI are (Xiao et al., 2025b): a) Lack of training data, where data collection involving human feedback/interactions is typically costly, b) Incorporation of human bias, where robots learn in an unconstrained way from their human-provided feedback, and c) Semantic drift occurrences, where human intent is not always easy to interpret as contextual/environmental conditions may change.

The main categories of HRI methods are:

- Conversational policy alignment: This category of methods focuses on using natural language dialogue to dynamically adjust a robot's behavior in real-time, so that it matches a human's specific intent, preferences, and safety boundaries. In particular, DRAGON (Liu et al., 2024d) comprises a dialogue-based navigation framework that grounds free-form commands in visual landmarks, describes the environment, and asks clarification questions when the reference is unclear. Additionally, PlanCollabNL (Izquierdo-Badiola et al., 2024) translates spoken instructions into editable plans so that users can insert, remove, or reorder steps in collaborative manipulation and assembly settings.

- Reciprocal social tuning: This aims at developing a high-level, semantic communication framework, where both humans and robots continuously adjust their behaviors, social cues, and expectations to harmonize each other. In this context, incremental system updates combine natural-language feedback with on-robot sensing so that the controller can refine prompts, code snippets, or skill graphs after each mistake and also to reuse the acquired knowledge at a later stage (Bärmann et al., 2024). Additionally, design studies on conversational companion robots provide concrete guidelines for everyday HRI settings, such as clear grounding, turn-taking, and repair under uncertainty (Irfan et al., 2024). TidyBot (Wu et al., 2023a) demonstrates that LLMs can learn user-specific preferences for household clean-up from a few examples and then generalize these preferences to new objects and

scenes. Similarly, LAMS (Tao et al., 2025) extends this idea to assistive teleoperation, by using an LLM to switch control modes based on task context and improving performance over time based on user feedback.

- Active alignment and mitigation: This targets to ensure that a robot remains synchronized with human intent, while proactively handling errors or deviations. In this respect, RoboVQA (Sermanet et al., 2024) queries egocentric video to check preconditions, to verify whether a sub-task has succeeded, and to proactively request human intervention in case of identified failures. Additionally, embodied LLM controllers also incorporate human feedback for adjusting their plans when they detect that the current state drifts from the expected goal (Mon-Williams et al., 2025). Overall, the common practice relies on combining a latency-aware VLM with a dialogue-based manager, so that the robot can provide concise explanations, ask specific clarifications, and implement targeted re-planning (Bärmann et al., 2024; Sermanet et al., 2024).

# H   Application domains

This section provides the full details of the different real-world application domains in which robotic FM-based solutions have been deployed, complementing the condensed presentation of Section 10 of the main paper. For each application domain (namely, agentic mobility, industrial manipulation, supply operations, service robots, medical robots, cognitive agrisystems, crisis agents, maritime robotics, and space robotics), the complete domain definition, as well as the extensive list of representative methods are provided below.

## H.1   Agentic mobility

The integration of FMs has induced a paradigm shift in autonomous movement, transitioning from rigid path-following routines to autonomous agentic solutions, where robots leverage high-level reasoning to understand mission objectives and to adapt to environmental changes.

A core application of these models is in semantic spatial grounding, which replaces traditional, inflexible spatial maps with queryable memories. For instance, VLMaps (Huang et al., 2023a) constructs CLIP-indexed spatial memories by directly ingesting visual-language features into 3D maps; this allows robotic systems to navigate to specific rooms or objects via natural language queries, without requiring task-specific retraining. Similarly, methods like EffoNAV (Shen et al., 2025) pair CLIP-based goal detection with exploration routines and low-level controllers to handle visual targets in challenging settings.

FMs also address the per-robot engineering bottleneck through cross-embodiment and cross-site generalization. Generalist models, such as GNM (Shah et al., 2023b) and ViNT (Shah et al., 2023c), formalize navigation as a sequence prediction problem over images and poses. This enables a single visual backbone to drive diverse mobile platforms across varied environments, significantly facilitating the rollout of new robots in novel sites. This scalability is further demonstrated by City Walker (Liu et al., 2025b), which learns policies that transfer across road networks in different cities.

For ensuring real-time feasibility on embedded hardware, particular focus has been given on model efficiency. In particular, FASTNav (Chen et al., 2024c) utilizes compact, LoRA-adapted language models to provide instruction-following capabilities that run on-board in real-time. Additionally, frameworks like NaviLLM (Zheng et al., 2024a) unify instruction following with embodied question-answering to provide a flexible interface for operators.

In road environments, FMs enhance safety and transparency, by conditioning planning on linguistic reasoning or graph-based visual question answering. Frameworks like DriveGPT-4 (Xu et al., 2024b), Drive Anywhere (Wang et al., 2024c), and DriveLM (Sima et al., 2024) go beyond black-box end-to-end policies, by producing human-readable rationales and intermediate decisions that are easier to inspect, to debug, and to align with safety standards. For assistive scenarios, DRAGON (Liu et al., 2024d) uses dialogue-based navigation to allow users to describe targets and constraints in natural language, where the robot provides verbal explanations of its planned route.

## H.2 Industrial manipulation

FMs are fundamentally redefining industrial automation by replacing rigid, task-specific routines with flexible, general-purpose agents that excel in unstructured manufacturing environments. In particular, models like RFM-1 (Sohn et al., 2024) are deployed within high-throughput production cells, leveraging visuo-motor backbones trained on millions of real-world pick actions to achieve zero-shot generalization, when encountering novel objects. For large-scale logistics, PRIME-1 (Inc., 2025) provides a foundation tailored to parcel workflows, facilitating site-specific adaptation and the rapid rollout of autonomous systems across diverse distribution centers. In order to address hardware variability, OpenVLA (Kim et al., 2025) utilizes parameter-efficient fine-tuning to simplify transfer across different robotic platforms, effectively reducing the need for extensive per-line retraining, while enabling operators to issue high-level, language-conditioned tasks. Moreover, RT-2 (Zitkovich et al., 2023) capitalizes on web-to-robot knowledge transfer to significantly boost robustness against long-tail objects and complex instructions on assembly and kitting lines, ensuring that robots can adapt to items outside their original training data.

## H.3 Supply operations

FMs are drastically evolving autonomous supply operations, by providing robots with the flexibility and intelligence needed to handle unstructured, dynamic, and complex logistics tasks. This paradigm shift allows robotic agents to adapt to new items, facility layouts, and operational disruptions without requiring manual reprogramming. At a broader scale, logistics operations utilize FMs for sustainable planning, multi-agent coordination, and end-to-end decision support, specifically targeting 'Logistics 5.0' goals, such as greener routing (Nicoletti & Appolloni, 2024). By integrating multi-agent systems with foundation backbones, autonomous supply chains can effectively link demand forecasting, resource allocation, and transport routing through shared representations and agentic reasoning (Xu et al., 2024a). Consequently, FM-driven solutions can support a wide array of critical industrial tasks, including demand sensing, inventory positioning, warehouse tasking, and transport orchestration (Nicoletti, 2025; Agaskar et al., 2025).

## H.4 Service robots

The integration of FMs represents a major paradigm shift for service robots, transforming them to adaptable agents, capable of navigating through unstructured domestic environments and interpreting natural language instructions. In this context, TidyBot (Wu et al., 2023a) exemplifies the use of few-shot LLM reasoning to learn personalized tidying preferences directly from user conversations, allowing a mobile manipulator to ground abstract 'put-away' commands into specific physical actions across novel scenes. Extending these capabilities to more complex, multi-step procedures, ELLMER (Mon-Williams et al., 2025) performs tasks such as coffee brewing, by integrating visual, force, and linguistic feedback; specifically, ELLMER utilizes a retrieval-augmented memory mechanism to adapt its plans on the fly if environmental conditions change or failures are detected, ensuring robust execution under uncertainty.

## H.5 Medical robots

FMs in the medical domain serve as a critical bridge between high-level clinical knowledge and low-level robotic execution. These models are increasingly integrated into surgical perception and decision-making pipelines to handle high-stake, high-variability tasks. A key application is monocular depth estimation; for instance, Surgical-DINO (Cui et al., 2024) adapts DINOv2 using a LoRA adapter specifically for endoscopic imagery. Similarly, DARES (Zeinoddin et al., 2024) tailors a Depth Anything Model, using self-supervised Vector-LoRA, to better align with surgical scene statistics. Additionally, EndoDDC (Lin et al., 2026) addresses sparse-to-dense depth reconstruction for endoscopic robotic navigation through diffusion-based depth completion, supporting more reliable 3D reconstruction and safe instrument guidance. These advancements allow for the robust implementation of surgical assistance systems, including real-time instrument tracking, tool detection, and intra-operative guidance within hospital settings (He et al., 2024). Ultimately, this integration provides context-aware intelligence and precise assistance, enabling robotic platforms to robustly support complex clinical procedures.

### H.6 Cognitive agrisystems

The integration of FMs into cognitive agrisystems marks a transition towards adaptive, embodied intelligence in field operations. By utilizing VLA architectures, these systems can ingest high-level semantic instructions and translate them into precise motor outputs in real-time. Unlike traditional agricultural robots that rely on hard-coded rules, FMs bridge the semantic gap, by using massive pre-trained datasets to reason through the unpredictable variability of biological environments, such as shifting light, overlapping foliage, and irregular crop shapes (Yin et al., 2025b). For instance, HarvestFlex (Zhao et al., 2026) employs VLA policies for real greenhouse tabletop strawberry harvesting, a long-horizon, unstructured task, challenged by occlusion and specular reflections. Additionally, FM-based reasoning supports task planning and action selection in crop monitoring and field-management scenarios (Cuaran et al., 2026).

### H.7 Crisis agents

FMs are fundamentally reforming disaster response and public safety, by enabling crisis agents to transition from rigid, remote-controlled setups to autonomous systems capable of high-level reasoning in unpredictable and hazardous environments. In particular, SafeGuard ASF (Canh et al., 2026) combines multi-modal hazard perception with agentic reasoning for real-time fire-risk detection and disaster recovery. Additionally, a robotic fire-risk detection system based on dynamic knowledge graphs and LLM-enhanced multi-modal reasoning is developed (Pan et al., 2025a), demonstrating how FM-based reasoning can support emergency response in safety-critical settings.

### H.8 Maritime robotics

The incorporation of FM intelligence has significantly bolstered the capabilities of autonomous maritime systems, allowing them to perceive, to reason, and to act more effectively within complex aquatic environments. These models are specifically engineered to navigate typical underwater challenges, such as high turbidity, limited visibility, and severe communication constraints that often degrade traditional robotic sensors. In particular, UnderwaterVLA (Wang et al., 2025e) introduces a dual-brain VLA architecture for autonomous underwater navigation, combining multi-modal reasoning with embodied control for improving robustness under degraded visual and communication conditions. Additionally, MarineInst (Zheng et al., 2024b) and MarineGPT (Zheng et al., 2023) demonstrate FM capabilities in bridging raw marine visual data, semantic understanding, and domain-specific natural-language knowledge, thereby supporting richer perception and reasoning modules for maritime robotic platforms.

### H.9 Space robotics

The integration of FMs is critically transforming the field of astro-embodied intelligence, enabling robotic agents to reason, to adapt, and to perceive within unstructured, off-world environments, where human intervention is physically impossible. These models provide strong priors and zero-shot generalization capabilities that are crucial for operating under the extreme conditions and data scarcity typical of planetary missions. A primary application involves the usage of SAM for universal crater detection (Giannakis et al., 2024), which utilizes promptable segmentation to identify features across diverse planetary imagery without requiring domain-specific retraining. Beyond basic detection, such FMs are being extended to facilitate autonomous terrain understanding and complex geological analysis (Giannakis et al., 2024; Zhao & Ye, 2024; Holden et al., 2026), allowing robots to make high-stake decisions independently in remote and hazardous space settings.

## I  Public datasets

This section provides the complete dataset catalog summarized in Section 11 of the main paper. In particular, Tables 2-6 group the various benchmarks with respect to the main robotic task concerned, while they also include information about the following aspects for each entry: a) Year, b) Scale, c) Semantic classes, d) Modalities involved, e) Annotation type, f) Domain, g) Environment type (Simulation (S), Real (R)), h)

Temporality (Static (ST), Video (V), Sequential (SE)), i) Embodiment (Yes (Y), No (N)), and j) Short description. Apart from per task remarks, the following global observations can be made:

- Focus on massive cross-embodiment datasets: In order to address the need for constructing generalist policies that can perform robustly under different kinematic structures and environmental conditions, dataset creation activities have shifted from specialized, single-robot setups to large-scale, multi-robot, heterogeneous ones. For example, Open X-Embodiment (O'Neill et al., 2024) and AgiBot World (Bu et al., 2025) contain over a million trajectories across dozens of robotic platforms.

- Emphasis on vision-language-action capturings: Following the research trend of developing embodied VLA solutions, data collection procedures increasingly support sensorial types that aim at integrating perception, reasoning, and control actions. For example, BridgeData V2 (Walke et al., 2023) and CALVIN (Mees et al., 2022) include synchronized information streams of RGB-D video, natural language instructions, and low-level action tokens.

- Translation to high-fidelity, real-world capturing settings: In order to robustly address the sim-to-real gap in the FM era, intense efforts have been devoted on creating massive, real-world benchmarks, instead of simulation ones. In this respect, datasets like Ego4D (Grauman et al., 2022) and DROID (Khazatsky et al., 2024) support geographic and environmental diversity at unprecedented scale.

- Incorporation of human reasoning aspects: In order to enable FMs to develop common-sense, human-like reasoning and interaction capabilities, such aspects are increasingly incorporated in the dataset formation procedures. For instance, RoboVQA (Sermanet et al., 2024) and TEACh (Padmakumar et al., 2022) include hundreds of thousands of video-text pairs and clarification-oriented Q&A dialogues.

- Lack of tactile sensing data: Despite the high availability of visual and textual data, tactile (touch) and force-sensing information streams remain critically under-represented in the current benchmarks. Incorporation of the aforementioned modalities is essential for developing efficient, high-precision dexterous manipulation.

- Lack of failure and recovery data: A common observation across all datasets comprises the typical lack of recordings corresponding to rare failure modes and successful recovery actions. The latter is also essential during the training phase for developing models that are likely to be robust in real-world execution settings.

Table 2: Public datasets concerning in principle perception tasks. Abbreviations used: Environment type (Simulation (S), Real (R)), Temporality (Static (ST), Video (V), Sequential (SE)), and Embodiment (Yes (Y), No (N)).

| Dataset | Year | Scale | Classes | Modalities | Annotation type | Domain | Env. type | Temp. | Emb. | Description |
|---|---|---|---|---|---|---|---|---|---|---|
| **Matterport3D** (Chang et al., 2017) | 2017 | • 90 building scenes
• 10,800 panoramas
• 194,400 RGB-D images
• 50,811 object instances
• 38,328 3D bounding boxes | • 20 structural
• 20 objects | • RGB-D panoramas
• 3D meshes
• Camera poses | • 3D reconstruction
• 2D masks
• 3D semantic & instance labels | Indoors | R | ST | N | RGB-D dataset of entire buildings for global scene understanding |
| **2D-3D-S** (Armeni et al., 2017) | 2017 | • 6K $m^2$ coverage
• 270 rooms / 6 areas
• 70,496 regular RGB images
• 1,413 equirectangular RGB images
• 695M 3D points | • 7 structural
• 6 objects | • RGB-D images
• Surface normals
• 3D meshes & point clouds
• Camera poses & metadata | • 2D & 3D semantic & instance labels
• 3D bounding boxes
• Room categorizations | Indoors | R | ST | N | Jointly registered 2D and 3D data for office-style indoor scene understanding |
| **Semantic–KITTI** (Behley et al., 2019) | 2019 | • 22 sequences
• 43,552 LiDAR scans
• 360° FOV coverage | • 11 structural
• 9 objects
• 8 actors | • LiDAR point clouds
• 3D point trajectories
• Sensor poses | • Point-wise semantic labels
• Sequence-level ID tracking
• Scene completion targets | Driving | R | SE | Y | Large-scale LiDAR driving dataset with point-level semantic labels for sequences |
| **BDD100K** (Yu et al., 2020) | 2020 | • 100K HD video clips
• 40M video frames
• 100M total distance (km)
• 10 vision tasks | • 11 structural
• 8 objects | • RGB video
• GPS/IMU metadata
• Time/Weather/Lighting tags | • 2D bounding boxes
• Instance & driveable masks
• Lane marking
• Object tracking | Driving | R | V | Y | Diverse driving dataset covering varying weather, times, and city environments |
| **nuScenes** (Caesar et al., 2020) | 2020 | • 1K scenes
• 1.4M images
• 390K LiDAR
• 1.4M radar sweeps
• 1.4M 3D object boxes | • 9 structural
• 23 objects | • 6 RGB cams (360°)
• 32-beam LiDAR
• 5 long-range radars
• GPS/IMU metadata | • 3D bounding boxes
• LiDAR semantic labels | Driving | R | SE | N | Driving dataset providing full AV sensor suite (LiDAR, radar, 6 cameras) |
| **Waymo Open** (Sun et al., 2020) | 2020 | • 1K sequences
• 200K LiDAR frames
• 1M RGB images
• 12M 3D boxes
• 12.6M 2D boxes | • 15 structural
• 13 objects
• 8 actors | • 5 LiDAR
• 5 RGB cameras
• Sensor poses | • 2D/3D tracking boxes
• Global 3D point IDs
• Cross-camera 2D labels | Driving | R | SE | N | High-resolution, multi-sensor driving data focused on perception and motion prediction |
| **Ego4D** (Grauman et al., 2022) | 2022 | • 3,670 hours of video
• 931 participants
• 74 locations
• 9 countries
• 5M+ episodic annotations | • N/A (open) | • Egocentric RGB video
• Multi-channel audio
• 3D eye gaze
• IMU
• Stereo | • Episodic memory tags
• Hand-object interactions
• Audio-visual diarization
• Social interaction labels | Daily life | R | V | N | Large-scale egocentric video captured by hundreds of people worldwide |
| **HM3D-SEM** (Yadav et al., 2023) | 2022 | • 216 3D spaces
• 3,100 rooms
• 142,646 object instances
• 14,200+ human hours | • 12 structural
• 28 objects | • Textured 3D meshes
• Semantic textures
• Room-level metadata | • Pixel-level semantics
• 40 Matterport categories
• Instance-level labels | Indoors | R | ST | N | Dataset of semantically annotated building-scale 3D indoor reconstructions |
| **ScanNet++** (Yeshwanth et al., 2023) | 2023 | • 460 scenes
• 1,858 laser scans
• 280K 33MP DSLR images
• 3.7M RGB-D frames | • 50 structural
• 950+ objects | • RGB-D video
• 3D meshes
• Camera poses | • 3D semantic & instance labels
• Multi-label ambiguity tags | Indoors | R | ST | N | Sub-millimeter fidelity indoor scans paired with high-resolution 33MP DSLR imagery |

Table 3: Public datasets concerning in principle planning tasks. Abbreviations used: Environment type (Simulation (S), Real (R)), Temporality (Static (ST), Video (V), Sequential (SE)), and Embodiment (Yes (Y), No (N)).

| Dataset | Year | Scale | Classes | Modalities | Annotation type | Domain | Env. type | Temp. | Emb. | Description |
|---|---|---|---|---|---|---|---|---|---|---|
| **AI2-THOR** (Kolve et al., 2017) | 2017 | • 120 room scenes
• 4 room categories
• 3,578 interactive objects | • 100+ object types | • Egocentric RGB-D videos
• Object metadata | • Object state changes
• Navigation actions
• Arm manipulation | House-hold | S | SE | Y | Near photo-realistic 3D environments for visual AI agents navigation and interaction |
| **Virtual-Home** (Puig et al., 2018) | 2018 | • 2,821 action programs
• 6 furnished houses
• 357 objects per house | • ~300 objects
• 70 actions | • Natural language
• Synthetic video
• 3D poses | • Action programs
• Timestamps
• Atomic interactions | House-hold | S | SE | Y | Simulation of complex daily activities via executable programs and scripts |
| **BabyAI** (Chevalier-Boisvert et al., 2019) | 2018 | • 19 difficulty levels
• $2.48 \times 10^{19}$ instructions | • 6 object types | • Synthetic language
• 2D grid | • Expert demonstrations
• Sub-goal decompositions | Indoors | S | SE | Y | Dataset focused on sample efficiency and grounded language learning in grid worlds |
| **ALFRED** (Shridhar et al., 2020) | 2020 | • 120 indoor scenes
• 25,743 language directives
• 8,055 expert demos | • 7 task types
• 80 objects | • Egocentric RGB videos
• Natural language | • High/low-level instructions
• Action sequences
• Pixel-wise interaction masks | House-hold | S | V,SE | Y | Mapping of natural language to sequences of actions for visual AI agents |
| **CALVIN** (Mees et al., 2022) | 2022 | • 4 environments
• 389 instructions | • 34 tasks | • RGB-D videos
• Vision-based tactile
• Proprioception
• Natural language | • Language goals
• Pre-task locomotion behavior
• Precomputed MiniLM embeddings | Tabletop | S | SE | Y | Long-horizon, language-conditioned policy learning for continuous control |
| **BEHAVIOR-1K** (Li et al., 2023a) | 2024 | • 50 scenes | • 1K activities
• 1.2K objects | • Egocentric RGB-D videos
• Proprioception | • Logic (BDDL)
• Transition rules
• Semantic properties | House-hold | S | SE | Y | Human-centered activities with realistic physics and state changes |
| **LAMBDA** (Jaafar et al., 2025) | 2024 | • 31 rooms
• 8 environments | • 571 tasks | • Natural language
• Egocentric RGB-D videos
• Robot poses & actions | • Human-collected trajectories
• Semantic maps | House-hold | S,R | SE | Y | Focus on data-efficiency for multi-room, multi-floor mobile manipulation |

Table 4: Public datasets concerning in principle navigation tasks. Abbreviations used: Environment type (Simulation (S), Real (R)), Temporality (Static (ST), Video (V), Sequential (SE)), and Embodiment (Yes (Y), No (N)).

| Dataset | Year | Scale | Classes | Modalities | Annotation type | Domain | Env. type | Temp. | Emb. | Description |
|---|---|---|---|---|---|---|---|---|---|---|
| **REVERIE** (Qi et al., 2020) | 2020 | • 10,567 panoramas
• 86 scenes
• 23,536 instructions
• 4,140 objects | • 489 objects | • Natural language
• RGB panoramas | • Instructions
• Target boxes
• Nav-graph sequences | Indoors | S | SE | Y | Remote embodied visual referring expressions in unseen environments |
| **VLN-CE** (Krantz et al., 2020) | 2020 | • 90 scenes
• 7,189 trajectories
• 21,567 instructions | • N/A (path-based) | • Natural language
• Egocentric RGB-D videos | • Low-level continuous actions
• Navigation paths | Indoors | S | SE | Y | Navigation in continuous environments, emphasizing fine-grained control |
| **RxR** (Ku et al., 2020) | 2020 | • 126,069 instructions
• 16.5M total words
• 3 languages | • N/A (path-based) | • Multilingual text
• Virtual pose traces
• RGB panoramas | • Dense spatiotemporal grounding
• Time-aligned text-to-pose | Indoors | S | SE | Y | Large-scale multilingual vision-and-language navigation |
| **Habitat-Web** (Ramrakhya et al., 2022) | 2022 | • 80K navigation trajectories
• 12K Pick&Place trajectories
• 29.3M actions
• 22,600 hours | • 21 object trajectories | • Egocentric RGB-D videos
• Teleoperation traces | • Human task trajectories
• Implicit search heuristics | Indoors | S | SE | Y | Imitation learning from large-scale human demonstrations collected on the web |
| **ScaleVLN** (Wang et al., 2023) | 2023 | • 1.2K+ scenes
• 4.9M trajectories | • N/A (path-based) | • Synthesized language
• Egocentric RGB-D videos | • Synthesized trajectory-instruction pairs | Indoors | S | SE | Y | Navigation using automatically generated large-scale synthetic data |
| **GOAT-Bench** (Khanna et al., 2024) | 2024 | • 90 scenes
• 9 tasks
• 3K+ goal entities | • N/A (open-vocabulary) | • Multi-modal goals
• Egocentric RGB-D videos | • Target object locations
• Sequential goal sequences | Indoors | S | SE | Y | Lifelong navigation to open-vocabulary goals |
| **HM3D-OVON** (Yokoyama et al., 2024) | 2024 | • 216 scenes
• 15K+ object instances | • 379 objects | • Free-form language
• Egocentric RGB-D videos | • Open-vocabulary object goals
• 3D bounding boxes | Indoors | S | SE | Y | Open-vocabulary object goal navigation, based on free-form natural language |

Table 5: Public datasets concerning in principle manipulation tasks. Abbreviations used: Environment type (Simulation (S), Real (R)), Temporality (Static (ST), Video (V), Sequential (SE)), and Embodiment (Yes (Y), No (N)).

| Dataset | Year | Scale | Classes | Modalities | Annotation type | Domain | Env. type | Temp. | Emb. | Description |
|---|---|---|---|---|---|---|---|---|---|---|
| **RoboNet** (Dasari et al., 2020) | 2019 | • 15M frames 
 • 162K trajectories 
 • 4 locations | • N/A (pushing) | • RGB videos 
 • Robot actions 
 • Gripper states | • Action trajectories 
 • Video targets | Tabletop | R | SE | Y | Multi-robot dataset on learning visual foresight and video prediction for non-prehensile manipulation |
| **RLBench** (James et al., 2020) | 2020 | • Infinite demonstrations | • 100+ tasks | • RGB-D videos 
 • Segmentation 
 • Proprioception | • Motion-planned trajectories 
 • Target way-points | Tabletop | S | SE | Y | Tasks algorithmically generated using ground-truth state information |
| **CALVIN** (Mees et al., 2022) | 2022 | • 4 environments 
 • 2.4M interaction steps | • 34 tasks | • RGB-D videos 
 • Tactile 
 • Proprioception | • Language goals 
 • Pre-task locomotion | Tabletop | S | SE | Y | Long-horizon language-conditioned continuous control |
| **LIBERO** (Liu et al., 2023) | 2023 | • 6.5K trajectories | • 130 tasks | • RGB video 
 • Proprioception 
 • Language instructions | • Expert demonstrations 
 • Task completion tags | Tabletop | S | SE | Y | Evaluation of knowledge transfer across sequentially learned task suites |
| **RH20T** (Fang et al., 2024) | 2023 | • 110K trajectories | • 150+ skills | • RGB video 
 • Force 
 • Tactile 
 • Audio | • Action trajectories 
 • Language descriptions | Tabletop | R | SE | Y | Multi-modal dataset including force and audio, targeting contact-rich skills |
| **BridgeData V2** (Walke et al., 2023) | 2023 | • 60,096 trajectories 
 • 24 environments | • 13 skills | • RGB videos 
 • Proprioception | • Goal images 
 • Language instructions | Tabletop | R | SE | Y | Use of low-cost robots across 24 environments to boost generalization |
| **Open X-Embodiment** (O'Neill et al., 2024) | 2024 | • 1M+ episodes 
 • 60 datasets | • 500+ skills | • RGB video 
 • End-effector poses 
 • Language instructions | • Action trajectories | Multi-domain | R,S | SE | Y | Aggregation of 60+ datasets and 22 robotic platforms into a unified format for cross-embodiment scenarios |
| **DROID** (Khazatsky et al., 2024) | 2024 | • 76K trajectories 
 • 564 scenes | • 86 tasks | • RGB-D videos 
 • Proprioception | • Teleoperated actions 
 • Language instructions | In-the-wild | R | SE | Y | In-the-wild dataset collected by 50 people across 52 buildings to maximize scene and lighting diversity |
| **RoboMIND** (Wu et al., 2025a) | 2024 | • 107K trajectories 
 • 96 objects | • 279 tasks | • RGB-D videos 
 • Proprioception 
 • Natural language | • Expert teleoperation 
 • Failure demonstrations 
 • Fine-grained instructions | Indoors | S,R | SE | Y | Unified-standard dataset covering humanoids and dual-arm robots for multi-embodiment intelligence |
| **AgiBot World** (Bu et al., 2025) | 2025 | • 1,001,552 trajectories 
 • 3K+ objects | • 217 tasks | • RGB-D video 
 • Visuo-tactile 
 • Proprioception | • Human-in-the-loop teleoperation actions | Multi-domain | S,R | SE | Y | Large-scale facility-based platform using 100 humanoid robots to collect high-fidelity, bi-manual, long-horizon task data |

Table 6: Public datasets concerning in principle human-robot interaction tasks. Abbreviations used: Environment type (Simulation (S), Real (R)), Temporality (Static (ST), Video (V), Sequential (SE)), and Embodiment (Yes (Y), No (N)).

| Dataset | Year | Scale | Classes | Modalities | Annotation type | Domain | Env. type | Temp. | Emb. | Description |
|---|---|---|---|---|---|---|---|---|---|---|
| **CVDN** (Thomason et al., 2020) | 2020 | • 2,050 human dialogues • 7K+ trajectories • 83 scenes | • N/A (goal-driven) | • Natural language dialogues • RGB panoramas | • Dialogue history • Shortest-path actions • Navigation traces | Household | S | SE | Y | Multi-turn human-human dialogues for navigation, where agents ask for help |
| **TEACh** (Padmakumar et al., 2022) | 2022 | • 3,047 dialogues • 39.5K utterances | • 12 tasks | • Natural language dialogues • Egocentric RGB videos • Discrete actions | • Dialogue history • Human demonstrations • Object state changes | Household | S | V,SE | Y | Task-driven agents that communicate to complete complex household tasks |
| **DialFRED** (Gao et al., 2022) | 2022 | • 53K Q&A pairs | • 25 subgoals | • Natural language dialogues • Egocentric RGB videos | • Q&A pairs • Action sequences • Oracle responses | Household | S | V,SE | Y | Active questioning framework where agents ask humans for clarifications to solve household tasks |
| **RoboVQA** (Sermanet et al., 2024) | 2023 | • 829,502 video-text pairs • 29,520 instructions | • N/A (open-ended QA) | • RGB videos • Natural language • Robot actions | • VQA pairs • Multi-embodiment demonstrations | Daily life | R | V | Y | Large-scale reasoning dataset using interleaved vision-text-action for long-horizon robot planning |
| **NatSGD** (Shrestha et al., 2025) | 2024 | • 1,143 commands • 18 participants | • 11 actions • 20 objects | • Speech • Audio • Gestures • Robot actions | • Intent labels • Time-aligned behavior | Cooking | S,R | V,SE | Y | Synchronized speech, gestures, and robot demonstrations for natural human-robot interaction |
| **HA-R2R** (Li et al., 2024c) | 2024 | • 21,567 instructions • 486 motion sequences | • 145 activities | • Natural language • RGB-D & fisheye videos • Human activity data | • Human activity descriptions • Navigation trajectories | Indoors | S,R | V,SE | Y | Human-aware navigation focusing on social constraints and dynamic human interactions |