# OpenReview forum: "Foundation Models in Robotics: A Comprehensive Review of Methods, Models, Datasets, Challenges and Future Research Directions"
_TMLR — Accepted by TMLR_

### Review · Reviewer_ayRD · 2026-04-29

**Summary Of Contributions:**

This survey covers a few hundreds of publications from the last ~5 years in the space of foundation models in robotics. Motivated by significant recent advancements, the author motivate the need for a survey that covers several types of FMs (LLMs, VLMs, VLAs, …) and systematically classifies and analyses existing works. This is what the authors try to accomplish. First, they briefly summarize their methodology in navigating the literature, then propose a chronological classifications spanning from 2018 to today. The literature is then organized and presented along 7 axes: the type of FM (determined by the modalities it handles), the nature of its architecture, the learning paradigm it conforms to, the different phases of training each method tackles, the general robotic problem it solves, the various application domains models are applied to, and finally the datasets that have so far been published. The survey ends in a relatively brief conclusion.

**Strengths:**
- To the best of my knowledge, this is the broadest survey of FMs to date.
- On top of presenting existing works, the author propose a taxonomy and a chronological interpretation of recent developments.
- The survey is overall well-written, and clear enough to be understandable by newcomers.

**Main weaknesses and questions:**
- The conclusion are somewhat short in comparison to the rest of the survey. While a large amount of works is summarized, this survey would be of much more impact if it also elaborated at length on open problems and how they related to existing works.
- To what extend were LLMs used in writing this survey? While LLM-assisted writing is not an issue per-se, the authors should clearly describe if and how LLMs were leveraged, particularly in a survey of this size.
- Listing “Pre-training” as both a learning paradigm and a learning phase is somewhat confusing. To the best of my knowledge, most of the works cited for pre-training as a learning paradigm are based on standard supervised learning - would it be best to change the name to supervised learning?
- Following from the above, on page 34 the authors state that “The most common warning paradigms adopted during the pre-training stage are: pre-training […]”. This statement is circular, and focusing on the supervised objective would be clearer in my opinion.
- The survey does not list its own limitations - while it may be broader than existing ones, what type of topics should future surveys span?

**Other comments:**
- I am not sure whether the query used for literature review are broad enough. The didactic queries presented in Section 2.2 would for instance ignore important work such as GATO, to the best of my understanding. The authors should discuss why this mechanism is preferable to a semantic search over paper embeddings, for instance.
- The milestones in Section 3.1 are insightful, but it would be best to describe how they were chosen. Do they align with timelines suggested by previous works?
- The Open-X-Embodiment dataset, which is arguably one of the most important ones, is mentioned but not referenced to properly.

**Audience:**

Yes

**Audience Explanation:**

I think that this submission constitutes an good point of contact with FMs in robotics, particularly for people who are unfamiliar with either robotics or FMs, but have a good grasp of the other.

**Broader Impact Concerns:**

I do not think this statement is necessary.

**Claims And Evidence:**

Yes

**Claims Explanation:**

Despite a few weaknesses highlighted above, the survey is extensive and accurate. No methodological claims are made, as it is a survey.

**Requested Changes:**

My five requested changes are listed under "Main weaknesses".

---

> ### Author Response · Authors · 2026-06-13
> **Response to Reviewer ayRD comments**
>
> We appreciate the feedback provided by the reviewer. In response, the revision focused on the following aspects per reviewer Comment (C):
> - C1: We agree that the original conclusion section incorporated a relatively dense discussion on (few) open challenges and main potential future research directions, without tying them to specific literature entries. In the revised version, the Conclusion (Section 14) has been substantially expanded and restructured, in order to describe in a more comprehensive way the field's main open problems, while connecting each of them explicitly to existing works that aim to address it (the open problems are now organized into clusters tied to the reviewed works via the C-labels and citations).
> - C2: Check the answer to the second comment of Reviewer Yc2F (the dedicated LLM-assistance statement has been added to Section 2).
> - C3: We agree with this observation that using the same 'pre-training' term leads to confusion. Following the reviewer's suggestion, we have renamed the 'Pre-training' paradigm to 'Supervised learning' in the learning paradigms section (Section 7), since the cited methods/works are trained with supervised objectives. Additionally, we maintained the 'pre-training' term only in the learning stages section (Section 8), where it is now better clarified that pre-training comprises a stage of the training pipeline that can be carried out using different learning paradigms (e.g., supervised, self-supervised, imitation).
> - C4: This is a continuation of the revisions carried out for C3. In the revised version (where the term 'pre-training' is only used in Section 8), the highlighted bullet point (previously 'Pre-training: Supervised pre-training aims at…') has been renamed 'Supervised learning: …' and reworded to emphasize the supervised objective.
> - C5: In response to this reviewer's comment, we have added in the conclusion section a 'Limitations of current survey' paragraph and a 'Scope for future surveys' one. These state clearly what we do not cover (or cover only partly), e.g., hardware and low-level control theory aspects, tactile/auditory and other under-represented modalities, comprehensive quantitative benchmarking/analysis, etc. Additionally, we also included concrete suggestions for future surveys (e.g., dedicated surveys on world models for robotics, cross-embodiment benchmarking, safety verification of embodied FMs, etc.).
> - C6: This reviewer's comment has been addressed along the following two axes: a) The Scopus-style string (now provided in Section B of the supplementary document) is only an illustrative example search and not the complete literature search protocol. The actual search process combined six databases, was applied iteratively with keyword refinement, and was supplemented by an extensive backward reference-checking/chaining procedure. This overall literature review process is considered (relatively) robust in not missing critical/important works in the field, achieving also to include works like Gato (already incorporated in Table 2). This statement has been made clearer in Section 2 of the revised manuscript. b) We have added a brief justification on using a keyword plus reference search approach over pure embedding-based semantic retrieval, explaining that the adopted methodology favours reproducibility, transparency, and precision/recall control aspects. Nevertheless, it is clearly stated now that semantic search can facilitate as a complementary tool.
> - C7: We have added a short paragraph in Section 3 explaining the main selection criteria for the identified milestone works (e.g., introduction of a new capability or modality, demonstrable influence measured by adoption/citations and follow-up systems, and first work for a given phase, etc.). Additionally, we compared our five-phase timeline against corresponding schemes in previous works and surveys (e.g., Kawaharazuka et al., Ma et al., Sapkota et al.), discussing upon their alignment/difference.
> - C8: In the revised manuscript we have corrected the improper citation (incorrect title in the bibliography/BibTeX entry) to the 'Open x-embodiment' dataset (the entry now has the correct title, full author list, and ICRA 2024 venue).

---

### Review · Reviewer_Yc2F · 2026-06-09

**Summary Of Contributions:**

This report is a comprehensive summary of recent advances in foundation models for robotics. It discusses the five phases of the development of these foundation models, and includes most types of foundation models, including LLMs, VFMs and VLMs that were not included in some previous summaries. It also includes a detailed summary of the training paradigms, learning stages, and tasks for training.

**Audience:**

Yes

**Audience Explanation:**

I think early-career researchers in robotics and foundation models will find this summary useful. However, I have not read other similar summaries, so I am not sure about the key difference between this submission and the existing summaries. From Section 2, it seems that this summary includes more papers and things that were not included in existing ones, such as LLMs.

**Claims And Evidence:**

Yes

**Claims Explanation:**

This submission is a very lengthy summary, so I am not able to read it line by line. As mentioned in Section 2.2, this summary includes all literature obtained from a database search. I think this submission could serve as a catelogue for researchers who just enter this field. This submission is also good material for LLM agent consumption. So while I believe such a report is meaningful, whether or not it should be accpeted by TMLR depends on the scope of TMLR.

**Requested Changes:**

I feel that a large part of this manuscript is written by LLM agents, so the format is not very friendly to humans. For example, the tables in the manuscript are very wordy and hard to read. I suggest the authors to make the report more concise, for example by running an agent to do that.

---

> ### Author Response · Authors · 2026-06-11
> **Response to Reviewer Yc2F comments**
>
> We appreciate the feedback provided by the reviewer. In response to the raised comments, the revision focused on the following three axes:
> 1) Conciseness: A global restructuring (as suggested also by Reviewer hZDj) has been performed, directly targeting the length and density aspects of the paper. In particular, the main part of the paper has been substantially reduced (focusing more on synthesis and comparison over enumeration aspects), while parts of the original submission (e.g. extensive list of references) have been moved to appropriate sections of the supplementary document.
> 2) Tables: The original consideration when formulating the table contents was for them to be self-contained, so that for the potential reader not to be necessary to check/visit the whole paper to understand. However, in the revised version, we have rewritten/reduced wordy tables towards a more compact, keyword-based style (e.g., Table 1, Table 2, and the per-criterion comparative analysis tables), with the full detailed tables moved to the supplementary document, so that they can be more easily studied.
> 3) Readability: We have performed a full human editing pass for improving clarity and readability of the overall manuscript.
>
> Regarding LLM assistance: We have added a dedicated statement in the methodology section (Section 2) describing the specific role of automated tools (LLMs) in preparing this manuscript. Specifically, LLM assistance was only used in certain/specific occasions and solely as a complementary aid for cross-checking potential gaps in the defined literature search methodology (i.e., as a supplementary means of identifying relevant works that might have been missed by the database queries).

---

### Review · Reviewer_hZDj · 2026-06-09

**Summary Of Contributions:**

The paper provides a comprehensive survey of foundation models in robotics by tracing the evolution of the field, organizing existing work through a multi-axis taxonomy and identifying key open challenges and future research directions.

**Audience:**

Yes

**Audience Explanation:**

The topic is timely and relevant to TMLR’s audience, especially researchers interested in foundation models, robot learning, multimodal learning, and embodied AI. The survey provides a broad synthesis of recent work and identifies useful taxonomic distinctions, datasets, challenges, and future directions.

**Claims And Evidence:**

Yes

**Claims Explanation:**

The submission is generally well supported by a broad and systematically organized body of literature. The taxonomy, dataset discussion, and challenges/future directions are convincing, and the paper’s main claims about the growing role of foundation models in robotics are well aligned with recent developments.

**Requested Changes:**

1. **Shorten the manuscript**. The paper is comprehensive and useful, but it is currently very long. I recommend separating the main conceptual synthesis from the more exhaustive cataloging of papers, datasets, and methodological details.

2. **Move most of Section 2 to the supplementary material**. The main paper should retain only a concise summary of the review methodology. The full details can be moved to the supplement.

3. **Condense Table 1**. The comparison with prior surveys is useful, but the table is not concise. I suggest to include a summary of Table 1 in the main text (using keywords rather than sentences to describe pros and cons). Then the full table can be moved to the supplementary material.

4. **Shorten the taxonomy discussion in Sections from 5 to 10.** These sections are informative, but many subsections follow a similar structure and include long lists of representative works. The main paper would be stronger if it emphasized the key conceptual distinctions, comparative insights, and takeaways, while moving exhaustive paper-by-paper descriptions to the supplement.

5. **Reduce the dataset catalog in Section 11.** The main text should summarize the main dataset families, their uses, and current gaps. Detailed dataset tables or extended descriptions could be moved to the supplementary material.

6. **Consider merging or tightening Sections 12 and 13.** Challenges and future directions are naturally connected. Presenting each challenge together with the corresponding research direction would make the discussion more compact and easier to follow.

7. **Verify the labels in Table 2.** Some “Emb.” labels and model-type classifications should be checked carefully. For example, general-purpose models such as CLIP or SAM are widely used in robotics, but they are not embodied robotic models themselves. Similarly, systems such as SayCan, SayPlan, Eureka, or DrEureka may be better described as FM-based robotic systems rather than standalone foundation models.

---

> ### Author Response · Authors · 2026-06-12
> **Response to Reviewer hZDj comments**
>
> We appreciate the feedback provided by the reviewer. In response, the revision focused on the following aspects per reviewer Comment (C):
> - C1: We agree with this recommendation, which has been highlighted by all three reviewers and has led to a substantial restructuring of our paper. During the revision, the main part of the paper now maintains its conceptual contributions, while extensive/detailed listings of papers and datasets have been moved to appropriate sections of the supplementary document (organized as Sections A-I, mirroring the main sections). Detailed content re-allocation is described as response to comments C2-C6 below.
> - C2: We maintained in the main text (Section 2) a short paragraph indicating the searched databases, the temporal search window, and the screening process. All remaining materials (e.g., Scopus query, refinement steps, bibliometric figures, etc.) have been moved to Section B (Details of literature review methodology) of the supplementary document.
> - C3: Table 1 (in Section 1) has been replaced with a compact keyword-style version of it, while a short paragraph explains the gap that our survey fills (relative to prior ones) as well as the limitations of our paper (that future surveys can address). The original sentence-based table has been moved to Section A of the supplementary document.
> - C4: For each defined criterion (FM types, NN architectures, etc.), the respective section (Sections 5-10) has been significantly reduced, by maintaining only the following key information: a) Category definition and main usefulness (without enumerations of pros and cons), b) A significantly reduced number of papers/works per main category of methods (e.g., maximum 2-3 methods per category), c) the comparative analysis/insights table/discussion, and d) A representative figure (where applicable). All remaining lengthy descriptions and extensive references have been moved to the corresponding Sections C-H of the supplementary document. In this way, the main survey's analytical contributions are maintained in its main body.
> - C5: The five dataset tables (perception, planning, navigation, manipulation, human-robot interaction) in Section 11 have been moved to Section I of the supplementary document. Additionally, the contents of Section 11 have been replaced by a concise/comprehensive discussion on the main dataset families (e.g., large-scale cross-embodiment trajectory corpora), their uses and their current critical gaps (e.g., scarcity of real-world physical-interaction data).
> - C6: Following the reviewer's suggestion, we have made the connections between Sections 12 (Current challenges) and 13 (Future research directions) more explicit and specific. The main consideration for not merging to a single section is that there is not always an 1-to-1 mapping between each challenge and future direction (and vice versa), which may probably render a single section more difficult to read. In the updated manuscript, each challenge is now assigned an explicit label (C1.1-C6.2) and every future research direction states which challenge(s) it (primarily) addresses.
> - C7: The reviewer comments are accurate. As a response, in the updated manuscript we have: a) Revisited and carefully verified all labels/entries in Table 2, b) Embodiment: Indeed, CLIP and SAM are general-purpose FMs, not embodied robotic models; so, these entries have been corrected (now marked Emb. = N). Additionally, an explicit definition of the 'embodied' FMs has been included in the explanatory text immediately following Table 2 (Section 3, 'Key foundation models') for clarification purposes, denoting models 'trained on, or directly producing, robot-executable actions/observations of a physical or simulated agent', and c) a new 'Category' column has been included in Table 2 to distinguish between 'foundation models' and 'FM-based robotic systems'; SayCan, SayPlan, Eureka, and DrEureka are modular/compositional systems that orchestrate existing FMs rather than standalone FMs themselves (these entries have been corrected, i.e., labelled 'System', in the updated Table 2).

---

### Decision · Action_Editor_kaXo · 2026-07-10

**Recommendation:** Accept as is

**Additional Comments:**

The paper provides a comprehensive and well-structured survey of foundation models in robotics, supported by a systematic review methodology, a broad taxonomy, extensive coverage of datasets and application domains, and a thoughtful synthesis of challenges and future research directions. The revisions successfully addressed reviewer concerns, and the work is likely to become a valuable reference for researchers entering or working in this area.

**Audience:**

Yes

**Audience Explanation:**

This survey addresses a highly active and important research area at the intersection of machine learning, foundation models, and robotics. Its comprehensive coverage of model architectures, learning paradigms, datasets, robotic tasks, application domains, and open challenges will be valuable to researchers working on machine learning, embodied AI, robot learning, multimodal learning, and foundation models. The paper also provides a useful entry point for newcomers and a structured reference for practitioners seeking an overview of the rapidly evolving landscape of foundation models in robotics.

**Claims And Evidence:**

Yes

**Claims Explanation:**

The claims are well supported by a comprehensive literature review. The authors employ a transparent review methodology, analyze a large corpus of publications, and organize the field through a broad taxonomy spanning foundation model types, architectures, learning paradigms, learning stages, robotic tasks, application domains, datasets, challenges, and future directions. The survey provides comparative analyses and synthesized insights, and its main conclusions regarding the growing role of foundation models in robotics, embodied vision-language-action systems, and cross-embodiment generalization are well aligned with the reviewed literature. The revised manuscript improves over the original submission by clarifying the review methodology, correcting taxonomy inconsistencies, and more clearly linking challenges to future research directions. While the paper does not provide independent empirical benchmarking or quantitative analysis and relies on results reported in the surveyed studies, these limitations are acknowledged and do not weaken the survey.